# PROSPECT: LEARN MLPS ROBUST AGAINST GRAPH ADVERSARIAL STRUCTURE ATTACKS

## ABSTRACT

Current adversarial defense methods for GNNs exhibit critical limitations obstructing real-world application: **1)** inadequate adaptability to graph heterophily, **2)** absent generalizability to early GNNs like GraphSAGE used downstream, and **3)** low inference scalability unacceptable for resource-constrained scenarios. To simultaneously address these challenges, we propose PROSPECT, the *first* online graph distillation multi-layer perceptron (GD-MLP) framework for learning GNNs and MLPs robust against adversarial structure attacks on both homophilous and heterophilous graphs. PROSPECT fits into GraphSAGE seamlessly and achieves inference scalability exponentially higher than conventional GNNs. Through decision boundary analysis, we formally prove the robustness of PROSPECT against successful adversarial attacks. Furthermore, by leveraging the Banach fixed-point theorem, we analyze the convergence condition of the MLP in PROSPECT and propose a quasi-alternating cosine annealing (QACA) learning rate scheduler, inspired by our convergence analysis and the alternating iterative turbo decoding from information theory. Experiments on five homophilous and three heterophilous graphs demonstrate the advantages of PROSPECT over current defense methods and offline GD-MLPs in adversarial robustness and clean accuracy[1], the inference scalability of PROSPECT orders of magnitude higher than existing defenders, and the effectiveness of QACA.

## 1 INTRODUCTION

GNNs (Kipf & Welling, 2017; Hamilton et al., 2017; Wu et al., 2021), emerging as the most promising tools for graph data analysis, have been utilized in a wide array of domains, such as drug discovery Jiang et al. (2021), financial risk management (Liu et al., 2021), and recommendation systems (Fan et al., 2019; Pal et al., 2020). Despite their enormous success, GNNs have been shown susceptible, like other deep learning models, to malicious data perturbations known as adversarial attacks (Dai et al., 2018; Zügner et al., 2018; Zügner & Günnemann, 2019). For GNNs, perturbing the graph structure is more destructive than modifying the node features because edges can affect all feature dimensions and the impact of feature manipulation can be easily weakened or even eliminated by the neighborhood aggregation of GNNs (Wu et al., 2019b). Therefore, like almost all prior works, we focus on the adversarial robustness against structure attacks.

Since graph attacks alter properties such as homophily[2], purification defense methods detect and counter adversarial attacks by identifying these induced changes and accordingly purifying the message passing (Gilmer et al., 2017) process. Reported alterations encompass decreased homophily (Wu et al., 2019b; Zhang & Zitnik, 2020; Jin et al., 2020; Li et al., 2022; Zhu et al., 2022), heightened variance in node feature distributions (Zhu et al., 2019), changed singular values of the adjacency matrix (Entezari et al., 2020; Jin et al., 2020), elevated adjacency matrix rank (Jin et al., 2020), and amplified distribution shift (Li et al., 2023). However, most purification methods require computation for presented edges or all node pairs to identify, weaken, or remove suspicious graph components, demanding substantial costs. More critically, these attack patterns stem either from heuristics (Zhu et al., 2019) or investigation limited to homophilous graphs (Wu et al., 2019b; Zhang & Zitnik, 2020; Jin et al., 2020; Entezari et al., 2020; Li et al., 2022; 2023), so the adaptability of them (and

---

[1] Clean accuracy refers to the model's classification accuracy on clean, unperturbed data.
[2] Heterophily/homophily means that neighboring nodes tend to have different/similar labels and features

hence that of the induced defense methods) to heterophilous graphs remains uncertain. Despite this uncertainty, our experiments (in Section 6.1 and Appendix G.2.1) indicate that purification methods fail on heterophilous graphs, suggesting their inadequate adaptability to heterophily.

In fact, there are some attempts to understand and counter adversarial attacks on heterophilous graphs. Zhu et al. (2022) show the distinctions in evasion attack[3] patterns on homophilous versus heterophilous graphs, and claim that some heterophily GNN designs, e.g., those in $H_2$GCN (Zhu et al., 2020), contribute to adversarial robustness. Lei et al. (2022) find successful adversarial evasion attacks enlarge the homophily gap between training and test graphs, and propose spectral EvenNet to generalize across different homophily levels. Though the adversarial robustness is only theoretically proved with an evasion attack setting, both $H_2$GCN and EvenNet are also empirically shown robust against poisoning attacks. Nevertheless, these robust GNNs are designed ad-hoc. Thus, the downstream models for recommendation (Ying et al., 2018; Pal et al., 2020; He et al., 2020) and security (Pujol-Perich et al., 2022) cannot benefit, as they build on early GNNs like SAGE.

Beyond adversarial robustness, inference scalability is also critical for industrial adoption. Inference acceleration for CNN models can be achieved through pruning and quantization to reduce multiplication-and-accumulation (MAC) operations (Cheng et al., 2018). However, inference costs for GNNs arise primarily from neighbor fetching and aggregation. The savings from pruning and quantization are trivial compared to numerous aggregation operations (Zhang et al., 2022). Current purification defense methods and heterophilous GNNs rely on neighbor aggregation during inference and thus have low inference scalability.

To solve the above problems with one bullet, we propose the first online graph distillation MLP (GD-MLP) framework **PROSPECT**, which learns GNNs and ML**P**s **RO**bust again**S**t gra**P**h adv**E**rsarial stru**C**ture at**T**acks. PROSPECT coordinates the online collaborative learning and mutual distillation between a GNN and an MLP. This enables surpassing their individual clean accuracy and adversarial robustness ceilings on both homophilous and heterophilous graphs. We denote the engaged MLP and GNN as Prospect-MLP and Prospect-GNN, respectively, while referring to the corresponding instance that employs SAGE (Hamilton et al., 2017) as Prospect-SAGE. Offline GD-MLP frameworks, e.g., GLNN (Zhang et al., 2022), distill knowledge from pre-trained GNNs into MLPs for inference, bypassing expensive neighborhood aggregation. Similarly, PROSPECT enables high inference scalability utilizing Prospect-MLP. In the following sections, we elucidate the advantages of PROSPECT from both heuristic and theoretical perspectives. Our main contributions can be summarized as follows.

- According to our present knowledge, we propose the *first* online GD-MLP framework PROSPECT, which incorporates an adversarial robustness mechanism catering to both homophilous and heterophilous graphs, enables seamless integration with SAGE, and achieves inference scalability orders of magnitude higher than conventional GNNs.

- To the best of our knowledge, we are the *first* to investigate the graph structure adversarial robustness of GD-MLPs, revealing the vulnerability of offline GD-MLPs to poisoning attacks. In contrast, the proposed online GD-MLP framework PROSPECT is robust against both evasion and poisoning structure attacks.

- We prove in Theorem 1 that when the graph of an arbitrary heterophily level is *successfully* attacked, the MLP-to-GNN distillation in PROSPECT can correct the poisoned knowledge such that the GNN output features are pushed farther from the decision boundary versus without MLP correction.

- We analyze the convergence condition of Prospect-MLP with Banach fixed-point theorem in Theorem 2. Inspired by this analysis and the alternating iterative turbo decoding from information theory, we design the QACA learning rate scheduler for PROSPECT.

- Through the experiments on five homophilous and three heterophilous graphs, we demonstrate the effectiveness of QACA and the superior adversarial robustness, clean accuracy, and inference scalability of PROSPECT over baselines.

---

[3]Evasion attacks modify the graph for inference/testing while poisoning attacks change that for training

## 2 PRELIMINARIES AND NOTATIONS

Given an undirected and connected graph $\mathcal{G} = (\mathcal{V}, \mathcal{E})$ consisting of $N$ nodes $\mathcal{V} = \{1, \cdots, N\}$ and $M = |\mathcal{E}|$ edges, we denote the adjacency and degree matrices respectively by $\mathbf{A} \in \{0,1\}^{N \times N}$ and $\mathbf{D} = \mathrm{diag}\,(\mathbf{A} \cdot \mathbf{1})$ wherein $\mathbf{1}$ is an all-one column vector and $\mathrm{diag}\,(\mathbf{r})$ outputs a diagonal matrix taking vector $\mathbf{r}$ as the diagonal. Real-world graphs exhibit varying degrees of homophily, i.e., the tendency to link similar nodes. Various homophily measures exist (Zhu et al., 2020; Pei et al., 2020; Lim et al., 2021), with the edge-based homophily (Zhu et al., 2020) prevalent as the most widely adopted quantification. Given the node label vector $\mathbf{y} \in \mathbb{R}^N$, the edge-based homophily ratio (HR) is defined as the fraction of edges linking same-label nodes $h(\mathcal{G}, \mathbf{y}) = \frac{1}{|\mathcal{E}|} \sum_{(i,j) \in \mathcal{E}} y_i == y_j$.

Given the propagation matrix $\mathbf{P} = \mathbf{D}^{-1}\mathbf{A}$, the early GNN model GraphSAGE (Hamilton et al., 2017) widely used downstream can be formulated as (without sampling)

$$\mathbf{H}^{(l+1)} = \phi\left(\mathbf{H}^{(l)}\mathbf{W}_1^{(l)} + \mathbf{P}\mathbf{H}^{(l)}\mathbf{W}_2^{(l)}\right), \tag{1}$$

where $\phi\,(\cdot)$ is the activation function, $\mathbf{H}^{(l)}$ denotes the input features of the $l$-th layer, and $\mathbf{W}^{(l)}$ is the weight matrix. The node-wise formula for node $i$ aggregates within the neighborhood $\mathcal{N}(i)$

$$\mathbf{h}_i^{(l+1)} = \left(\mathbf{W}_1^{(l)}\right)^\top \mathbf{h}_i^{(l)} + \frac{1}{|\mathcal{N}(i)|} \sum_{j \in \mathcal{N}(i)} \left(\mathbf{W}_2^{(l)}\right)^\top \mathbf{h}_j^{(l)}. \tag{2}$$

An $L$-layer GNN is a function mapping the $d$-dimensional input node features $\mathbf{H}^{(0)} = \mathbf{X} \in \mathbb{R}^{N \times d}$ to the normalized logits $\mathbf{Z} = f_\theta(\mathcal{G}) = f_\theta(\mathbf{X}, \mathbf{A}) \in \mathbb{R}^{N \times C}$ over $C$ classes.

Supervised node classification minimizes $\mathcal{L}_{train}\,(f_\theta(\mathcal{G})) = \frac{1}{|\mathcal{V}_L|} \sum_{i \in \mathcal{V}_L} \ell_{CE}\,(y_i, [f_\theta(\mathbf{X}, \mathbf{A})]_i)$ on the training set $\mathcal{V}_L$, where $\ell_{CE}$ is the cross entropy loss, $[\mathbf{M}]_i$ extracts the $i$-th row of matrix $\mathbf{M}$, and $y_i$ is the ground-truth label of node $i$. Furthermore, we denote by $\mathcal{V}_{obs}$ the set of nodes observed during training and by $\mathcal{V}_{test}$ the test nodes. Poisoning structure adversarial attacks, e.g., MetaAttack (Zügner & Günnemann, 2019), tamper the graph to $\hat{\mathcal{G}} = \left(\hat{\mathbf{A}}, \mathbf{X}\right)$ before training, decreasing the accuracy on $\mathcal{V}_{test}$. Offline GD-MLPs, including GLNN (Zhang et al., 2022), LLP (Guo et al., 2023), and NOSMOG (Tian et al., 2023b), impart the knowledge from a pre-trained GNN to lightweight MLPs via knowledge distillation (KD) (Hinton et al., 2015). The loss function of GLNN is Eq. 3b.

## 3 PROPOSED FRAMEWORK: PROSPECT

### 3.1 OVERVIEW

**The architecture and optimization objective**  The PROSPECT architecture incorporates both GNN-to-MLP and MLP-to-GNN disiatlltion, as illustrated in Figure 3 (in Appendix B.1). The optimization objective of PROSPECT comprises GNN (Eq. 3a) and MLP (Eq. 3b) parts, and can be formulated as $\mathcal{L}_{pro} = \mathcal{L}_g + \mathcal{L}_m$

$$\mathcal{L}_g = \frac{1}{|\mathcal{V}_L|} \sum_{i \in \mathcal{V}_L} \ell_{CE}\,(y_i, [\mathbf{Z}_g]_i) + \frac{\alpha_1 t_1^2}{|\mathcal{V}_{obs}|} \sum_{i \in \mathcal{V}} \ell_{KLD}\left(\left[\mathbf{Z}_m^{(t_1)}\right]_i, \left[\mathbf{Z}_g^{(t_1)}\right]_i\right) \tag{3a}$$

$$\mathcal{L}_m = \frac{1}{|\mathcal{V}_L|} \sum_{i \in \mathcal{V}_L} \ell_{CE}\,(y_i, [\mathbf{Z}_m]_i) + \frac{\alpha_2 t_2^2}{|\mathcal{V}_{obs}|} \sum_{i \in \mathcal{V}} \ell_{KLD}\left(\left[\mathbf{Z}_g^{(t_2)}\right]_i, \left[\mathbf{Z}_m^{(t_2)}\right]_i\right), \tag{3b}$$

where $\ell_{KLD}$ is the Kullback-Leibler divergence (KLD), $\alpha_1$ and $\alpha_2$ are the weights of distillation losses, the subscripts of $\mathbf{Z}_g$ and $\mathbf{Z}_m$ denote the prediction matrices separately belonging to Prospect-GNN and Prospect-MLP, and the superscripts $t_1$ and $t_2$ of $\mathbf{Z}^{(t_1)}$ and $\mathbf{Z}^{(t_2)}$ are softmax temperatures.

**Adversarial robustness against structure attacks**  Prospect-GNN and Prospect-MLP usually achieve comparable acccuracy (see Section 6). Thus we prefer to deploy the later for fast inference. Since MLPs do not need graph structures, evasion structure attacks cannot infect them. That is, *PROSPECT can be completely immune to evasion structure attacks.* And we hence focus on poisoning attacks here. Regarding poisoning, GNN is infected by poisoned structures during training,

while MLP only absorbs clean node features. So MLP-to-GNN distillation can purify the incorrect knowledge of Prospect-GNN with the clean knowledge from Prospect-MLP (as proved in Theorem 1), imparting poisoning robustness to Prospect-GNN (and consequently Prospect-MLP through GNN-to-MLP distillation).

**Clean accuracy improvement via mutual distillation**   Offline GD-MLPs, e.g., GLNN (Zhang et al., 2022), based on GNN-to-MLP distillation, can improve student MLPs to match the performance of teacher GNNs. Therefore, it is reasonably expected that PROSPECT, incorporating this direction of distillation, should achieve at minimum the clean accuracy of GLNN. Moreover, recent research on non-graph models (Li & Jin, 2022) shows that the key for online distillation to outperform offline distillation is the reverse distillation from student to teacher, which narrows the knowledge gap between them and enables better transfer of knowledge from teacher to student. As PROSPECT adopts the reverse distillation from MLP to GNN, it is highly probable that Prospect-MLP will attain even superior performance over unidirectional GLNN.

**Adaptability to heterophilous graphs**   Regardless of homophilous or heterophilous graphs, the core goal and ultimate impact of any *effective* structural attack is a significant accuracy decrease on target nodes. In this case, the MLP unaffected by the poisoned structure is very likely to have a higher probability on the ground-truth class than the peer GNN, and PROSPECT can thus leverage MLP-to-GNN distillation to diminish or even eliminate the poisoned knowledge (in GNN weights) with the (relatively more correct) MLP knowledge, as proved in Theorem 1. Since such rectification depends solely on attack success irrespective of homophily levels, the adversarial robustness of PROSPECT can adapt to heterophilous graphs.

**Generalizability to early GNNs**   The heuristic analysis above and the theoretical analysis to be presented later can be extended to many neighborhood-aggregation-based simple GNNs that are extensively used in downstream tasks with a little effort. Specifically, we demonstrate the generalizability of PROSPECT to SAGE (Hamilton et al., 2017) in Theorem 1. And a similar theorem for GCN (Kipf & Welling, 2017) may be obtained by discarding the root projection and changing the propagation matrix in SAGE convolution (Eq. 1 or Eq. 2).

**High inference scalability**   Most GNNs must load the entire graph into (CPU or GPU) memory in advance for inference, even when only a few nodes are to predict. Since graph data usually take up more space than model parameters, Prospect-MLP, neglecting structures, has much lower spatial complexity than existing GNNs. Regarding time complexity, GNNs often exhibit prohibitive latency due to graph dependency. That is, adding several layers may require aggregating neighbors from too wider graph neighborhood. For instance, given the average degree $D$ and the feature dimension $d$, the inference on one node with an $L$-layer SAGE incurs an exponential complexity of $\mathcal{O}(D^L d^2 L)$. In contrast, an $L$-layer MLP only uses the node features and thus has a complexity of $\mathcal{O}(d^2 L)$.

### 3.2   THE BENEFITS OF MLP-TO-GNN DISTILLATION

The most significant distinction between PROSPECT and offline GD-MLPs lies in MLP-to-GNN distillation. So we first analyze the effect of MLP-to-GNN distillation on the graphs generated with the prevalent contextual stochastic block model (CSBM) (Deshpande et al., 2018; Ma et al., 2022) and our aCSBM (described in Appendix B.2), and then discuss how the resulting theorem relates to the adversarial robustness, clean accuracy, and heterophily adaptability of PROSPECT. The benefits of MLP-to-GNN distillation are shown by Theorem 1 with auxiliary Proposition 1 and Proposition 2, and the proofs can be found in Appendices D, C.1, and C.2, respectively. Furthermore, based on Theorem 1, we make some phenomenon predictions, which are highlighted as the claims in colored boxes in this section and empirically validated in Section 6.

**Proposition 1.** *Given a two-class $\{0, 1\}$ CSBM graph $\mathcal{G} \sim \mathrm{CSBM}\left(\boldsymbol{\mu}_0, \boldsymbol{\mu}_1, p, q\right)$ (Ma et al., 2022) or aCSBM graph $\mathcal{G} \sim \mathrm{aCSBM}\left(\boldsymbol{\mu}_0, \boldsymbol{\mu}_1, p, q\right)$, the node feature vector of node $i$ is sampled from a multi-variant Gaussian $\mathbf{x}_i \sim \mathcal{N}(\boldsymbol{\mu}_{y_i}, \sigma^2 \mathbf{I})$. Then the node feature vector $\mathbf{h}_i$ obtained via a SAGE layer (Eq. 2) follows the Gaussian distribution $\mathbf{h}_i \sim \mathcal{N}\left(\mathbb{E}_{y_i}\left[\mathbf{h}_i\right], \mathbb{D}_{y_i}\left[\mathbf{h}_i\right]\right)$, where*

$$\mathbb{E}_{y_i}\left[\mathbf{h}_i\right] = \mathbf{W}_1^\top \boldsymbol{\mu}_{y_i} + \mathbf{W}_2^\top \frac{p\boldsymbol{\mu}_{y_i} + q\boldsymbol{\mu}_{1-y_i}}{p+q}, \ \mathbb{D}_{y_i}\left[\mathbf{h}_i\right] = \sigma^2 \mathbf{W}_1^\top \mathbf{W}_1 + \frac{\sigma^2(p^2+q^2)}{(p+q)^2}\mathbf{W}_2^\top \mathbf{W}_2. \ (4)$$

**Proposition 2.** *Following the setting in Proposition 1, the decision boundary $\mathcal{P}_h$ of the optimal linear classifier on the processed features $\mathbf{h}_i$ is the hyperplane crossing the midpoint $\mathbf{m}_h$ and orthogonal to the line connecting two cluster centers, i.e., $\mathbb{E}_0[\mathbf{h}_i]$ and $\mathbb{E}_1[\mathbf{h}_i]$.*

$$\mathcal{P}_h = \left\{ \mathbf{h} \mid \mathbf{s}_h^\top \mathbf{h} - \mathbf{s}_h^\top \mathbf{m}_h \right\} \tag{5a}$$

$$\mathbf{m}_h = \frac{\left(\mathbf{W}_1^\top + \mathbf{W}_2^\top\right)(\boldsymbol{\mu}_0 + \boldsymbol{\mu}_1)}{2}, \quad \mathbf{s}_h = \frac{\left(\mathbf{W}_1^\top + \frac{p-q}{p+q}\mathbf{W}_2^\top\right)(\boldsymbol{\mu}_0 - \boldsymbol{\mu}_1)}{\left\|\left(\mathbf{W}_1^\top + \frac{p-q}{p+q}\mathbf{W}_2^\top\right)(\boldsymbol{\mu}_0 - \boldsymbol{\mu}_1)\right\|_2} \tag{5b}$$

**Theorem 1.** *Consider the binary classification task on a CSBM or aCSBM graph (in Proposition 1) using a one-layer SAGE paired with an MLP. After one forward-backward optimization step on any node $i$, the SAGE outputs with and without MLP-to-GNN distillation gradients are denoted as $\mathbf{h}_i^{kd}$ and $\mathbf{h}_i$, respectively. If the MLP has a higher prediction probability than the SAGE on the ground-truth class, then in expectation:*

- *the optimal linear classifier defined by the decision boundary $\mathcal{P}_h$ in Proposition 2 has a lower misclassification probability on $\mathbf{h}_i^{kd}$ than $\mathbf{h}_i$ no matter how heterophilous the graph is (or is changed to be by adversarial attacks);*
- *the misclassification probability gap between using $\mathbf{h}_i^{kd}$ and $\mathbf{h}_i$ gets minimized at the heterophilous demarcation point of 0.5 homophily ratio, and is maximized as the homophily ratio approaches to 0 or 1.*

*And the if condition is referred to as **Prospect Condition**.*

**Adversarial robustness against structure attacks**   Per the first conclusion of Theorem 1, MLP-to-GNN distillation can provide beneficial gradients shifting the GNN outputs away from the decision boundary, mitigating or even eliminating poisoning effects. While our theorem is, like other GNN theories (Baranwal et al., 2021; Ma et al., 2022; Chien et al., 2022a; Lei et al., 2022; Li et al., 2023; Gosch et al., 2023), only proven on stochastic graphs, we conjecture that

*Claim 1.* PROSPECT defends against poisoning structure attacks on various real-world graphs.

Since the MLP ignores the poisoned structure, it likely maintains a higher ground-truth probability than the corrupted GNN, meeting Prospect Condition. Note that Prospect Condition in Theorem 1 does not require a high MLP probability, only higher than the (expected) small value of attacked GNN. Since more drastic attacks can amplify this probability gap and hence increase the likelihood of satisfying Prospect Condition, we conjecture for PROSPECT that under the poisoning attacks

*Claim 2.* The robustness advantages over SAGE get greater with the attacks get more destructive.

**Clean accuracy improvement via mutual distillation**   Given their respective emphases on node attributes and topological structures, MLPs and GNNs may exhibit divergence in the correctly classified node sets $\mathcal{V}_{mlp}$ and $\mathcal{V}_{gnn}$. As implied by offline GD-MLPs (Zhang et al., 2022; Tian et al., 2023a) , GNN-to-MLP distillation can enhance MLPs on $\mathcal{V}_{gnn}$, whereas Theorem 1 indicates that MLP-to-GNN distillation can improve GNNs on $\mathcal{V}_{mlp}$. This suggests that the bidirectional (mutual) distillation in PROSPECT can confer complementary benefits. Hence we conjecture that

*Claim 3.* PROSPECT can boost the clean accuracy of the participating MLP and GNN.

The second conclusion of Theorem 1 shows that the minimal effect condition of MLP-to-GNN distillation is a homophily ratio of 0.5 and the maximum effect is obtained at extreme homophily ratio values approaching 0 or 1. Real-world graphs usually have various homophily levels (as shown in Table 4), and we conjecture for PROSPECT that

*Claim 4.* The clean accuracy improvement is significant on graphs with extreme homophily ratios.

**Adaptability to heterophilous graphs**   Theorem 1 shows that MLP-to-GNN distillation can enhance the participant GNN (and consequently the involved MLP via reverse distillation) when Prospect Condition is met. Since this condition is unaffected by heterophily levels, the performance gains can be attained on both homophilous and heterophilous graphs once the condition is satisfied. Hence, for PROSPECT we conjecture that

*Claim 5.* The adversarial robustness adapts to graphs with arbitrary homophily ratios.

*Claim 6.* The clean accuracy improvement adapts to graphs with arbitrary homophily ratios.

# 4 PROPOSED SCHEDULER: QACA

The optimization objective of PROSPECT comprises four parts (i.e., the terms in Eq. 3a and Eq. 3b), making the learning more challenging than regular MLPs and GNNs. To enable better optimization, we analyze the convergence condition of Prospect-MLP (in Theorem 2), and accordingly employ the cosine annealing (QA) (Loshchilov & Hutter, 2016) learning rate schedule. Furthermore, to prevent the potential knowledge conflicts between Prospect-GNN and Prospect-MLP, we adopt an alternating training strategy silencing each participant in turn. Collectively, these two insights lead to our quasi-alternating cosine annealing (QACA) learning rate scheduler for PROSPECT.

**Theorem 2.** *Given a Prospect-MLP trained with the loss function Eq. 3b and assume that $\exists u > 0$,*

$$\text{Tr}\left\{(w_1 - w_2)^\top [\nabla \mathcal{L}_m(w_1) - \nabla \mathcal{L}_m(w_2)]\right\} \geq u\|w_1 - w_2\|_F^2, \tag{6}$$

*one global or local optimal MLP weight $w^*$ of the last layer can be found by gradient descent if*

$$0 \leq \left(1 + \eta^2\beta^2 - 2\eta u\right) < 1 \tag{7a}$$

$$\beta = \frac{1}{|\mathcal{V}_L|}\sigma(\mathbf{H}^\top \mathbf{S}^\top \mathbf{S})\sigma(\mathbf{H}) + \frac{\alpha t_2}{N}\sigma^2(\mathbf{H}), \tag{7b}$$

*where $\eta$ is the learning rate, $\mathbf{H}$ is the input feature matrix of last MLP layer, $\sigma(\mathbf{M})$ is the spectral norm of matrix $\mathbf{M}$, and $\mathbf{S} \in \{0,1\}^{|\mathcal{V}_L|\times N}$ is the row selection matrix to extract the training node rows of a matrix $\mathbf{X} \in \mathbb{R}^{N\times d}$ into $\mathbf{SX} \in \mathbb{R}^{|\mathcal{V}_L|\times d}$.*

**Why cosine annealing (CA)?** Although Theorem 2 (proved in Appendix E) only considers the last MLP layer, similar quadratic inequalities on $\eta$ can be derived for all layers by extending the main proof tool (i.e., Proposition 5 in Appendix E.1). Still, this simplified theorem sufficiently motivates us. Theorem 2 delineates the feasible convergence regions for the learning rate. With a fixed learning rate schedule, excessively high values may fall outside these feasible regions, while exceedingly low values can drastically slow learning. Furthermore, the feasible regions usually dynamically shift during training as $\beta$ and $u$ change with the training process, meaning that a fixed learning rate risks deviating from the feasible regions over time. In contrast, the cosine annealing (Loshchilov & Hutter, 2016) scheduler can dynamically adjust the learning rate to increase the likelihood of remaining within the feasible regions, as elaborated in Appendix B.3.

**Why quasi-alternating (QA) learning?** The model structure difference between GNN and MLP may lead to knowledge conflicts in PROSPECT, and poisoned structures may further exacerbate such conflicts. With both learning rates high, the intense rapid knowledge exchange can confuse the participants of PROSPECT. By contrast, with only one side high, the unique knowledge from the side of low learning rate will remain roughly stable and thus accessible. Additionally, according to Theorem 2, alternating learning can also stabilize the training dynamics of Prospect-MLP. That is, when Prospect-GNN keeps silenced and near stable knowledge, $u$ in Eq. 6 is more likely to be less oscillatory since the gradients from GNN-to-MLP distillation are less fluctuated with stable GNN knowledge. Furthermore, drawing parallels to the alternating iterative turbo decoding (Berrou et al., 1993) in information theory, we hypothesize that an alternating knowledge exchange mechanism can aid PROSPECT in eliminating the errors induced by poisoning structure attacks.

**QACA learning rate scheduler** The above analysis results in our QACA scheduler, which can be formulated as

$$\eta_T = \begin{cases} \eta_{\min} + \frac{1}{2}(\eta_{\max} - \eta_{\min})\left(1 + \cos\left(\frac{2T_{\text{cur}}}{T_0}\pi\right)\right) & T_{\text{cur}} < T_0/2 \\ \eta_{\min} & T_{\text{cur}} \geq T_0/2 \end{cases}, \tag{8}$$

where $\eta_{\min}$ and $\eta_{\max}$ determine the range of learning rates, $T_{\text{cur}} = (T + B) \mod T_0$ accounts for how many epochs have been performed since the last restart, $B$ is the offset before starting scheduling, and $T_0$ epochs constitute a minimal schedule period. QACA performs annealing in the first $T_0/2$ epochs of one minimal period while silencing in the rest epochs of it. In PROSPECT, we set the offsets $B = T_0/2$ for GNN and $B = 0$ for MLP to train them alternatingly. A small $\eta_{\min}$ retains some model activity vs. complete muting, making QACA "quasi"-alternating. Figure 5 (in Appendix B.3) illustrates a QACA example. And the ablation experiments in Section 6.3 demonstrate the effectiveness of both cosine annealing and quasi-alternating in QACA.

Table 1: The results of adversarial robustness on four datasets. The mean and std over five splits are reported. The top two performing models are highlighted in bold, with the best further underlined.

| | Polblogs (HR=0.906) | | Citeseer (HR=0.736) | | UAI (HR=0.364) | | Texas (HR=0.061) | |
|---|---|---|---|---|---|---|---|---|
| | 5% | 15% | 5% | 15% | 5% | 15% | 5% | 15% |
| MLP | 52.21±0.61 | | 66.01±1.37 | | 61.74±2.11 | | 65.71±4.42 | |
| MLPw2 | 52.09±0.24 | | 66.82±1.83 | | 64.13±1.40 | | 67.21±3.02 | |
| MLPw4 | 51.72±0.87 | | 66.43±1.73 | | 62.71±1.91 | | 68.98±3.32 | |
| GCN | 77.18±1.76 | 67.53±0.99 | 72.03±1.23 | 64.74±2.70 | 56.72±4.68 | 54.22±3.17 | 49.25±5.43 | 49.39±2.29 |
| SGC | 77.71±1.79 | 66.95±1.36 | 71.94±1.31 | 64.51±2.44 | 58.78±3.34 | 56.52±2.64 | 53.88±2.23 | 55.24±2.12 |
| SAGE | 90.39±0.66 | 77.34±3.74 | 72.68±1.25 | 70.40±1.05 | 60.02±3.21 | 60.18±2.65 | 62.99±3.39 | 64.35±2.99 |
| RGCN | 75.42±1.29 | 66.18±0.64 | 71.71±2.04 | 64.02±1.90 | 49.89±2.85 | 48.40±2.74 | 52.93±1.89 | 49.52±8.10 |
| SVD | 92.43±0.70 | 73.44±1.77 | 69.82±0.86 | 65.15±2.01 | 48.65±1.14 | 44.87±1.18 | 49.66±4.02 | 48.57±5.66 |
| Jaccard | 50.88±1.69 | 50.88±1.69 | 72.18±1.81 | 66.96±2.71 | 54.08±4.18 | 50.64±2.69 | 49.25±5.43 | 49.39±2.29 |
| Guard | 51.58±0.57 | 51.58±0.57 | 69.79±1.24 | 67.35±0.62 | 20.28±10.99 | 20.36±8.27 | 48.03±12.96 | 47.76±11.40 |
| ProGNN | 85.97±5.16 | 72.78±3.43 | 71.60±1.84 | 65.12±2.38 | 49.22±5.22 | 38.43±11.55 | 47.89±10.06 | 45.31±14.47 |
| STABLE | 92.80±2.38 | 88.55±0.38 | 74.33±1.08 | 73.32±1.14 | 51.78±2.08 | 47.63±2.26 | 52.27±2.82 | 50.52±3.24 |
| EvenNet | 87.04±1.45 | 68.06±1.50 | 74.08±1.02 | 70.95±1.71 | 67.8±2.029 | 66.91±2.18 | 62.45±2.70 | 63.27±2.85 |
| GLNN | 91.62±1.35 | 77.46±3.73 | 74.25±1.20 | 71.92±1.38 | 62.46±2.91 | 62.02±2.00 | 66.40±2.57 | 66.53±5.45 |
| GLNNw2 | 91.55±1.11 | 77.14±3.98 | 74.44±1.51 | 72.13±1.36 | 62.89±3.10 | 63.15±1.22 | 66.67±2.93 | 66.12±5.49 |
| GLNNw4 | 91.19±1.41 | 77.12±3.58 | 74.01±1.32 | 71.94±1.38 | 62.62±2.39 | 62.75±1.99 | 67.21±2.70 | 66.40±4.92 |
| Prospect-SAGE | **93.95±1.34** | **92.27±2.26** | **75.01±0.75** | **74.81±0.41** | **69.86±0.58** | **69.52±0.46** | 68.84±5.65 | 71.02±2.30 |
| Prospect-MLP | **93.99±0.76** | **93.95±0.34** | **75.31±1.18** | **74.79±0.64** | **68.31±0.59** | **69.10±0.45** | **72.11±2.06** | **73.20±1.53** |

## 5 RELATED WORK

In Appendix F, we present a comparative discussion of PROSPECT versus adversarial defense methods and offline GD-MLPs, highlighting the differences between PROSPECT and them.

## 6 EXPERIMENTS

In this section, we empirically study the proposed across four aspects: adversarial robustness, clean accuracy, QACA effectiveness, and inference scalability. We compare our PROSPECT with the following methods. **1)** Method only using node features: MLP. **2)** Early simple GNNs: GCN (Kipf & Welling, 2017), SAGE (Hamilton et al., 2017), and SGC (Wu et al., 2019a). **3)** Purification-based adversarial defense methods: SVD (Entezari et al., 2020), Jaccard (Wu et al., 2019b), RGCN (Zhu et al., 2019), Guard (Zhang & Zitnik, 2020), ProGNN (Jin et al., 2020), and STABLE (Li et al., 2022). **4)** heterophily-aware GNNs: EvenNet (Lei et al., 2022). **5)** Offline GD-MLPs: GLNN (Zhang et al., 2022). The details of the experiment settings are in Appendix G.1.

### 6.1 ADVERSARIAL ROBUSTNESS

In Section 3.2, Theorem 1 implies the adversarial robustness of PROSPECT, and we now conduct robustness experiments to validate it. Table 1 presents the results on two homophilous and two hetherphilous graphs at 5% and 15% attack budgets. The full results on eight graphs with 5%, 10%, 15%, and 20% budgets are in Appendix G.2.1. Since MLPs ignore graph structures, each MLP row has an identical performance across budgets. So we only report for a budget of 5%. As shown by Tables 1, 5, 6, 7, and 8, Prospect-MLP and Prospect-SAGE rank first or second on nearly all attacked real-world graphs, supporting Claim 1. Further, the improvement of PROSPECT over standalone SAGE grows with the attack budget, e.g. from ~3.5% at Meta-5 to ~14.9% at Meta-15 on Polblogs (Table 1), confirming Claim 2. On the extremely heterophilous Texas (Table 1 or 7), the defenders with homophilous GNNs fail with <53% accuracy. In contrast, the heterophily-adapted EvenNet outperforms them by >9% across all budgets, and our Prospect-MLP exceeds EvenNet by ~10%. In fact, significant advantages can be observed on almost all homophilous and heterophilous graphs, validating the adaptability of PROSPECT robustness to heterophily declared in Claim 5. In addition, we have three observations about SAGE, GLNN, and PROSPECT discussed in Appendix G.2.2.

### 6.2 CLEAN ACCURACY

In Section 3.2, Theorem 1 implies the clean accuracy improvment of PROSPECT. To verify this, we conduct comparison experiments on eight clean graphs, and the results are shown in Table 2. The

Table 2: Performance comparison on clean graphs. The mean and std over five splits are reported. The top two performing models are highlighted in bold, with the best further underlined.

| | homophilous (HR>0.5) | | | | | heterophilous (HR<0.5) | | |
|---|---|---|---|---|---|---|---|---|
| | Cora | Citeseer | Polblogs | ACM | CoraML | Texas | Chameleon | UAI |
| MLP | 65.69±1.43 | 66.02±1.37 | 52.21±0.61 | 87.44±0.28 | 71.31±0.65 | 65.71±4.42 | 42.65±0.86 | 61.74±2.11 |
| MLPw2 | 66.58±0.94 | 66.83±1.83 | 52.09±0.24 | 87.36±0.25 | 70.69±1.78 | 67.21±3.02 | 42.91±0.48 | 64.13±1.40 |
| MLPw4 | 67.02±1.12 | 66.43±1.73 | 51.72±0.87 | 87.39±0.69 | 71.60±0.94 | 68.98±3.32 | 41.08±2.05 | 62.71±1.91 |
| GCN | 84.20±0.92 | 73.40±2.01 | 94.79±1.20 | 89.65±0.75 | 85.94±0.68 | 51.29±6.46 | **56.69±2.68** | 63.51±1.29 |
| SGC | 82.58±0.75 | 73.57±1.49 | 94.68±0.90 | 90.03±0.79 | 84.66±0.64 | 53.06±1.86 | 50.91±2.30 | 62.28±2.32 |
| SAGE | 83.56±0.86 | 74.53±1.01 | 94.40±0.77 | 90.31±0.90 | 84.51±1.09 | 64.76±1.76 | 51.66±2.21 | 60.56±3.83 |
| RGCN | 83.85±0.63 | 72.94±1.68 | 94.87±0.81 | 89.40±2.08 | 86.19±0.57 | 51.57±2.17 | 55.74±1.46 | 54.80±1.68 |
| SVD | 77.72±0.37 | 69.65±1.53 | 93.58±0.85 | 86.41±1.49 | 81.09±0.61 | 51.02±2.85 | 47.87±2.25 | 50.58±1.82 |
| Jaccard | 82.95±0.68 | 73.50±1.72 | 50.88±1.69 | 89.65±0.75 | 84.81±0.47 | 51.29±6.46 | 45.32±1.27 | 61.09±1.05 |
| Guard | 78.33±1.15 | 70.14±2.30 | 51.58±0.57 | 89.23±1.14 | 77.06±1.07 | 48.44±8.61 | 40.89±2.27 | 32.84±20.31 |
| ProGNN | 83.84±0.77 | 73.72±0.99 | 94.83±0.51 | 90.17±0.76 | 85.64±0.73 | 51.84±3.31 | 53.63±1.39 | 57.65±1.01 |
| STABLE | 83.09±0.58 | 74.44±0.56 | 94.68±0.45 | 85.40±0.83 | 83.62±0.46 | 50.27±4.70 | 46.66±1.57 | 56.47±0.48 |
| EvenNet | **84.89±0.35** | 74.46±0.80 | **95.24±0.55** | 90.54±0.55 | **86.48±0.31** | 67.21±1.22 | 51.73±1.22 | **70.07±1.16** |
| GLNN | 83.17±0.68 | 75.14±0.84 | 94.15±0.63 | 91.90±0.45 | 84.87±0.86 | 67.48±3.38 | 48.36±2.19 | 62.54±3.34 |
| GLNNw2 | 83.47±0.77 | 75.33±0.86 | 94.42±0.79 | 92.05±0.66 | 84.69±0.86 | 68.30±4.55 | 48.55±1.80 | 63.06±3.25 |
| GLNNw4 | 83.23±0.79 | **75.60±0.52** | 94.58±0.82 | 91.89±0.51 | 84.99±1.00 | 68.44±4.60 | 49.36±2.07 | 62.94±2.79 |
| Prospect-SAGE | **84.94±0.51** | 75.20±0.70 | 95.22±0.24 | **93.15±0.86** | 85.93±0.91 | **72.79±3.22** | **55.88±1.12** | 69.90±0.92 |
| Prospect-MLP | 84.50±0.58 | **75.81±0.68** | **95.32±0.41** | **93.22±0.71** | **86.54±0.75** | **73.06±1.64** | 53.43±1.45 | 68.97±0.66 |

results indicate that Prospect-SAGE and Prospect-MLP achieve higher accuracy than standalone SAGE and MLP, respectively. This supports Claim 3. And such improvements are witnessed across all eight graphs with various homophily ratios, confirming Claim 6. On extremely heterophilous Texas (see Table 2), PROSPECT exceeds SAGE by about 9 percent, EvenNet by 6 percent, and GLNN by 5 percent, while the improvement on other datasets such as CoraML with not extreme HRs is less significant. This matches with Claim 4. Besides, we get some findings in Appendix G.2.2 by jointly checking the adversarial robustness and clean accuracy results.

## 6.3 QACA EFFECTIVENESS

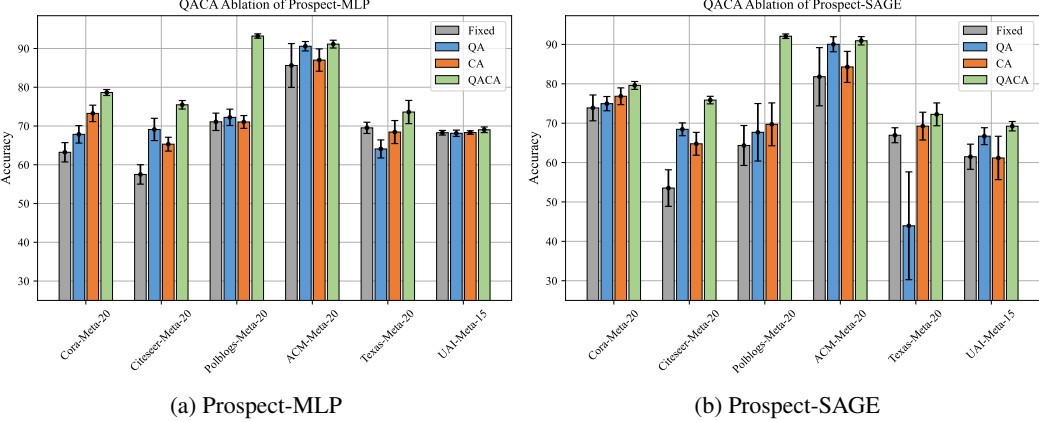

(a) Prospect-MLP          (b) Prospect-SAGE

Figure 1: Ablation study of QACA learning rate scheduling. Fixed indicates no learning rate change, QA enables quasi-alternating learning, CA utilizes the cosine annealing with warm restart (Loshchilov & Hutter, 2016), and QACA combines QA and CA.

To validate the effectiveness of cosine annealing (CA) and quasi-alternating (QA) learning in QACA designed in Section 4, we conduct ablation experiments on multiple attacked datasets with different learning rate schedulers, including fixed scheduler (Fixed), cosine annealing scheduler (CA), and quasi-alternating scheduler (QA). The CA scheduler is just adopted from Loshchilov & Hutter (2016), and formulated as $\eta_T = \eta_{\min} + \frac{1}{2}\left(\eta_{\max} - \eta_{\min}\right)\left(1 + \cos\left(\frac{T_{\text{cur}}}{T_0}\pi\right)\right)$, which differs with our QACA (Eq. 8) only in that there is no silence time during one minimal period $T_0$. To solely

check the QA impact, we construct the quasi-alternating scheduler that replaces the formula of first half period in Eq. 8 with $\eta_T = \eta_{\max}$. For the experiment on each dataset, the initial learning rates of Fxied, CA, QA, and QACA schedulers are all set as $\eta_{\max}$, and the latter three schedulers share the same $T_0$ and $\eta_{\min}$. The results on Cora/Citeseer/Polblogs/ACM/Texas/UAI-Meta-15 are presented in Figure 1, which tells that within the PROSPECT framework, using either QA or CA alone leads to better performance than not adjusting the learning rate on almost all graphs. Although on Texas-Meta-20, QA is inferior to Fixed, QACA combing QA and CA still significantly outperforms all other schedulers. These results demonstrate that cosine annealing and alternating learning in QACA scheduler are usually beneficial for PROSPECT.

## 6.4 INFERENCE SCALABILITY

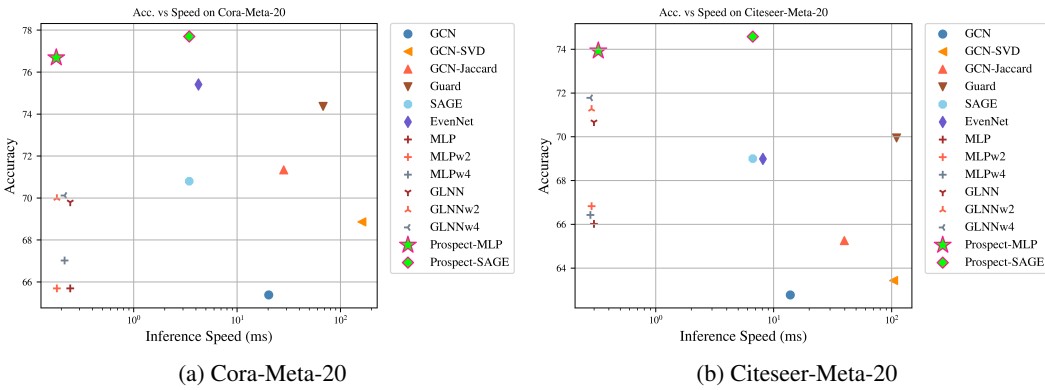

|   (a) Cora-Meta-20   |   (b) Citeseer-Meta-20   |

Figure 2: Acc. vs. inference speed in the production setting. The x-axis is logarithmically scaled.

Following the realistic semi-inductive setting in Appendix G.2.3, we measure the inference latency on 10 inductive test nodes and the overall accuracy on both inductive and transductive test nodes on an Ubuntu20.04 server equipped with RTX4090 (introduced in Appendix G.1). Experiments on Cora/Citeseer-Meta-20 (Figure 2) and Chameleon/Texas/Polblogs/CoraML-Meta-20 (Figures 7 and 8) show that Prospect-MLP achieves competitive accuracy compared to Prospect-SAGE, substantially higher than other methods, while matching the inference speed of MLPs. Notably, Prospect-MLP infers within 1ms on all graphs, making it promising for high-throughput industrial tasks.

## 7 CONCLUSIONS

This study set out to provide a "simple" yet efficient solution PROSPECT, to address key limitations of existing GNN adversarial defense methods: **1)** inadequate adaptability to heterophily, **2)** absent generalizability to early GNNs such as SAGE, and **3)** low inference scalability. PROSPECT pioneers the online GD-MLP framework via the novel mutual distillation between MLP and GNN. It can inference as efficiently as MLPs and seamlessly fit into early SAGE. We analyze the benefits of MLP-to-GNN distillation in Theorem 1, which indicates that Prospect-MLP can correct the wrong knowledge of Prospect-GNN irrespective of homophily ratios, endowing the adversarial robustness and clean accuracy improvements that adapt to heterophily. Furthermore, Theorem 2 analyzes the convergence condition of Prospect-MLP, which constitutes, with inspiration from the alternating iterative turbo decoding, our QACA scheduler. Experiments on homophilous and heterophilous graphs demonstrate the superior adversarial robustness and clean accuracy of PROSPECT over previous defenders and offline GD-MLPs, the inference scalability of PROSPECT significantly higher than existing defense methods, and the effectiveness of QACA scheduler.

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

# A    LIST OF SYMBOLS

Part of used symbols are summarized here.

Table 3: List of part of used symbols

| Symbol | Meaning |
|---|---|
| $\mathcal{G} = (\mathcal{V}, \mathcal{E})$ | The clean graph |
| $\mathcal{E}$ | The set of graph edges |
| $\mathcal{V}$ | The set of graph nodes |
| $\mathcal{V}_L / \mathcal{V}_{val} / \mathcal{V}_{test}$ | The set of training/validation/test nodes |
| $\mathcal{V}_{obs}$ | The set of graph nodes observed during training |
| $|\mathcal{V}|$ | The capacity of set $\mathcal{V}$ |
| $C$ | The number of node classes |
| $\mathbf{A}$ | The graph adjacency matrix |
| $\mathbf{I}$ | The identity matrix with an appropriate shape |
| $\mathbf{D} = \mathrm{diag}\left(\mathbf{A} \cdot \mathbf{1}\right)$ | The graph degree matrix |
| $y_i == y_j$ | return 1 if $y_i$ equals to $y_j$ otherwise 0 |
| $\hat{\mathcal{G}} = \left(\hat{\mathbf{A}}, \hat{\mathbf{X}}\right)$ | The attacked (perturbed) graph |
| $\mathrm{Tr}\{\mathbf{M}\}$ | The trace of matrix $\mathbf{M}$ |
| $\sigma(\mathbf{M})$ | The spectral norm of matrix $\mathbf{M}$ |
| $\phi'(\mathbf{X}) \in \mathbb{R}^{m \times n}$ | The element-wise derivative of $\phi(\mathbf{X})$ w.r.t. $\mathbf{X} \in \mathbb{R}^{m \times n}$ |
| $\mathbf{S} \in \{0, 1\}^{|\mathcal{V}_L| \times N}$ | The row selection matrix to pick out the training node rows |
| $\delta\mathbf{W} = \partial\mathcal{L} / \partial\mathbf{W}$ | The gradients of the scalar loss $\mathcal{L}$ w.r.t. a matrix $\mathbf{W}$ |
| $\|\mathbf{M}\|_2$ | The $L_2$ operator norm of matrix $\mathbf{M}$ |
| $\mathbf{L} = \mathbf{I} - \mathbf{D}^{-1/2}\mathbf{A}\mathbf{D}^{-1/2}$ | The symmetric Laplacian matrix |
| $\mathbf{P} \in \mathbb{R}^{N \times N}$ | The propagation matrix of GNN |
| $\mathbf{y} \in \{1, \cdots, C\}^N$ | The ground-truth label vector of all nodes |
| $y_i \in \{1, \cdots, C\}$ | The ground-truth label of node $i$ |
| $\mathbf{H}^{(0)} = \mathbf{X} \in \mathbb{R}^{N \times d}$ | The initial feature matrix |
| $\mathbf{H}^{(l)}$ | The input feature matrix of the $l$-th GNN layer |
| $\mathbf{Z} \in \mathbb{R}^{N \times C}$ | The normalized logits of $N$ graph nodes over $C$ classes |
| $\mathbf{h}_i$ | The input column feature vector of node $i$ |
| $\mathbf{Z}_m^{(t_1)}$ | The MLP logits normalized by $t_1$-softmax function |
| $[\mathbf{M}]_i$ | Pick out the $i$-th row of the matrix $\mathbf{M}$ as a row vector |
| $\boldsymbol{\mu}_0 / \boldsymbol{\mu}_1$ | The initial class centers of one two-class CSBM graph |
| $\mathbf{s}_h = \frac{\left(\mathbf{W}_1^\top + \frac{p-q}{p+q}\mathbf{W}_2^\top\right)(\boldsymbol{\mu}_0 - \boldsymbol{\mu}_1)}{\|\left(\mathbf{W}_1^\top + \frac{p-q}{p+q}\mathbf{W}_2^\top\right)(\boldsymbol{\mu}_0 - \boldsymbol{\mu}_1)\|_2}$ | The line connecting the two class centers of the SAGE outputs |
| $\mathbf{m}_h = \frac{\left(\mathbf{W}_1^\top + \mathbf{W}_2^\top\right)(\boldsymbol{\mu}_0 + \boldsymbol{\mu}_1)}{2}$ | The midpoint between two class centers |
| $\mathcal{P}_h = \left\{\mathbf{h} \mid \mathbf{s}_h^\top \mathbf{h} - \mathbf{s}_h^\top \mathbf{m}_h\right\},$ | The decision boundary for the GNN outputs on CSBM graphs |
| $\mathbf{A} \circ \mathbf{B}$ | The Hadamard/element-wise product between two matrices |
| $f \circ g$ | The composition of functions $f$ and $g$ |
| $\|f\|_{Lip}$ | The Lipschitz constant of scalar-valued function $f$ |
| $\nabla f(x)$ | The gradients of the scalar function $f$ w.r.t. $x$ |
| $\|\mathbf{M}\|_F$ | The Frobenius norm of matrix $\mathbf{M}$ |

# B MORE DETAILS OF THE PROPOSED

## B.1 THE PROSPECT ARCHITECTURE

To circumvent the expensive neighborhood fetching and aggregation operations during GNN inference, offline GD-MLP frameworks distill the knowledge of cumbersome pre-trained GNNs into MLPs by matching logits (Zhang et al., 2022) or/and representational similarity (Tian et al., 2023b). Logit matching minimizes the KLD between MLP and GNN logits, as formulated in Eq. 3b. While such unidirectional offline distillation enables MLPs to match the performance of GNN teachers, this approach remains constrained by the limitations of teacher models. Specifically, the teacher GNNs are usually limited in clean accuracy and vulnerable to poisoning structure attacks. In contrast, our proposed PROSPECT surpasses these performance limitations through complementary mutual regularization enabled by the bidirectional distillation between a GNN and an MLP. As depicted in Figure 3, Prospect-MLP leveraging only pristine node attributes can correct the Prospect-GNN infected by poisoned structures as suggested by Theorem 1 and the robustness evaluation (in Section 6.1). Such correction averts the deterioration of participants after poisoning GNN, which is a capability that offline GD-MLPs lack. Moreover, given the respective prioritization of node features versus graph structures by MLPs and GNNs, the mutual distillation can integrate both information types on unperturbed clean graphs, thereby benefiting the participants in clean accuracy, as demonstrated by the clean accuracy experiments (in Section 6.2).

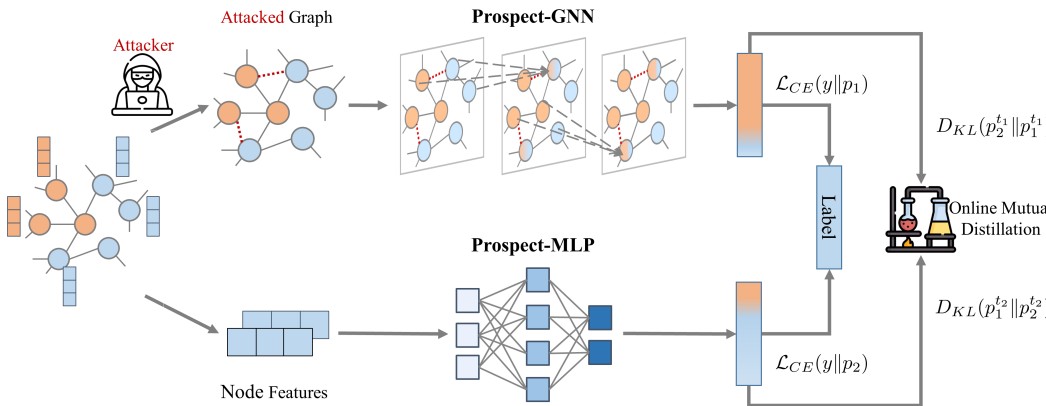

Figure 3: The PROSPECT framework. During the online training, a GNN and an MLP collaboratively extract information from the graph structure and node features via empirical risk minimization and mutual distillation. After training, either Prospect-GNN or Prospect-MLP can be deployed.

## B.2 THE CSBM AND aCSBM MODELS

The contextual stochastic block model (CSBM) (Deshpande et al., 2018) widely adopted for GNN analysis (Baranwal et al., 2021; Chen et al., 2021; Ma et al., 2022; Chien et al., 2022a;b; Gosch et al., 2023; Li et al., 2023), can flexibly generate random graphs with community structures and node features. And we adopt the two-class CSBM from Ma et al. (2022). Let $y_i \in \{0, 1\}$ be the label for any node $i$. The edge generation probability between intra-class nodes is denoted by $0 \le p \le 1$ and that of inter-class ones by $0 \le q \le 1$. The initial features of node $i$ is sampled from a Gaussian distribution $\mathbf{x}_i \sim \mathcal{N}(\boldsymbol{\mu}(y_i), \sigma^2 \mathbf{I})$ where $\boldsymbol{\mu}(y_i) = \boldsymbol{\mu}_0$ if $y_i = 0$ otherwise $\boldsymbol{\mu}_1$. Actually $\boldsymbol{\mu}_0$ and $\boldsymbol{\mu}_1$ ($\boldsymbol{\mu}_0 \ne \boldsymbol{\mu}_1$) are two cluster centers of raw node features. The values of $p$ and $q$ determine the density and homophily ratio of the generated graph $\mathcal{G} \sim \text{CSBM}(\boldsymbol{\mu}_0, \boldsymbol{\mu}_1, p, q), p^2 + q^2 \ne 0, p, q \in [0, 1]$. Though prevalent, CSBM tends to produce dense graphs when $p$ or $q$ is not small enough (Ma et al., 2022), because the edge generation is performed between all pairs of nodes. To extend our analysis to more common sparse graphs, we also consider an adapted setting where $p$ and $q$ denote intra-/inter-class neighbor ratios instead of generation probabilities. We denote this adapted model as $\mathcal{G} \sim \text{aCSBM}(\boldsymbol{\mu}_0, \boldsymbol{\mu}_1, p, q), p + q = 1, p, q \in [0, 1]$, which decouples the neighborhood sizes from $p$ and $q$ to enable sparsity even when $p$ or $q$ is large.

### B.3   COSINE ANNEALING INSPIRED BY THEOREM 2

Let $g(\eta) = \eta^2\beta^2 - 2\eta u$, then the convergence inequality (i.e., Eq. 7a) becomes $-1 \le g(\eta) < 0$. The zero points of $g(\eta)$ are 0 and $\frac{2u}{\beta^2}$, and the points $\left(\frac{u-\sqrt{u^2-\beta^2}}{\beta^2}, -1\right)$ and $\left(\frac{u+\sqrt{u^2-\beta^2}}{\beta^2}, -1\right)$ are on $g(\eta)$ if $u > \beta$. Figure 4 illustrates two cases of $g(\eta)$, with a gap between the feasible regions in the left case. For a fixed learning rate $\eta_0$, the possibilities in the left case include:

- $\eta_0 > \frac{2u}{\beta^2}$: it does not meet the convergent condition

- $\frac{u-\sqrt{u^2-\beta^2}}{\beta^2} < \eta_0 < \frac{u+\sqrt{u^2-\beta^2}}{\beta^2}$: it does not satisfies the convergent condition

- $\frac{u+\sqrt{u^2-\beta^2}}{\beta^2} \le \eta_0 \le \frac{2u}{\beta^2}$ or $0 < \eta_0 \le \frac{u-\sqrt{u^2-\beta^2}}{\beta^2}$ : $\eta_0$ may fall outside these two regions as the training proceeds since $u$ and $\beta$ generally vary throughout the training process, thus changing the feasible regions.

For the right case in Figure 4, similar issues persist, just with different feasible regions. To address these issues, a simple solution is to chose a very small $\eta_0$ close to origin such that $\eta_0$ always lies in the feasible region even if $u$ and $\beta$ change. However, this may yield intolerably slow training.

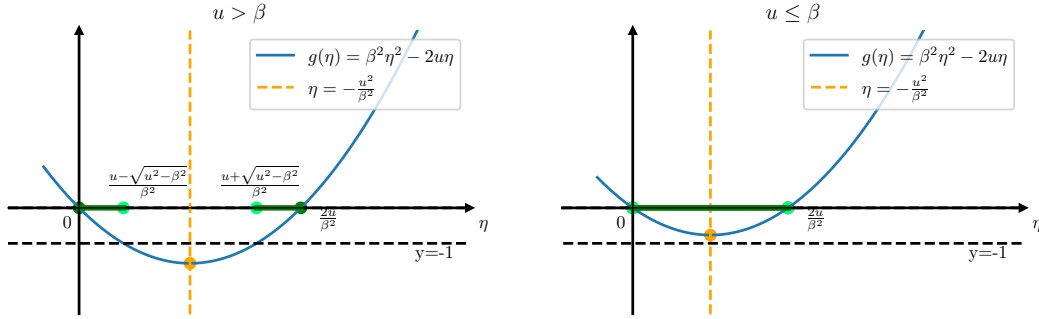

Figure 4: Plot of $g(\eta) = \eta^2\beta^2 - 2\eta u$. The orange lines are the symmetry axes. The green segments on the horizontal axis are the feasible regions of the convergence inequality in Eq. 7a.

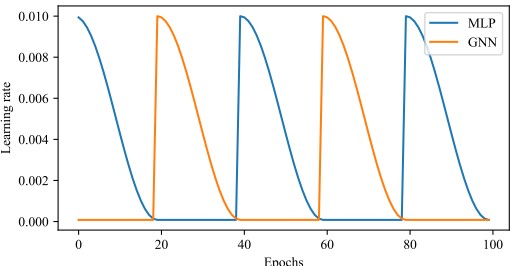

Figure 5: A QACA example for PROSPECT over 100 epochs with $T_0 = 20, \eta_{\min} = 0.00008, \eta_{\max} = 0.01$. We set $B = 0$ for Prospect-MLP and $B = T_0/2$ for Prospect-GNN.

To resolve this issue, we propose to use annealing during each schedule period. With an appropriately large initial learning rate $\eta_0$ and a cold lower bound $\eta_{\min}$ near origin, it enables fast training in early epochs. Then the learning rate adapts to stay in the feasible regions for more epochs than a fixed $\eta_0$ as it can walk across or within these regions even if these regions change with $u$ and $\beta$ during training. Finally, $\eta$ decays to $(0, \eta_{\min}], \eta_{\min} \ll \left(u - \sqrt{u^2-\beta^2}\right)/\beta^2$, an interval insensitive to the change of $u$ and $\beta$. Given the widespread verification of fast convergence and improved accuracy enabled by the cosine annealing with warm restart (Loshchilov & Hutter, 2016), we base the annealing component of our scheduler on this schedule.

# C  PROOFS OF SOME PROPOSITIONS

## C.1  PROOF OF PROPOSITION 1

*Proof.* For any node $i$, we denote the neighbor set by $\mathcal{N}_i$ and the neighborhood size by $N_i$. Eq. 2 shows that $\mathbf{h}_i$ is a linear combination of affine mapped $\mathbf{x}_i$ and $\mathbf{x}_j, j \in \mathcal{N}_i$. Since all node features are sampled from Gaussians, $\mathbf{h}_i$ still follows a Gaussian. As there are $\frac{pN_i}{p+q}$ intra-class neighbors and $\frac{qN_i}{p+q}$ inter-class neighbors, we readily get that

$$\mathbb{E}\left\{\frac{1}{N_i}\sum_{j\in\mathcal{N}(i)}\mathbf{x}_j\right\} = \frac{1}{N_i}\left\{\mathbb{E}\left[\sum_{j\in\mathcal{N}(i)\cap\{j,y_j=y_i\}}\mathbf{x}_j\right] + \mathbb{E}\left[\sum_{j\in\mathcal{N}(i)\cap\{j,y_j\neq y_i\}}\mathbf{x}_j\right]\right\} \tag{9a}$$

$$= \frac{1}{N_i}\left(\frac{pN_i}{p+q}\boldsymbol{\mu}_{y_i} + \frac{qN_i}{p+q}\boldsymbol{\mu}_{1-y_i}\right) = \frac{p\boldsymbol{\mu}_{y_i} + q\boldsymbol{\mu}_{1-y_i}}{p+q} \tag{9b}$$

$$\mathbb{D}\left\{\frac{1}{N_i}\sum_{j\in\mathcal{N}(i)}\mathbf{x}_j\right\} = \frac{1}{N_i^2}\left\{\mathbb{D}\left[\sum_{j\in\mathcal{N}(i)\cap\{j,y_j=y_i\}}\mathbf{x}_j\right] + \mathbb{D}\left[\sum_{j\in\mathcal{N}(i)\cap\{j,y_j\neq y_i\}}\mathbf{x}_j\right]\right\} \tag{9c}$$

$$= \frac{1}{N_i^2}\left[\frac{(pN_i)^2}{(p+q)^2}\sigma^2\mathbf{I} + \frac{(qN_i)^2}{(p+q)^2}\sigma^2\mathbf{I}\right] = \frac{p^2+q^2}{(p+q)^2}\sigma^2\mathbf{I} \tag{9d}$$

Thus we have

$$\frac{1}{N_i}\sum_{j\in\mathcal{N}(i)}\mathbf{x}_j \sim \mathcal{N}\left(\frac{p\boldsymbol{\mu}_{y_i}+q\boldsymbol{\mu}_{1-y_i}}{p+q}, \frac{p^2+q^2}{(p+q)^2}\sigma^2\mathbf{I}\right) \tag{10a}$$

$$\mathbf{W}_2^\top\frac{1}{N_i}\sum_{j\in\mathcal{N}(i)}\mathbf{x}_j \sim \mathcal{N}\left(\mathbf{W}_2^\top\frac{p\boldsymbol{\mu}_{y_i}+q\boldsymbol{\mu}_{1-y_i}}{p+q}, \mathbf{W}_2^\top\frac{p^2+q^2}{(p+q)^2}\sigma^2\mathbf{W}_2\right) \tag{10b}$$

Owing to that $\mathbf{W}_1^\top\mathbf{x}_i \sim \mathcal{N}\left(\mathbf{W}_1^\top\boldsymbol{\mu}_{y_i}, \sigma^2\mathbf{W}_1^\top\mathbf{W}_1\right)$, $\mathbf{h}_i = \mathbf{W}_1^\top\mathbf{x}_i + \frac{1}{N_i}\sum_{j\in\mathcal{N}(i)}\mathbf{W}_2^\top\mathbf{x}_j$ is the sum of two Gaussian distributions, amounting to the results in Proposition 1. □

## C.2  PROOF OF PROPOSITION 2

*Proof.* Proposition 1 suggests that the processed feature distributions of class 0 and 1 are two Gaussians separately centered at $\mathbb{E}_0\left[\mathbf{h}_i\right]$ and $\mathbb{E}_1\left[\mathbf{h}_i\right]$ with the same distribution 'radius', i.e., variance. Hence the optimal decision boundary of GNN outputs is naturally the hyperplane defined in Eq. 5a, analogous to Ma et al. (2022). To get the optimal decision boundary, we directly compute the midpoint between $(\mathbb{E}_0\left[\mathbf{h}_i\right], \mathbb{E}_1\left[\mathbf{h}_i\right])$, and the line $\mathbb{E}_0\left[\mathbf{h}_i\right] - \mathbb{E}_1\left[\mathbf{h}_i\right]$ connecting these two centers. According to Proposition 1, we readily get the following.

$$\mathbf{m}_h = \frac{\mathbb{E}_0\left[\mathbf{h}_i\right] + \mathbb{E}_1\left[\mathbf{h}_i\right]}{2} = \frac{1}{2}\left[\mathbf{W}_1^\top\left(\boldsymbol{\mu}_0+\boldsymbol{\mu}_1\right) + \mathbf{W}_2^\top\frac{p\boldsymbol{\mu}_0+q\boldsymbol{\mu}_1+p\boldsymbol{\mu}_1+q\boldsymbol{\mu}_0}{p+q}\right] \tag{11a}$$

$$= \frac{1}{2}\left[\mathbf{W}_1^\top\left(\boldsymbol{\mu}_0+\boldsymbol{\mu}_1\right) + \mathbf{W}_2^\top\left(\boldsymbol{\mu}_0+\boldsymbol{\mu}_1\right)\right] \tag{11b}$$

$$= \frac{\left(\mathbf{W}_1^\top+\mathbf{W}_2^\top\right)\left(\boldsymbol{\mu}_0+\boldsymbol{\mu}_1\right)}{2} \tag{11c}$$

$$\mathbb{E}_0\left[\mathbf{h}_i\right] - \mathbb{E}_1\left[\mathbf{h}_i\right] = \mathbf{W}_1^\top\boldsymbol{\mu}_0 + \mathbf{W}_2^\top\frac{p\boldsymbol{\mu}_0+q\boldsymbol{\mu}_1}{p+q} - \left(\mathbf{W}_1^\top\boldsymbol{\mu}_1 + \mathbf{W}_2^\top\frac{p\boldsymbol{\mu}_1+q\boldsymbol{\mu}_0}{p+q}\right) \tag{12a}$$

$$= \mathbf{W}_1^\top\left(\boldsymbol{\mu}_0-\boldsymbol{\mu}_1\right) + \mathbf{W}_2^\top\frac{p\left(\boldsymbol{\mu}_0-\boldsymbol{\mu}_1\right)+q\left(\boldsymbol{\mu}_1-\boldsymbol{\mu}_0\right)}{p+q} \tag{12b}$$

$$= \mathbf{W}_1^\top\left(\boldsymbol{\mu}_0-\boldsymbol{\mu}_1\right) + \mathbf{W}_2^\top\frac{(p-q)\left(\boldsymbol{\mu}_0-\boldsymbol{\mu}_1\right)}{p+q} \tag{12c}$$

$$= \left(\mathbf{W}_1^\top + \frac{p-q}{p+q}\mathbf{W}_2^\top\right)\left(\boldsymbol{\mu}_0-\boldsymbol{\mu}_1\right) \tag{12d}$$

$$\mathbf{s}_h = \frac{(\mathbb{E}_0\left[\mathbf{h}_i\right] - \mathbb{E}_1\left[\mathbf{h}_i\right])}{\|\mathbb{E}_0\left[\mathbf{h}_i\right] - \mathbb{E}_1\left[\mathbf{h}_i\right]\|_2} \tag{13}$$

$\square$

### C.3 PROOF OF PROPOSITION 3

**Lemma 1.** *For a $C-$category classification task, when training a one-layer MLP $\mathbf{h} = \mathbf{W}\mathbf{x}$ with $t$-softmax and cross-entropy functions, the $t$-softmax logits and gradients w.r.t. $\mathbf{h}$ are*

$$z_i = \frac{\exp\left(h_i/t\right)}{\sum_{k=1}^{C} \exp\left(h_k/t\right)}, \quad \mathcal{L} = \sum_{i=1}^{C} y_i \log \frac{1}{z_i}, \quad \delta\mathbf{h} = \frac{\partial \mathcal{L}}{\partial \mathbf{h}} = \frac{\mathbf{z} - \mathbf{y}}{t},$$

*where $c$ is the true label of $\mathbf{x}$ and $\mathbf{y}$ is the corresponding ground-truth one-hot vector.*

*Proof.* The derivative of $\mathcal{L}$ w.r.t. unnormalized logits $\mathbf{h}$ are composed by

$$\delta h_i = \frac{\partial \mathcal{L}}{\partial h_i} = -\sum_{k=1}^{C} y_k \frac{\partial \log z_k}{\partial h_i} = -\sum_{i=1}^{C} \frac{y_k}{z_k} \frac{\partial z_k}{\partial h_i} \tag{14a}$$

$$= -\frac{y_i}{z_i} \frac{\partial z_i}{\partial h_i} - \sum_{k \neq i}^{C} \frac{y_k}{z_k} \frac{\partial z_k}{\partial h_i} = -\frac{y_i}{z_i} \frac{1}{t} z_i(1 - z_i) - \sum_{k \neq i}^{C} \frac{y_k}{z_k} \frac{1}{t}(-z_k z_i) \tag{14b}$$

$$= \frac{-y_i + y_i z_i}{t} + \frac{1}{t} z_i \sum_{k \neq i}^{C} y_k \tag{14c}$$

$$= \frac{-y_i + z_i \sum_{k=i}^{C} y_k}{t} = \frac{z_i - y_i}{t}. \tag{14d}$$

$\square$

**Lemma 2.** *Denote by $\mathbf{P}$ the propagation matrix, by $\mathbf{W}$ the learnable weights, by $\mathbf{B}$ the learnable bias, by $\mathbf{H}$ the input features, and by $\delta\mathbf{H}' = \partial\mathcal{L}/\partial\mathbf{H}'$ the derivative matrix of scalar loss $\mathcal{L}$ w.r.t. the output features $\mathbf{H}'$ activated by an element-wise activation function $\phi(\cdot)$, then it follows that all related gradients of the layer*

$$\mathbf{X} = \mathbf{PHW} + \mathbf{B}, \ \mathbf{H}' = \phi(\mathbf{X}) \tag{15}$$

*are*

$$\delta\mathbf{W} = \frac{\partial \mathcal{L}}{\partial \mathbf{W}} = (\mathbf{PH})^{\top}\left(\delta\mathbf{H}' \circ \phi'(\mathbf{X})\right) \tag{16a}$$

$$\delta\mathbf{H} = \frac{\partial \mathcal{L}}{\partial \mathbf{H}} = \mathbf{P}^{\top}\left(\delta\mathbf{H}' \circ \phi'(\mathbf{X})\right)\mathbf{W}^{\top} \tag{16b}$$

$$\delta\mathbf{B} = \frac{\partial \mathcal{L}}{\partial \mathbf{B}} = \delta\mathbf{H}' \circ \phi'(\mathbf{X}), \tag{16c}$$

*where $\phi'(\mathbf{X})$ denotes the element-wise differentiation and $\circ$ is the Hadamard product.*

*Proof.* The differential of loss $\mathcal{L}$ is

$$d\mathcal{L} = \mathrm{Tr}\left\{\left(\frac{\partial \mathcal{L}}{\partial \mathbf{H}'}\right)^{\top} d\mathbf{H}'\right\}$$

$$= \mathrm{Tr}\left\{\left(\frac{\partial \mathcal{L}}{\partial \mathbf{H}'}\right)^{\top}\left[\frac{\partial \phi(\mathbf{X})}{\partial \mathbf{X}} \circ (\mathbf{PH}d\mathbf{W} + \mathbf{P}d\mathbf{HW} + d\mathbf{B})\right]\right\}. \tag{17a}$$

The Hadamard and Frobenius products commute in a trace operation, so we have the following when focusing on $\mathbf{W}$.

$$d\mathcal{L} = \text{Tr}\left\{ (\delta\mathbf{H}')^\top \left[ \phi'(\mathbf{X}) \circ (\mathbf{PH}d\mathbf{W}) \right] \right\} = \text{Tr}\left\{ (\delta\mathbf{H}' \circ \phi'(\mathbf{X}))^\top \mathbf{PH}d\mathbf{W} \right\}$$

$$= \text{Tr}\left\{ \left[ \mathbf{H}^\top \mathbf{P}^\top (\delta\mathbf{H}' \circ \phi'(\mathbf{X})) \right]^\top d\mathbf{W} \right\}$$

According to the relationships between matrix derivatives and total differential, i.e., $d\mathcal{L} = \text{Tr}\left\{ \frac{\partial\mathcal{L}^\top}{\partial\mathbf{R}} d\mathbf{R} \right\}$ (Minka, 2000), we readily get the gradient w.r.t. $\mathbf{W}$. Similarly, we can get the rest results by manipulating $\mathbf{H}$ and $\mathbf{B}$. □

**Lemma 3.** *Denote by $\mathbf{W}$ the learnable weights, by $\mathbf{B}$ learnable bias, by $\mathbf{H}$ the input features, and by $\delta\mathbf{H}' = \partial\mathcal{L}/\partial\mathbf{H}'$ the derivative matrix of scalar loss $\mathcal{L}$ w.r.t. the output features $\mathbf{H}'$ activated by an element-wise activation function $\phi(\cdot)$, then we can get that all involved gradients of the layer*

$$\mathbf{X} = \mathbf{HW} + \mathbf{B}, \ \mathbf{H}' = \phi(\mathbf{X}) \tag{19}$$

*are*

$$\delta\mathbf{W} = \frac{\partial\mathcal{L}}{\partial\mathbf{W}} = \mathbf{H}^\top (\delta\mathbf{H}' \circ \phi'(\mathbf{X})) \tag{20a}$$

$$\delta\mathbf{H} = \frac{\partial\mathcal{L}}{\partial\mathbf{H}} = (\delta\mathbf{H}' \circ \phi'(\mathbf{X})) \mathbf{W}^\top \tag{20b}$$

$$\delta\mathbf{B} = \frac{\partial\mathcal{L}}{\partial\mathbf{B}} = \delta\mathbf{H}' \circ \phi'(\mathbf{X}), \tag{20c}$$

*where $\phi'(\mathbf{X})$ denotes element-wise differentiation and $\circ$ is the Hadamard product.*

*Proof.* The proof can be finished in a way analogous to Lemma 2 or just by substituting $\mathbf{P} = \mathbf{I}$ into the conclusion of Lemma 2. □

**Proposition 3.** *For an $L$-layer GraphSAGE (stacked by Eq. 1) with cross entropy loss $\mathcal{L}$, $\delta\mathbf{W}$ is the matrix derivative, i.e., $\delta\mathbf{W} = \partial\mathcal{L}/\partial\mathbf{W}$, and the gradients of weights and hidden features are*

$$\delta\mathbf{X}^{(L-1)} = \mathbf{S}^\top \delta\mathbf{X}^{(L-1)}_{\mathcal{V}_L} = \frac{\mathbf{S}^\top\mathbf{S}}{t|\mathcal{V}_L|}\left( \mathbf{Z}^{(t)} - \mathbf{Y} \right) \tag{21a}$$

$$\delta\mathbf{W}_2^{(L-1)} = \left( \mathbf{PH}^{(L-1)} \right)^\top \delta\mathbf{X}^{(L-1)} \tag{21b}$$

$$\delta\mathbf{W}_1^{(L-1)} = \left( \mathbf{H}^{(L-1)} \right)^\top \delta\mathbf{X}^{(L-1)} \tag{21c}$$

$$\delta\mathbf{H}^{(L-1)} = \delta\mathbf{X}^{(L-1)} \left( \mathbf{W}_1^{(L-1)} \right)^\top + \mathbf{P}^\top \delta\mathbf{X}^{(L-1)} \left( \mathbf{W}_2^{(L-1)} \right)^\top \tag{21d}$$

$$\delta\mathbf{W}_2^{(l)} = \left( \mathbf{PH}^{(l)} \right)^\top \left( \delta\mathbf{H}^{(l+1)} \circ \phi'(\mathbf{X}^{(l)}) \right) \tag{21e}$$

$$\delta\mathbf{W}_1^{(l)} = \left( \mathbf{H}^{(l)} \right)^\top \left( \delta\mathbf{H}^{(l+1)} \circ \phi'(\mathbf{X}^{(l)}) \right) \tag{21f}$$

$$\delta\mathbf{H}^{(l)} = \left( \delta\mathbf{H}^{(l+1)} \circ \phi'(\mathbf{X}^{(l)}) \right) \left( \mathbf{W}_1^{(l)} \right)^\top + \mathbf{P}^\top \left( \delta\mathbf{H}^{(l+1)} \circ \phi'(\mathbf{X}^{(l)}) \right) \left( \mathbf{W}_2^{(l)} \right)^\top \tag{21g}$$

*where $t$ is the softmax temperature, $\mathbf{Y} \in \mathbb{R}^{N \times C}$ is stacked by one-hot ground-truth row vectors, $\phi'(\mathbf{X}^{(l)})$ is the derivative of element-wise activation function $\phi(\cdot)$ w.r.t. $\mathbf{X}^{(l)} = \mathbf{H}^{(l)}\mathbf{W}_1^{(l)} + \mathbf{PH}^{(l)}\mathbf{W}_2^{(l)}$, and $\mathbf{S} \in \{0,1\}^{|\mathcal{V}_L| \times N}$ is the row selection matrix to pick out the rows corresponding to the training nodes, e.g., $\mathbf{X}^{(L-1)}_{\mathcal{V}_L} = \mathbf{SX}^{(L-1)} \in \mathbb{R}^{|\mathcal{V}_L| \times C}$.*

*Proof.* In $L$-layer GraphSAGE, the data flow during training is

$$\mathbb{R}^{N \times d} \ni \mathbf{X} = \mathbf{H}^{(0)} \xrightarrow{1st\ conv} \mathbf{X}^{(0)} \xrightarrow{1st\ \phi} \mathbf{H}^{(1)} \xrightarrow{2nd\ conv} \mathbf{X}^{(1)} \xrightarrow{2nd\ \phi} \mathbf{H}^{(2)} \to \cdots$$

$$\to \mathbf{H}^{(L-1)} \xrightarrow{L-th\ conv} \mathbf{X}^{(L-1)} \xrightarrow{t-\text{softmax}} \mathbf{H}^{(L)} = \mathbf{Z} \in \mathbb{R}^{N \times C}$$

$$\xrightarrow{pick\ out\ \mathcal{V}_L} \mathbf{Z}_{\mathcal{V}_L} \xrightarrow{with\ \mathbf{Y}_{\mathcal{V}_L} = \mathbf{SY}} \mathcal{L} \in \mathbb{R} \tag{22}$$

We first derive the gradients of $\mathbf{X}^{(L-1)}$ and then backpropagate through the data flow. Softmax is row-wise, and the cross entropy loss is also computed per row/node. The averaged loss over nodes from the training set $\mathcal{V}_L$ is usually taken as the final objective. According to Lemma 1 we have

$$\delta\mathbf{X}_{\mathcal{V}_L}^{(L-1)} = \frac{1}{|\mathcal{V}_L|}\left(\mathbf{Z}_{\mathcal{V}_L}^t - \mathbf{Y}_{\mathcal{V}_L}\right) = \frac{\mathbf{S}}{|\mathcal{V}_L|t}\left(\mathbf{Z}^{(t)} - \mathbf{Y}\right). \tag{23}$$

Taking $\mathbf{X}_{\mathcal{V}_L}^{(L-1)} = \mathbf{S}\mathbf{X}^{(L-1)}$ as the layer in Lemma 3 leads to Eq. 21a. Each SAGE layer comprises two parts respectively reducible to Lemmas 2 and 3. Recursively applying these two lemmas per layer through the inverse data flow (22) gives the remaining derivatives. $\square$

### C.4 PROOF OF PROPOSITION 4

**Proposition 4.** *For a $k$-dimensional Gaussian random vector $\mathbf{x} \in \mathbb{R}^k$ that admits the mean $\mathbf{u} \in \mathbb{R}^k$, $\lambda = \sum_{i=1}^{k} \mu_i^2$ and covariance matrix $\sigma^2\mathbf{I} \in \mathbb{R}^{k \times k}$, the expectation and variance of the squared $L_2$ norm are respectively*

$$\mathbb{E}\left[\|\mathbf{x}\|_2^2\right] = k\sigma^2 + \lambda \tag{24}$$

$$\mathbb{D}\left[\|\mathbf{x}\|_2^2\right] = 4\sigma^2\lambda + 2k\sigma^4 = 2\sigma^2(2\lambda + k\sigma^2) \tag{25}$$

*Proof.* Let $x_i = \sigma y_i + u_i$ such that $y_i \sim \mathcal{N}(0,1)$, then we have

$$\|x\|_2^2 = \sum_{i=1}^{k} x_i^2 = \sum_{i=1}^{k}(\sigma y_i + u_i)^2 = \sum_{i=1}^{k}\sigma^2 y_i^2 + \sum_{i=1}^{k} u_i^2 + 2\sum_{i=1}^{k}\sigma u_i y_i. \tag{26}$$

Since $y_i^2 \sim \chi^2(k)$ and $\sum_{i=1}^{k} u_i^2 + 2\sum_{i=1}^{k}\sigma u_i y_i \sim \mathcal{N}(\lambda, 4\sigma^2\lambda)$, $\|\mathbf{x}\|_2^2$ is actually a generalized chi-square distribution whose mean and variance are

$$\mathbb{E}\left[\|\mathbf{x}\|_2^2\right] = \lambda + \sum_{j=1}^{k}\sigma^2(1+0) = k\sigma^2 + \lambda \tag{27a}$$

$$\mathbb{D}\left[\|\mathbf{x}\|_2^2\right] = 4\sigma^2\lambda + 2\sum_{j}^{k}\sigma^4(1+0) = 4\sigma^2\lambda + 2k\sigma^4 = 2\sigma^2(2\lambda + k\sigma^2) \tag{27b}$$

$\square$

## D PROOF OF THEOREM 1

*Proof.* A one-layer SAGE can be formulated as $\mathbf{h}_i = (\mathbf{W}_1)^{\top}\mathbf{x}_i + \frac{1}{N_i}\sum_{j \in \mathcal{N}(i)}(\mathbf{W}_2)^{\top}\mathbf{x}_j$. With both empirical risk minimization and MLP-to-GNN distillation, the loss function concentrating on node $i$ can be formulated as

$$\mathcal{L}_g = \frac{1}{|\mathcal{V}_L|}\ell_{CE}\left(y_i, [\mathbf{Z}_g]_i\right) + \frac{\alpha_1 t_1^2}{N}\ell_{KLD}\left(\left[\mathbf{Z}_m^{(t_1)}\right]_i, \left[\mathbf{Z}_g^{(t_1)}\right]_i\right). \tag{28}$$

Denoting the gradients from the two terms in Eq. 28 separately by $\nabla\mathcal{L}_{g,CE}$ and $\nabla\mathcal{L}_{g,KLD}$, one forward-backward optimization step on node $i$ with both terms leads to the SAGE weights that output $\mathbf{h}_i^{kd}$ while that with only the first term $\mathbf{h}_i$.

Though the full-batch update is common for most GNNs (Kipf & Welling, 2017; Klicpera et al., 2019; Chen et al., 2020; Lei et al., 2022; Li et al., 2022), the sampling-based GNNs using ego-nets (Hamilton et al., 2017; Veličković et al., 2018; Pal et al., 2020) or subgraphs (Chiang et al., 2019; Zeng et al., 2020; Fey et al., 2021; Shi et al., 2023) are popular for large graphs (Hu et al., 2021). Since the analysis of the finest ego-net granularity can extend to subgraph and whole-graph cases where the distillation effects of multiple nodes are aggregated, we concentrate on one node here for extendibility. For one sampled ego-net centering at node $i$, Proposition 3 can be accordingly adapted by appropriately adjusting $\mathbf{P}$, $\mathbf{X}$, $\mathbf{Z}$ and $\mathbf{Y}$.

Since minimizing the KLD is equivalent to minimizing the CE given the target distribution $P$

$$\ell_{KLD}(P,Q) = \sum_{x \in \mathcal{X}} p(x) \log \frac{1}{q(x)} - \sum_{x \in \mathcal{X}} p(x) \log \frac{1}{p(x)} = \ell_{CE}(P,Q) - H(P), \qquad (29)$$

the gradient of $\ell_{KLD}(P,Q)$ turns out to be that of $\ell_{CE}(P,Q)$. Thus the gradients from MLP-to-GNN knowledge distillation, i.e., the second term in Eq. 28, are

$$\frac{\partial}{\partial \mathbf{W}_1} \left[ \alpha_1 t_1^2 \ell_{KLD} \left( \left[ \mathbf{Z}_m^{(t_1)} \right]_i, \left[ \mathbf{Z}_g^{(t_1)} \right]_i \right) \right] = \frac{\alpha_1 t_1^2}{N} \frac{1}{t_1} \frac{\partial \ell_{CE} \left( \left[ \mathbf{Z}_m^{(t_1)} \right]_i, \left[ \mathbf{Z}_g^{(t_1)} \right]_i \right)}{\partial \mathbf{W}_1} \qquad (30a)$$

$$= \frac{\alpha_1 t_1}{N} \mathbf{x}_i \left( \left[ \mathbf{Z}_g^{(t_1)} \right]_i - \left[ \mathbf{Z}_m^{(t_1)} \right]_i \right) \qquad (30b)$$

$$\frac{\partial}{\partial \mathbf{W}_2} \left[ \alpha_1 t_1^2 \ell_{KLD} \left( \left[ \mathbf{Z}_m^{(t_1)} \right]_i, \left[ \mathbf{Z}_g^{(t_1)} \right]_i \right) \right] = \frac{\alpha_1 t_1}{N} [\mathbf{PX}]_i^\top \left( \left[ \mathbf{Z}_g^{(t_1)} \right]_i - \left[ \mathbf{Z}_m^{(t_1)} \right]_i \right) \qquad (30c)$$

$$= \frac{\alpha_1 t_1}{N} \left( \frac{1}{N_i} \sum_{j \in \mathcal{N}(i)} \mathbf{x}_j \right) \left( \left[ \mathbf{Z}_g^{(t_1)} \right]_i - \left[ \mathbf{Z}_m^{(t_1)} \right]_i \right) \qquad (30d)$$

$$= \frac{\alpha_1 t_1}{N} \bar{\mathbf{x}}_i \left( \left[ \mathbf{Z}_g^{(t_1)} \right]_i - \left[ \mathbf{Z}_m^{(t_1)} \right]_i \right), \qquad (30e)$$

according to the Proposition 3 variant adapted to the ego-net granularity. Let $\mathbf{W}_1$ and $\mathbf{W}_2$ be the weights updated by normal supervised training. Denoting by $\mathbf{W}_1^{kd}$ and $\mathbf{W}_2^{kd}$ the weights updated with both normal and distillation gradients, then we have

$$\mathbf{W}_1^{kd} = \mathbf{W}_1 - \eta \mathbf{x}_i \left( \left[ \mathbf{Z}_g^{(t_1)} \right]_i - \left[ \mathbf{Z}_m^{(t_1)} \right]_i \right) = \mathbf{W}_1 - \eta \mathbf{x}_i \mathbf{r}_i \qquad (31a)$$

$$\mathbf{W}_2^{kd} = \mathbf{W}_2 - \eta \bar{\mathbf{x}}_i \left( \left[ \mathbf{Z}_g^{(t_1)} \right]_i - \left[ \mathbf{Z}_m^{(t_1)} \right]_i \right) = \mathbf{W}_2 - \eta \bar{\mathbf{x}}_i \mathbf{r}_i, \qquad (31b)$$

where $\mathbf{r}_i$ denotes $\left[ \mathbf{Z}_g^{(t_1)} \right]_i - \left[ \mathbf{Z}_m^{(t_1)} \right]_i$ and $\eta = \eta' \alpha_1 t_1 / N$ integrates the learning rate $\eta'$ and other hyperparameters.

As the case for the nodes from class 0 is symmetric to that from class 1, we only prove for the nodes from class 0. For the node $i$ with label $y_i = 0$, $\mathbf{W}_1$ and $\mathbf{W}_2$ lead to a Gaussian distribution whose expectation is, according to Proposition 1,

$$\mathbb{E}[\mathbf{h}_i] = \mathbb{E} \left[ \mathbf{W}_1^\top \mathbf{x}_i + \mathbf{W}_2^\top \frac{1}{N_i} \sum_{j \in \mathcal{N}(i)} \mathbf{x}_j \right] = \mathbb{E} \left[ \mathbf{W}_1^\top \mathbf{x}_i + \mathbf{W}_2^\top \bar{\mathbf{x}}_i \right] \qquad (32a)$$

$$= \mathbf{W}_1^\top \boldsymbol{\mu}_0 + \mathbf{W}_2^\top \frac{p\boldsymbol{\mu}_0 + q\boldsymbol{\mu}_1}{p+q}. \qquad (32b)$$

By contrast, $\mathbf{W}_1^{kd}$ and $\mathbf{W}_2^{kd}$ result in

$$\mathbf{h}_i^{kd} = \left( \mathbf{W}_1^{kd} \right)^\top \mathbf{x}_i + \left( \mathbf{W}_2^{kd} \right)^\top \frac{1}{N_i} \sum_{j \in \mathcal{N}(i)} \mathbf{x}_j \qquad (33a)$$

$$= \left( \mathbf{W}_1^\top - \eta \mathbf{r}_i^\top \mathbf{x}_i^\top \right) \mathbf{x}_i + \left( \mathbf{W}_2^\top - \eta \mathbf{r}_i^\top \bar{\mathbf{x}}_i^\top \right) \bar{\mathbf{x}}_i, \qquad (33b)$$

whose expectation is

$$\mathbb{E} \left[ \mathbf{h}_i^{kd} \right] = \mathbb{E} \left[ \mathbf{W}_1^\top \mathbf{x}_i + \mathbf{W}_2^\top \bar{\mathbf{x}}_i \right] - \mathbb{E} \left[ \eta \mathbf{r}_i^\top \mathbf{x}_i^\top \mathbf{x}_i + \eta \mathbf{r}_i^\top \bar{\mathbf{x}}_i^\top \bar{\mathbf{x}}_i \right] \qquad (34a)$$

$$= \mathbb{E}[\mathbf{h}_i] - \eta \mathbf{r}_i^\top \left( \mathbb{E} \left[ \mathbf{x}_i^\top \mathbf{x}_i \right] + \mathbb{E} \left[ \bar{\mathbf{x}}_i^\top \bar{\mathbf{x}}_i \right] \right). \qquad (34b)$$

According to Proposition 4 and Eq. 10a, we have

$$\mathbb{E} \left[ \mathbf{x}_i^\top \mathbf{x}_i \right] = \mathbb{E} \left[ \|\mathbf{x}_i\|_2^2 \right] = \boldsymbol{\mu}_0^\top \boldsymbol{\mu}_0 + d\sigma^2 \qquad (35)$$

$$\mathbb{E} \left[ \bar{\mathbf{x}}_i^\top \bar{\mathbf{x}}_i \right] = \mathbb{E} \left[ \|\bar{\mathbf{x}}_i\|_2^2 \right] = \frac{(p\boldsymbol{\mu}_0^\top + q\boldsymbol{\mu}_1^\top)(p\boldsymbol{\mu}_0 + q\boldsymbol{\mu}_1) + d \left( p^2 + q^2 \right) \sigma^2}{(p+q)^2}. \qquad (36)$$

Substituting them into Eq. 34b gives the output expectation of GNN with MLP-to-GNN distillation

$$\mathbb{E}\left[\mathbf{h}_i^{kd}\right] = \mathbb{E}\left[\mathbf{h}_i\right] - \eta\mathbf{r}_i^\top\left[\boldsymbol{\mu}_0^\top\boldsymbol{\mu}_0 + d\sigma^2 + \frac{(p\boldsymbol{\mu}_0^\top + q\boldsymbol{\mu}_1^\top)(p\boldsymbol{\mu}_0 + q\boldsymbol{\mu}_1) + d\left(p^2 + q^2\right)\sigma^2}{(p+q)^2}\right]. \quad (37)$$

Substituting $\mathbf{W}_1^{kd}$ and $\mathbf{W}_2^{kd}$ into Proposition 2, we get the ideal decision boundary on $\mathbf{h}^{kd}$

$$\mathcal{P}_h^{kd} = \left\{\mathbf{h}^{kd} \mid \left(\mathbf{s}_h^{kd}\right)^\top\mathbf{h}^{kd} - \left(\mathbf{s}_h^{kd}\right)^\top\mathbf{m}_h^{kd}\right\}, \quad (38a)$$

$$\mathbf{s}_h^{kd} = \frac{\left(\mathbf{W}_1^\top - \eta\mathbf{r}_i^\top\mathbf{x}_i^\top + \frac{p-q}{p+q}\mathbf{W}_2^\top - \frac{p-q}{p+q}\eta\mathbf{r}_i^\top\bar{\mathbf{x}}_i^\top\right)(\boldsymbol{\mu}_0 - \boldsymbol{\mu}_1)}{\|numerator\|_2} \quad (38b)$$

$$= \frac{\mathbf{s}_h - \eta\left(\mathbf{r}_i^\top\mathbf{x}_i^\top - \frac{p-q}{p+q}\mathbf{r}_i^\top\bar{\mathbf{x}}_i^\top\right)(\boldsymbol{\mu}_0 - \boldsymbol{\mu}_1)}{\|numerator\|_2} \quad (38c)$$

$$\mathbf{m}_h^{kd} = \frac{\left[\mathbf{W}_1^\top + \mathbf{W}_2^\top - \eta\left(\mathbf{r}_i^\top\mathbf{x}_i^\top + \mathbf{r}_i^\top\bar{\mathbf{x}}_i^\top\right)\right](\boldsymbol{\mu}_0 + \boldsymbol{\mu}_1)}{2}, \quad (38d)$$

where $numerator$ means the corresponding numerator formula.

**Proof of the first conclusion in Theorem 1.**

In analogy with Ma et al. (2022), for any node $i$ we have the follows

$$\mathbb{P}(\mathbf{h}_i\text{ is mis-classified}) = \mathbb{P}\left(\mathbf{s}_h^\top\mathbf{h}_i - \mathbf{s}_h^\top\mathbf{m}_h \leq 0\right) \quad (39)$$

$$\mathbb{P}(\mathbf{h}_i^{kd}\text{ is mis-classified}) = \mathbb{P}\left(\left(\mathbf{s}_h^{kd}\right)^\top\mathbf{h}_i^{kd} - \left(\mathbf{s}_h^{kd}\right)^\top\mathbf{m}_h^{kd} \leq 0\right), \quad (40)$$

which turns out to be, in expectation,

$$\mathbb{P}(\mathbb{E}\left[\mathbf{h}_i\right]\text{ is mis-classified}) = \mathbb{P}\left(R = \mathbf{s}_h^\top\mathbb{E}\left[\mathbf{h}_i\right] - \mathbf{s}_h^\top\mathbf{m}_h \leq 0\right) \quad (41)$$

$$\mathbb{P}(\mathbb{E}\left[\mathbf{h}_i^{kd}\right]\text{ is mis-classified}) = \mathbb{P}\left(R^{kd} = \left(\mathbf{s}_h^{kd}\right)^\top\mathbb{E}\left[\mathbf{h}_i^{kd}\right] - \left(\mathbf{s}_h^{kd}\right)^\top\mathbf{m}_h^{kd} \leq 0\right). \quad (42)$$

If $R^{kd} > R$, $R^{kd}$ is less likely to be $\leq 0$ than $R$ and thus less likely to be misclassified. Now let's prove $R^{kd} > R$ holds no matter how heterophilous the graph is (or is changed by attacks to be).

Without loss of generality, we can always establish an appropriate coordinate system or apply suitable appropriate affine transform beforehand such that the two class centers of raw node features admit $\boldsymbol{\mu}_0 = -\boldsymbol{\mu}_1$ and the midpoints $\mathbf{m}_h$ and $\mathbf{m}_h^{kd}$ become zeros. Then we have

$$R = \mathbf{s}_h^\top\mathbb{E}\left[\mathbf{h}_i\right] \quad (43)$$

$$R^{kd} = \left(\mathbf{s}_h^{kd}\right)^\top\mathbb{E}\left[\mathbf{h}_i^{kd}\right] \approx \mathbf{s}_h^\top\mathbb{E}\left[\mathbf{h}_i^{kd}\right], \quad (44)$$

where $\mathbf{s}_h^{kd}$ from Eq. 38b is approximately equal to $\mathbf{s}_h$ due to the small $\eta = \eta'\alpha_1 t_1/N$ that results from a small learning rate $\eta'$ (usually at an order of magnitude less than $10^{-2}$) and a large $N$ (usually at an order of magnitude greater than $10^3$). Then we rewrite $R^{kd}$ with Eq. 37

$$R^{kd} = \mathbf{s}_h^\top\mathbb{E}\left[\mathbf{h}_i^{kd}\right] \quad (45a)$$

$$= \mathbf{s}_h^\top\left\{\mathbb{E}\left[\mathbf{h}_i\right] - \eta\mathbf{r}_i^\top\left[\boldsymbol{\mu}_0^\top\boldsymbol{\mu}_0 + d\sigma^2 + \frac{(p\boldsymbol{\mu}_0^\top + q\boldsymbol{\mu}_1^\top)(p\boldsymbol{\mu}_0 + q\boldsymbol{\mu}_1) + d\left(p^2 + q^2\right)\sigma^2}{(p+q)^2}\right]\right\} \quad (45b)$$

$$= \mathbf{s}_h^\top\mathbb{E}\left[\mathbf{h}_i\right] - \eta\mathbf{s}_h^\top\mathbf{r}_i^\top B = R - \eta\mathbf{s}_h^\top\mathbf{r}_i^\top B, \quad (45c)$$

where $\mathbf{s}_h^\top\mathbf{r}_i^\top$ determines the relative magnitude between $R^{kd}$ and $R$, and $B$ controls the magnitude gap between them. Recap that $\mathbf{r}_i = \left[\mathbf{Z}_g^{(t_1)}\right]_i - \left[\mathbf{Z}_m^{(t_1)}\right]_i$ where these two terms are the predicted distributions for node $i$ from GNN and MLP, respectively.

Considering that $\boldsymbol{\mu}_0 = -\boldsymbol{\mu}_1$ on the CSBM graph, the formula below is certainly positive.

$$B(p,q) = \boldsymbol{\mu}_0^\top\boldsymbol{\mu}_0 + d\sigma^2 + \frac{(p-q)^2\boldsymbol{\mu}_0^\top\boldsymbol{\mu}_0 + d\left(p^2 + q^2\right)\sigma^2}{(p+q)^2} \quad (46a)$$

$$= \frac{2\left(p^2 + q^2\right)\boldsymbol{\mu}_0^\top\boldsymbol{\mu}_0 + 2\left(p^2 + q^2 + pq\right)d\sigma^2}{(p+q)^2} > 0 \quad (46b)$$

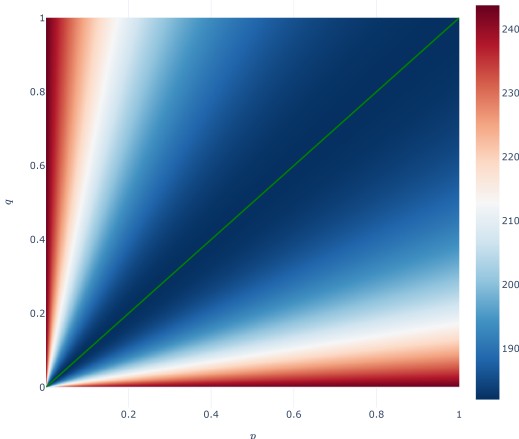

Figure 6: The value of $B$ changes with $p$ and $q$, and gets minimized when $p = q$, which is highlighted as the green line in the graph. In this picture, we set $d = 120$, $\sigma^2 = 1$, $\boldsymbol{\mu}_0^\top \boldsymbol{\mu}_0 = 2$.

On the aCSBM graph, Eq. 46b reduces to a quadratic form of $p$ since $p = 1 - q$

$$B(p) = 2\left(d\sigma^2 + 2\boldsymbol{\mu}_0^\top \boldsymbol{\mu}_0\right) p^2 - 2\left(d\sigma^2 + 2\boldsymbol{\mu}_0^\top \boldsymbol{\mu}_0\right) p + 2\left(d\sigma^2 + \boldsymbol{\mu}_0^\top \boldsymbol{\mu}_0\right), \tag{47}$$

where the derivative is $dB/dp = 2(2p - 1)(\sigma^2 + 2\boldsymbol{\mu}_0^\top \boldsymbol{\mu}_0)$ and the (global) minimum is hence $p = q = 0.5$. Hence, the minimum value of $B(p)$ is $(3d\sigma^2 + 2\boldsymbol{\mu}_0^\top \boldsymbol{\mu}_0)/2 > 0$. Therefore, for both CSBM and aCSBM graphs, $R^{kd} > R$ requires $\mathbf{s}_h^\top \mathbf{r}_i^\top < 0$ in Eq. 45c.

Let $\left[\mathbf{Z}_g^{(t_1)}\right]_i = [\phi_1, 1 - \phi_1]$ and $\left[\mathbf{Z}_{m,}^{(t_1)}\right]_i = [\phi_2, 1 - \phi_2]$. Then the delta between them turns out to be $\mathbf{r}_i = [\phi_1 - \phi_2, \phi_2 - \phi_1]$, $\phi_1, \phi_2 \in [0, 1]$. We further denote that

$$\mathbf{s}_h = \begin{bmatrix} x_0' \\ y_0' \end{bmatrix} = \frac{\left(\mathbf{W}_1^\top + \frac{p-q}{p+q}\mathbf{W}_2^\top\right)\boldsymbol{\mu}_0}{\|numerator\|_2} = \frac{\mathbf{W}\boldsymbol{\mu}_0}{\|\mathbf{W}\boldsymbol{\mu}_0\|_2}. \tag{48}$$

Then we have

$$\mathbf{s}_h^\top \mathbf{r}_i^\top = (x_0', y_0') \begin{bmatrix} \phi_1 - \phi_2 \\ \phi_2 - \phi_1 \end{bmatrix} = (\phi_1 - \phi_2)(x_0' - y_0'). \tag{49}$$

As stated in the theorem condition, i.e., Prospect Condition, the MLP has a higher prediction probability than the GNN on the ground-truth class, meaning that $\phi_1 < \phi_2$. Therefore, $\mathbf{s}_h^\top \mathbf{r}_i^\top < 0$ (i.e., $R^{kd} > R$) requires $x_0' < y_0'$, which can be satisfied by appropriately establishing the coordinate system (and hence appropriately assigning coordinates to $\boldsymbol{\mu}_0$) for arbitrary $2 \times d$ weight matrix $\mathbf{W}$.

**Proof of the second conclusion in Theorem 1**

According to Eq. 45a, the distillation effect has an amplitude factor $B$, which is determined by the class centers and heterophily. So we can build the relationship between heterophily and the MLP-to-GNN distillation effect.

To determine the influence of graph heterophily on $B$, we examine the partial derivatives of Eq. 46b w.r.t. $p$ and $q$, which are

$$\frac{\partial B}{\partial p} = \frac{2q(p - q)(d\sigma^2 + 2\boldsymbol{\mu}_0^\top \boldsymbol{\mu}_0)}{(p + q)^3} \tag{50}$$

$$\frac{\partial B}{\partial q} = \frac{2p(q - p)(d\sigma^2 + 2\boldsymbol{\mu}_0^\top \boldsymbol{\mu}_0)}{(p + q)^3}. \tag{51}$$

Hence the stationary points of $B(p, q)$ are on the line $p = q$. Despite the negative definite Hessian indicating that they are saddle points, for any fixed $q \in (0, 1]$, $B(p)$ has a local minimum $p = q$. Therefore, these saddle points are actually minima, meaning that the weakest MLP-to-GNN distillation effect emerges at the heterophilous demarcation of HR $= p/(p + q) = 0.5$. For an

intuitive understanding, we depict the value of $B$ with respect to various $p$ and $q$ in Figure 6, which also shows that the strongest effect is approached at the extreme homophily ratios, i.e., the left-top and right-bottom corners in the figure. Regarding the aCSBM graph, a similar analysis of Eq. 47 can lead to the same conclusion. $\qquad\square$

## E    PROOF OF THEOREM 2

### E.1    AUXILIARIES FOR THEOREM 2 PROOF

**Definition 1.** (Lipschitz constant) A function $f : \mathcal{X} \to \mathcal{Y}$ is $K$-Lipschitz (continuous) w.r.t. a norm $\|\cdot\|$ if there is a constant $K$ such that

$$\forall x_1, x_2 \in \mathcal{X}, \|f(x_1) - f(x_2)\| \leq K\|x_1 - x_2\|. \tag{52}$$

The smallest $K$ admits the inequality is one of the Lipschitz constants of $f$ and denoted as $\|f\|_{Lip}$.

**Theorem 3** (Rademacher Federer (2014), Theorem 3.1.6; Virmaux & Scaman (2018), Theorem 1). *If $f : \mathbb{R}^n \to \mathbb{R}^m$ is a locally Lipschitz continuous function[a] , then $f$ is differentiable almost everywhere. Moreover, if $f$ is Lipschitz continuous, then*

$$\|f\|_{Lip} = \sup_{x \in \mathbb{R}^n} \|\nabla f(x)\|_2, \tag{53}$$

*where $\|\mathbf{M}\|_2 = \sup_{\|x\| \leq 1} \|\mathbf{M}x\|_2$ is the operator norm of matrix $\mathbf{M} \in \mathbb{R}^{m \times n}$.*

---

[a]The functions whose restriction to some neighborhood around any point is Lipschitz are *locally* Lipschitz.

**Theorem 4** (Banach fixed-point theorem). *Let $(\mathcal{X}, \mathcal{D})$ be a non-empty complete metric space with a contraction mapping $f : \mathcal{X} \to \mathcal{X}$ such that*

$$\mathcal{D}(f(x_1), f(x_2)) \leq q\mathcal{D}(x_1, x_2), \exists q \in [0, 1), \tag{54}$$

*then there is a unique fixed-point $f(x^*) = x^*$ that can be found by generating a sequence $\{x_n \mid x_{n+1} = f(x_n)\}_{n \in \mathbb{N}}$ with an initial point $x^{(0)} \in \mathcal{X}$. $\mathcal{D}$ is usually a vector or matrix norm.*

**Proposition 5.** *Given two functions $f : \mathcal{X} \to \mathcal{U}$ and $g : \mathcal{U} \to \mathcal{Y}$ whose Lipschitz constants are respectively $\|g\|_{Lip}$ and $\|f\|_{Lip}$, the Lipschitz constant of the composite function $g \circ f : \mathcal{X} \to \mathcal{Y}$ satisfies the inequality*

$$\|g \circ f\|_{Lip} \leq \|g\|_{Lip}\|f\|_{Lip}. \tag{55}$$

*Proof.* Let $u = f(x)$ and $y = g(u)$. The gradient of $g \circ f$ is

$$\nabla(g \circ f)(x) = \nabla g(f(x))\nabla f(x). \tag{56}$$

Then we have

$$\sup_{\|x\| \leq 1} \|[\nabla g(f(x))][\nabla f(x)]\|_2 \leq \sup_{\|u\| \leq 1} \|[\nabla g(u)]\|_2 \sup_{\|x\| \leq 1} \|[\nabla f(x)]\|_2, \tag{57}$$

which amounts to the desired according to Theorem 3. $\qquad\square$

**Lemma 4** (Gao & Pavel (2018), Proposition 4). *The $t$-softmax function is Lipschitz with respect to $L_2$ vector norm and the Lipschitz constant is $1/t$. That is, for all $x_1, x_2 \in \mathbb{R}^n$*

$$\|\mathrm{softmax}_t(x_1) - \mathrm{softmax}_t(x_2)\|_2 \leq \frac{1}{t}\|x_1 - x_2\|_2. \tag{58}$$

**Proposition 6.** *The Lipschitz constant w.r.t. the Frobenius norm of the linear mapping operator $\mathbf{Y}_{m \times n} = \mathbf{A}_{m \times k}\mathbf{X}_{k \times n}$ is the spectral norm of $\mathbf{A}$, and that of the row-wise $t$-softmax function $\mathbf{Y}_{m \times n} = \mathrm{softmax}_t(\mathbf{X}_{m \times n})$ is $1/t$.*

*Proof.* 1) According to Theorem 3, the Lipschitz constant of matrix-vector multiplication $\mathbf{y}_{m \times 1} = \mathbf{A}_{m \times k}\mathbf{x}_{k \times 1}$ w.r.t. $L_2$ vector norm can be directly obtained, which is the spectral norm $\sigma(\mathbf{A})$, as shown in Virmaux & Scaman (2018).

$$\|\mathbf{y}_1 - \mathbf{y}_2\|_2 \leq \sigma(\mathbf{A})\|\mathbf{x}_1 - \mathbf{x}_2\|_2 \tag{59}$$

Given two matrices $\mathbf{X}_1 = [\mathbf{x}_{11}, \mathbf{x}_{12}, ..., \mathbf{x}_{1k}]$ and $\mathbf{X}_2 = [\mathbf{x}_{21}, \mathbf{x}_{22}, ..., \mathbf{x}_{2k}]$ composed by column vectors, we have

$$\mathbf{Y}_1 = [\mathbf{y}_{11}, \mathbf{y}_{12}, ..., \mathbf{y}_{1k}] = \mathbf{A}\mathbf{X}_1 = [\mathbf{A}\mathbf{x}_{11}, \mathbf{A}\mathbf{x}_{12}, ..., \mathbf{A}\mathbf{x}_{1k}] \tag{60}$$

$$\mathbf{Y}_2 = [\mathbf{y}_{21}, \mathbf{y}_{22}, ..., \mathbf{y}_{2k}] = \mathbf{A}\mathbf{X}_2 = [\mathbf{A}\mathbf{x}_{21}, \mathbf{A}\mathbf{x}_{22}, ..., \mathbf{A}\mathbf{x}_{2k}]. \tag{61}$$

The squared $L_2$ vector norms of all column vectors amount to the squared Frobenius norm

$$\|\mathbf{X}_1 - \mathbf{X}_2\|_F^2 = \sum_{j=1}^{k} \|\mathbf{x}_{1j} - \mathbf{x}_{2j}\|_2^2 \tag{62}$$

$$\|\mathbf{Y}_1 - \mathbf{Y}_2\|_F^2 = \sum_{j=1}^{k} \|\mathbf{y}_{1j} - \mathbf{y}_{2j}\|_2^2 \tag{63}$$

Thus according to Eq. 59, we have

$$\|\mathbf{Y}_1 - \mathbf{Y}_2\|_F^2 \le \sigma^2(\mathbf{A})\|\mathbf{X}_1 - \mathbf{X}_2\|_F^2, \tag{64}$$

meaning that the Lipschitz constant of linear mapping w.r.t. the Frobenius norm is $\sigma(\mathbf{A})$.

2) By distributing the Frobenius norm onto rows and using Lemma 4 on each row, we can get $1/t$ in a way like the proof of the first conclusion. □

### E.2 MAIN PROOF OF THEOREM 2

*Proof.* For the symbol simplicity, we replace $\mathcal{L}_m$ with $L_m$ and $\mathbf{W}$ with $w$ here. The local or global optimal MLP weight $w^*$ should satisfy $\nabla_w L_m(w^*) = 0$, meaning that $w^* = w^* - \eta \nabla_w L_m(w^*)$. We can thus construct a function $G(w) = w - \eta \nabla_W L_m(w)$ where $\eta$ is the step size of gradient decent and $w^*$ is the fixed-point of $G(w)$. We now derive the existence condition of $w^*$ based on Banach fixed-point theorem, i.e., Theorem 3.

Let the space metric $\mathcal{D}$ be Frobenius matrix norm. If follows that for any two weights $w_1, w_2 \in \mathcal{W}$

$$\|G(w_1) - G(w_2)\|_F^2 = \|w_1 - w_2 - \eta\left(\nabla L_m(w_1) - \nabla L_m(w_2)\right)\|_F^2 \tag{65a}$$

$$= \|w_1 - w_2\|_F^2 + \eta^2 \|\nabla L_m(w_1) - \nabla L_m(w_2)\|_F^2 - 2\operatorname{Tr}\left\{\eta(w_1 - w_2)^\top \left[\nabla L_m(w_1) - \nabla L_m(w_2)\right]\right\}. \tag{65b}$$

To apply Banach fixed-point theorem, we need construct the inequality relationship between $\|G(w_1) - G(w_2)\|_F^2$ and $\|w_1 - w_2\|_F^2$, implying that the second and third terms in Eq. 65b should be tackled. Since the assumption in Theorem 2 copes with the third term, we move on to the second term now.

The loss function $L_m$ of Prospect-MLP is Eq. 3b, the gradient of which w.r.t. MLP weights can be obtained by recursively applying Lemmas 1 and 3. For simplicity, we denote by $f_t(\cdot)$ the $t$-softmax function and only consider the weight of the last MLP layer. In spite of this, our proof can be extended to an arbitrary MLP layer since the main tool, i.e., Proposition 5, can be extended to an arbitrary layer[4]. However, such a generalization would lead to highly cumbersome formulas without providing additional insights. To proceed, we expand the second term to

$$\nabla L_m(w_1) - \nabla L_m(w_2)$$

$$= \frac{\mathbf{H}^\top}{N_{tr}} \mathbf{S}^\top \mathbf{S}\left(f(\mathbf{H}w_1) - \mathbf{Y}\right) + \frac{\alpha t_2}{N} \mathbf{H}^\top \left(f_{t_2}(\mathbf{H}w_1) - \mathbf{Z}_g^{t_2}\right)$$

$$- \frac{\mathbf{H}^\top}{N_{tr}} \mathbf{S}^\top \mathbf{S}\left(f(\mathbf{H}w_2) - \mathbf{Y}\right) - \frac{\alpha t_2}{N} \mathbf{H}^\top \left(f_{t_2}(\mathbf{H}w_2) - \mathbf{Z}_g^{t_2}\right) \tag{66a}$$

$$= \frac{\mathbf{H}^\top}{N_{tr}} \mathbf{S}^\top \mathbf{S}\left[f(\mathbf{H}w_1) - f(\mathbf{H}w_2)\right] + \frac{\alpha t_2}{N} \mathbf{H}^\top \left[f_{t_2}(\mathbf{H}w_1) - f_{t_2}(\mathbf{H}w_2)\right], \tag{66b}$$

where $\mathbf{H}$ is the input feature matrix of last MLP layer and $N_{tr} = |\mathcal{V}_L|$ is the size of training set.

---

[4] The extended proof also employs that the common activation functions (e.g., ReLU, LeakyReLU, Sigmoid, Tanh, and Sigmoid) are both 1-Lipschitz.

We construct two functions

$$g_1(w) = \frac{\mathbf{H}^\top}{N_{tr}} \mathbf{S}^\top \mathbf{S} f(\mathbf{H}w) \tag{67}$$

$$g_2(w) = \frac{\alpha t_2}{N} \mathbf{H}^\top f_{t_2}(\mathbf{H}w), \tag{68}$$

and then Eq. 66b turns out to be

$$\nabla L_m(w_1) - \nabla L_m(w_2) = g_1(w_1) - g_1(w_2) + g_2(w_1) - g_2(w_2). \tag{69}$$

It follows that

$$\|\nabla L_m(w_1) - \nabla L_m(w_2)\|_F^2 \tag{70a}$$

$$= \|g_1(w_1) - g_1(w_2) + g_2(w_1) - g_2(w_2)\|_F^2 \tag{70b}$$

$$= \|g_1(w_1) - g_1(w_2)\|_F^2 + \|g_2(w_1) - g_2(w_2)\|_F^2 \\ + 2\|g_1(w_1) - g_1(w_2)\|_F \|g_2(w_1) - g_2(w_2)\|_F \tag{70c}$$

$$\leq \|g_1(w)\|_{Lip}^2 \|w_1 - w_2\|_F^2 + \|g_2(w)\|_{Lip}^2 \|w_1 - w_2\|_F^2 \\ + 2\|g_1(w)\|_{Lip} \|g_2(w)\|_{Lip} \|w_1 - w_2\|_F^2 \tag{70d}$$

To find the inequality relationship between $\|\nabla L_m(w_1) - \nabla L_m(w_2)\|_F^2$ and $\|w_1 - w_2\|_F^2$, we need to check out the Lipschitz constants of $g_1(w)$ and $g_2(w)$. According to Propositions 6 and 5, we have

$$\|g_1(w)\|_{Lip} = \frac{1}{N_{tr}} \sigma(\mathbf{H}^\top \mathbf{S}^\top \mathbf{S}) \sigma(\mathbf{H}) \tag{71}$$

$$\|g_2(w)\|_{Lip} = \frac{\alpha t_2}{N} \sigma^2(\mathbf{H}). \tag{72}$$

Then the upper bound (i.e., Eq. 73) becomes

$$\|\nabla L_m(w_1) - \nabla L_m(w_2)\|_F^2 \leq U_1 = \left[ \frac{1}{N_{tr}} \sigma(\mathbf{H}^\top \mathbf{S}^\top \mathbf{S}) \sigma(\mathbf{H}) + \frac{\alpha t_2}{N} \sigma^2(\mathbf{H}) \right]^2 \|w_1 - w_2\|_F^2. \tag{73}$$

Substituting Eq. 73 and the theorem assumption Eq. 6 into Eq. 65b leads to

$$\|G(w_1) - G(w_2)\|_F^2 \tag{74a}$$

$$\leq \|w_1 - w_2\|_F^2 + \eta^2 \left[ \frac{1}{N_{tr}} \sigma(\mathbf{H}^\top \mathbf{S}^\top \mathbf{S}) \sigma(\mathbf{H}) + \frac{\alpha t_2}{N} \sigma^2(\mathbf{H}) \right]^2 \|w_1 - w_2\|_F^2 \tag{74b}$$

$$= \left\{ 1 + \eta^2 \left[ \frac{1}{N_{tr}} \sigma(\mathbf{H}^\top \mathbf{S}^\top \mathbf{S}) \sigma(\mathbf{H}) + \frac{\alpha t_2}{N} \sigma^2(\mathbf{H}) \right]^2 - 2\eta u \right\} \|w_1 - w_2\|_F^2 \tag{74c}$$

$$= \left( 1 + \eta^2 \beta^2 - 2\eta u \right) \|w_1 - w_2\|_F^2. \tag{74d}$$

$$\square$$

## F    COMPARISON WITH RELATED WORK

**Defense Methods**   Previous adversarial defense methods fall into four types. **1) Adversarial training.** Perturbing the clean adjacency matrix with random flips (Dai et al., 2018), gradient projection descent (Xu et al., 2019), or Nettack (Chen et al., 2019) during training can confer some evasion attack robustness. But it may impede training efficiency, fail to withstand poisoning attacks, and risk clean accuracy vs robustness trade-offs (Pang et al., 2022). **2) Preprocess purification.** The susceptible components, like high-rank adjacency components (Entezari et al., 2020) or dissimilar connections (Wu et al., 2019b), are removed before training/inference. **3) Learning purification.** Learning clean graphs during training can be done by assigning low propagation weights for susceptible elements (Zhu et al., 2019), attenuating edges connecting dissimilar nodes (Zhang & Zitnik, 2020), optimizing a dense adjacency matrix towards the properties of clean homophilous graphs (Jin et al., 2020), and extracting robust node features for subsequent reconstruction (Li et al., 2022). **4)**

**Heterphilous Design.** Many attack algorithms, e.g., (Zügner et al., 2018; Zügner & Günnemann, 2019), insert heterophily into homophilous graphs (Wu et al., 2019b; Zhang & Zitnik, 2020; Jin et al., 2020; Zhu et al., 2022; Li et al., 2022), to degrade GNNs designed assuming homophily. In contrast, GNNs designed for heterophily, including $H_2$GCN (Zhu et al., 2020) and EvenNet (Lei et al., 2022), can more or less adapt to the altered homophily levels. Hence, they exhibit some inherent robustness against adversarial attacks (Zhu et al., 2022).

Compared to type 1 models, PROSPECT defends against both poisoning and evasion attacks without any potential accuracy-robustness trade-offs. Unlike types 2 and 3, PROSPECT inherently adapts to heterophily and has no extra purification costs. Versus type 4, PROSPECT enables integration with simple GNNs (Ying et al., 2018; He et al., 2020; Pal et al., 2020) used downstream, rather than being ad-hoc. And PROSPECT has an inference scalability as high as MLPs, unlike all four types. Most crucially, the adversarial robustness of PROSPECT stems from counteracting the decreased true class probability, theoretically suitable for any graph and any effective structure attacks.

**GD-MLPs** PROSPECT pioneers online GD-MLPs, versus offline frameworks like GLNN (Zhang et al., 2022) and NOSMOG (Tian et al., 2023b). GLNN transfers the GNN knowledge learned from the graph structure and node features to MLPs that rely on no graph structures, by matching the temperated logits (Hinton et al., 2015; Phuong & Lampert, 2019). Such design is, however, shown unable to align the input node feature to the label space fully, capture the soft structural representational similarity among nodes, and resist node feature noises. To address these problems, NOSMOG incorporates the structure embeddings, e.g., DeepWalk (Perozzi et al., 2014), into node features, distills the relative node similarity (Tung & Mori, 2019), and employs Project Gradient Descent adversarial training (PGD-AT) (Madry et al., 2022) on node features.

The differences between these offline GD-MLPs and our PROSPECT are as follows. **1)** GLNN are vulnerable to poisoning structure attacks, while PROSPECT resists both poisoning and evasion structure attacks. **2)** The performance of offline GD-MLPs is constrained by the pre-trained teachers, whereas PROSPECT transcends this limit through mutual distillation. **3)** PROSPECT simultaneously trains robust GNNs and MLPs in one phase, avoiding the complex two-phase of offline distillation. **4)** PROSPECT concerns about the structure adversarial robustness, which is neglected by GLNN and NOSMOG but more destructive and prevalent in the graph machine learning context. In fact, the feature robustness methods such as the adversarial feature augmentation in NOSMOG are orthogonal to and compatible with PROSPECT, so we can integrate these modules into PROSPECT when necessary.

# G EXPERIMENTAL DETAILS: DATASETS, MODELS, AND MORE RESULTS

## G.1 DATASETS, MODELS, AND EXPERIMENT SETTINGS

**Datasets.** We consider eight public graph datasets: Cora, Citeseer, UAI (Sen et al., 2008), ACM (Wang et al., 2019), Polblogs (Adamic & Glance, 2005), Chameleon, Texas Pei et al. (2020) and CoraML (Bojchevski & Günnemann, 2018). The statistics of the largest connected components of these five homophilous and three heterophilous graphs are summarized in Table 4.

Table 4: The data statistics of the largest connected components. HR refers to the homophily ratio.

| Dataset | #Nodes | #Edges | #Features | #Classes | HR |
|---------|--------|--------|-----------|----------|-------|
| Cora | 2485 | 5069 | 1433 | 7 | 0.804 |
| Citeseer | 2110 | 3668 | 3703 | 6 | 0.736 |
| ACM | 3025 | 13128 | 1870 | 3 | 0.821 |
| Polblogs | 1222 | 16714 | 1490 | 2 | 0.906 |
| CoraML | 2810 | 7981 | 2879 | 7 | 0.784 |
| Chameleon | 2277 | 31371 | 2325 | 5 | 0.230 |
| Texas | 183 | 279 | 1703 | 5 | 0.061 |
| UAI | 3067 | 28311 | 4973 | 19 | 0.364 |

**Data Splitting** Following Zügner & Günnemann (2019), the largest connected component of each graph is taken and split with 10% nodes for training, 10% validation, and 80% testing. Furthermore,

we repeat such 1:1:8 data splitting with 5 random seeds on each graph, and the results averaged over these 5 distinct splits are reported as the eventual performance on that graph.

**Attack Methods**   With a two-layer GCN (Kipf & Welling, 2017) as the surrogate model, we use MetaAttack (Zügner & Günnemann, 2019), an effective meta-learning-based (Finn et al., 2017) poisoning method, for robustness evaluation, as most GNN defense works. For each graph split, we run the MetaAttack implemented in DeepRobust (Li et al., 2021) at 5%, 10%, 15%, and 20% attack budgets[5]. The accuracy under a budget, e.g., 20%, averages the results over 5 random splits. We abbreviate attack settings like Cora-Meta-20, which stands for the Cora dataset attacked by MetaAttack with a budget of 20%.

**Transductive vs. Semi-inductive**   In the transductive setting, the supervised signals come from the training set $\mathcal{V}_L$ while the distillation signals are from $\mathcal{V}_{obs} = \mathcal{V}_L \cup \mathcal{V}_{val} \cup \mathcal{V}_{test}$. In the inductive setting, we distill on the observed node set $\mathcal{V}_{obs} = \mathcal{V}_L \cup \mathcal{V}_{val}$ and test on a disjoint $\mathcal{V}_{test}$ . In the semi-inductive setting, $\mathcal{V}_{test}$ is further divided into two disjoint observed and inductive subsets $\mathcal{V}_{test} = \mathcal{V}_{test}^{trans} \cup \mathcal{V}_{test}^{ind}$ such that the distillation is performed on $\mathcal{V}_{obs} = \mathcal{V}_L \cup \mathcal{V}_{val} \cup \mathcal{V}_{test}^{trans}$. For example, the production setting proposed in Zhang et al. (2022) is semi-inductive. Across all three settings, $\mathcal{V}_L$, $\mathcal{V}_{val}$, $\mathcal{V}_{test}^{ind}$, and $\mathcal{V}_{test}^{trans}$ are disjoint and we report the accuracy on $\mathcal{V}_{test}$ when the model performs best on the validation set $\mathcal{V}_{val}$. In the settings having inductive test nodes, the edges between $\mathcal{V}_{obs}$ and these nodes are removed during training but used during the inference of neighbor-aggregation-based GNNs. Nevertheless, in our semi-inductive experiments (Section 6.4), the gray-box poisoning attack modifications are still generated in a transductive style. That is, the attackers are aware of the full graph during training the surrogate model and making attack decisions. Such an attack manipulates both the training graph supported by $\mathcal{V}_L \cup \mathcal{V}_{val} \cup \mathcal{V}_{test}^{trans}$ and the testing graph supported by $\mathcal{V}_L \cup \mathcal{V}_{val} \cup \mathcal{V}_{test}^{trans} \cup \mathcal{V}_{test}^{ind}$, so we regard it as a hybrid attack comprised of both poisoning and evasion elements.

**Models**   Here we instantiate Prospect-GNN with SAGE, which is denoted as Prospect-SAGE. For a fair comparison, the hidden sizes of Prospect-SAGE, all baseline GNNs (Kipf & Welling, 2017; Hamilton et al., 2017; Zhu et al., 2019; Wu et al., 2019b; Entezari et al., 2020; Zhang & Zitnik, 2020; Jin et al., 2020; Lei et al., 2022; Li et al., 2022) and GLNN teachers (Zhang et al., 2022) are fixed to 64. We denote the MLP with $64 \times i$ hidden units as MLPw$i$, and the GLNN employing MLPw$i$ as GLNNw$i$. PROSPECT comprises one 2-layer SAGE and an MLP with a hidden size from $64 \times \{2, 3, 4, 5\}$.

**Experiment Environment**   Part of the experiments are conducted on an Ubuntu20.04 server equipped with one Intel(R) Xeon(R) Gold 6240C CPU and four NVIDIA GeForce RTX 4090 GPUs. And the rest experiments are run on a Windows11 workstation equipped with one Intel(R) Core(TM) i7-12700 and one NVIDIA GeForce RTX 4090 GPU.

For RGCN, Jaccard, SVD, and ProGNN, we adopt the implementations from DeepRobust (Li et al., 2021). The implementation of Guard follows their official implementation[6]. The implementation of STABLE adheres to the official implementation[7]. The implementation of EvenNet is based on the official implementation[8]. And we implement MLP, SGC, SAGE, GLNN and PROSPECT based on Lightning 2.0.3 (Falcon & The PyTorch Lightning team, 2019) and PyG 2.3.1 (Fey & Lenssen, 2019), which is built on Pytorch 2.0.1 (Paszke et al., 2019) and CUDA Toolkit 11.8.

Besides, to facilitate future research on GD-MLPs, we have designed some LightningModules to eliminate boilerplate code so that researchers can focus on designing their distillation modules. These LightningModules support: **1)** offline, online, and hybrid distillation; **2)** multiple teachers and/or multiple students; **3)** plug-in (customized) feature and/or logit distillation strategy modules; **4)** mini-batch sampling or full-batch training and distillation; **5)** unified and friendly data interfaces for both clean and attacked graph data; **6)** easy switch between transductive, inductive and semi-inductive evaluation settings; **7)** easy switch between poisoning and evasion attack evaluation

---

[5]A budget of 5% (15%) means the attacker can flip $0.05|\mathcal{E}|$ ($0.15|\mathcal{E}|$) entries in the adjacency matrix.

[6]https://github.com/mims-harvard/GNNGuard

[7]https://github.com/likuanppd/STABLE

[8]https://github.com/leirunlin/evennet

Table 5: The full robustness results on Cora and Citeseer. The mean and std over five splits are reported. The top two performing models are highlighted in bold, with the best further underlined.

| | Cora (HR=0.804) | | | | Citeseer (HR=0.736) | | | |
|---|---|---|---|---|---|---|---|---|
| | 5% | 10% | 15% | 20% | 5% | 10% | 15% | 20% |
| MLP | 65.69±1.43 | | | | 66.01±1.37 | | | |
| MLPw2 | 66.58±0.94 | | | | 66.82±1.83 | | | |
| MLPw4 | 67.02±1.12 | | | | 66.43±1.73 | | | |
| GCN | 79.80±1.36 | 75.20±2.17 | 70.67±3.23 | 64.93±3.37 | 72.03±1.23 | 68.98±2.46 | 64.74±2.70 | 61.39±3.27 |
| SGC | 79.43±1.67 | 74.27±2.06 | 68.70±2.58 | 63.08±2.59 | 71.94±1.31 | 68.84±2.19 | 64.51±2.44 | 61.23±2.67 |
| SAGE | 80.33±1.31 | 77.20±1.90 | 73.79±2.70 | 71.39±2.74 | 72.68±1.25 | 72.29±0.52 | 70.40±1.05 | 67.97±2.31 |
| RGCN | 79.12±1.30 | 74.92±2.20 | 70.30±2.47 | 63.92±2.69 | 71.71±2.04 | 68.46±2.25 | 64.02±1.90 | 61.07±1.97 |
| SVD | 77.12±0.49 | 74.73±1.83 | 71.97±2.27 | 69.40±2.44 | 69.82±0.86 | 68.56±1.76 | 65.15±2.01 | 62.46±1.12 |
| Jaccard | 80.36±0.74 | 77.14±1.26 | 74.54±1.50 | 71.63±1.26 | 72.18±1.81 | 70.10±1.92 | 66.96±2.71 | 64.44±2.76 |
| Guard | 76.76±0.78 | 74.93±1.79 | 74.49±1.96 | 74.00±1.85 | 69.79±1.24 | 68.89±1.33 | 67.35±0.62 | 67.15±1.28 |
| ProGNN | 78.87±1.51 | 74.28±1.83 | 70.25±3.21 | 64.33±3.55 | 71.60±1.84 | 69.31±2.00 | 65.12±2.38 | 61.17±2.74 |
| STABLE | 81.68±0.73 | 80.09±0.38 | 79.20±1.00 | 77.58±1.53 | 74.33±1.08 | 73.95±0.74 | 73.32±1.14 | 72.50±0.88 |
| EvenNet | **83.40±0.96** | **80.80±0.63** | 78.36±1.10 | 75.14±1.79 | 74.08±1.02 | 72.97±1.19 | 70.95±1.71 | 69.40±1.14 |
| GLNN | 80.12±1.37 | 77.35±1.97 | 73.95±1.92 | 71.94±3.22 | 74.25±1.20 | 73.46±0.73 | 71.92±1.38 | 70.02±2.10 |
| GLNNw2 | 80.22±1.56 | 77.31±1.87 | 74.14±3.10 | 71.90±3.18 | 74.44±1.51 | 73.67±0.36 | 72.13±1.36 | 70.11±2.75 |
| GLNNw4 | 80.21±1.51 | 77.34±1.87 | 74.60±2.24 | 72.76±2.82 | 74.01±1.32 | 73.37±0.71 | 71.94±1.38 | 70.37±2.72 |
| Prospect-SAGE | **83.05±1.05** | **81.13±1.45** | **79.53±2.17** | **79.59±1.00** | **75.01±0.75** | **75.23±1.21** | **74.81±0.41** | **75.87±0.96** |
| Prospect-MLP | 82.87±1.05 | 80.72±1.63 | **79.27±2.33** | 78.64±0.77 | **75.31±1.18** | 75.15±0.98 | **74.79±0.64** | 75.47±1.10 |

Table 6: The full robustness results on ACM and CoraML. The mean and std over five splits are reported. The top two performing models are highlighted in bold, with the best further underlined.

| | ACM (HR=0.821) | | | | CoraML (HR=0.784) | | | |
|---|---|---|---|---|---|---|---|---|
| | 5% | 10% | 15% | 20% | 5% | 10% | 15% | 20% |
| MLP | 87.44±0.28 | | | | 71.31±0.65 | | | |
| MLPw2 | 87.36±0.25 | | | | 70.69±1.78 | | | |
| MLPw4 | 87.39±0.69 | | | | 71.60±0.94 | | | |
| GCN | 80.15±3.46 | 75.65±4.94 | 71.26±5.70 | 68.47±6.36 | 81.78±0.82 | 77.5±0.72 | 73.27±1.99 | 66.54±3.17 |
| SGC | 82.46±2.36 | 80.12±2.12 | 77.75±2.37 | 74.70±1.72 | 78.54±1.37 | 72.94±1.44 | 67.34±1.85 | 58.88±3.70 |
| SAGE | 86.18±2.35 | 85.38±3.07 | 84.70±3.48 | 83.57±3.637 | 83.09±0.70 | 81.08±0.85 | 79.00±1.57 | 76.67±2.07 |
| RGCN | 79.90±1.80 | 74.59±3.62 | 71.07±3.81 | 67.51±3.30 | 81.94±0.81 | 77.14±0.77 | 73.35±1.57 | 67.02±2.78 |
| SVD | 85.98±1.36 | 84.98±0.81 | 83.28±1.82 | 82.10±1.77 | 80.61±0.61 | 79.76±0.99 | 78.84±0.65 | 77.42±0.97 |
| Jaccard | 80.15±3.46 | 75.65±4.94 | 71.26±5.70 | 68.47±6.36 | 81.50±0.77 | 77.84±0.56 | 74.63±1.20 | 69.81±2.26 |
| Guard | 76.22±2.36 | 71.63±6.86 | 70.20±8.07 | 65.78±7.01 | 76.66±0.85 | 76.75±0.93 | 76.47±0.82 | 76.35±0.63 |
| ProGNN | 79.89±2.01 | 77.80±1.98 | 74.77±4.35 | 69.13±7.62 | 81.87±1.20 | 78.27±1.25 | 75.86±2.23 | 70.49±3.76 |
| STABLE | 82.61±1.73 | 78.02±3.23 | 75.78±4.79 | 72.03±4.78 | 82.54±0.38 | 81.13±0.91 | 79.58±1.27 | 76.55±3.21 |
| EvenNet | 88.09±1.31 | 87.99±1.57 | 88.34±1.84 | 87.82±1.51 | **85.55±0.38** | 83.88±0.50 | 82.69±0.92 | 80.35±1.61 |
| GLNN | 89.37±1.09 | 89.24±1.38 | 88.65±2.29 | 88.48±1.50 | 83.82±0.66 | 82.14±0.92 | 80.29±1.91 | 77.87±1.74 |
| GLNNw2 | 90.29±1.05 | 90.11±1.56 | 89.31±1.41 | 88.84±1.83 | 84.19±0.69 | 82.46±1.01 | 80.43±1.90 | 78.31±1.89 |
| GLNNw4 | 89.72±1.42 | 90.31±1.12 | 89.35±1.58 | 88.16±2.56 | 84.32±0.50 | 82.76±0.85 | 80.45±2.43 | 79.00±2.14 |
| Prospect-SAGE | **92.03±1.27** | **91.68±0.84** | **91.24±0.65** | **90.93±1.06** | **85.18±0.61** | **84.29±0.70** | **83.19±0.54** | **82.34±0.80** |
| Prospect-MLP | **92.12±0.60** | **91.74±0.93** | **91.31±0.59** | **91.12±1.01** | 85.17±0.63 | **84.33±0.74** | **83.82±0.66** | 82.28±0.56 |

settings; **8)** flexible experiment monitoring and management. Once the paper is accepted, we will release our preattacked datasets, codes, and hyperparameters.

## G.2 MORE EXPERIMENT RESULTS

### G.2.1 FULL ADVERSARIAL ROBUSTNESS RESULTS

The full adversarial robustness evaluation results on five homophilous and three heterophilous graphs are shown in Tables 5, 6, 7, and 8. Since the MetaAttack provided by DeepRobust (Li et al., 2021) encounters an unknown error when the attack budget is 20% and the random seed is 18 on UAI, we skip UAI-Meta-20.

Table 7: The full robustness results on Polblogs and Texas. The mean and std over five splits are reported. The top two performing models are highlighted in bold, with the best further underlined.

| | Polblogs (HR=0.906) | | | | Texas (HR=0.061) | | | |
|---|---|---|---|---|---|---|---|---|
| | 5% | 10% | 15% | 20% | 5% | 10% | 15% | 20% |
| MLP | 52.21±0.61 | | | | 65.71±4.42 | | | |
| MLPw2 | 52.09±0.24 | | | | 67.21±3.02 | | | |
| MLPw4 | 51.72±0.87 | | | | 68.98±3.32 | | | |
| GCN | 77.18±1.76 | 71.64±1.64 | 67.53±0.99 | 66.07±1.05 | 49.25±5.43 | 46.67±4.48 | 49.39±2.29 | 46.53±7.16 |
| SGC | 77.71±1.79 | 70.74±2.52 | 66.95±1.36 | 64.58±1.34 | 53.88±2.23 | 53.20±2.12 | 55.24±2.12 | 53.88±2.82 |
| SAGE | 90.39±0.66 | 82.60±1.31 | 77.34±3.74 | 73.31±2.74 | 62.99±3.39 | 63.67±3.35 | 64.35±2.99 | 65.31±3.33 |
| RGCN | 75.42±1.29 | 69.18±2.36 | 66.18±0.64 | 65.03±0.87 | 52.93±1.89 | 51.43±3.10 | 49.52±8.10 | 51.84±3.89 |
| SVD | 92.43±0.70 | 86.46±3.46 | 73.44±1.77 | 69.69±2.28 | 49.66±4.02 | 46.12±7.13 | 48.57±5.66 | 52.38±5.02 |
| Jaccard | 50.88±1.69 | 50.88±1.69 | 50.88±1.69 | 50.88±1.69 | 49.25±5.43 | 46.67±4.48 | 49.39±2.29 | 46.53±7.16 |
| Guard | 51.58±0.57 | 51.58±0.57 | 51.58±0.57 | 51.58±0.57 | 48.03±12.96 | 49.66±7.47 | 47.76±11.40 | 42.45±12.54 |
| ProGNN | 85.97±5.16 | 78.71±5.51 | 72.78±3.43 | 69.14±2.09 | 47.89±10.06 | 46.67±6.19 | 45.31±14.47 | 49.12±4.57 |
| STABLE | 92.80±2.38 | 86.28±4.17 | 88.55±0.38 | 88.55±0.38 | 52.27±2.82 | 51.47±3.00 | 50.52±3.24 | 47.91±7.55 |
| EvenNet | 87.04±1.45 | **94.68±0.45** | 68.06±1.50 | 64.05±1.25 | 62.45±2.70 | 66.26±2.95 | 63.27±2.85 | 63.13±3.95 |
| GLNN | 91.62±1.35 | 83.52±2.32 | 77.46±3.73 | 73.35±2.65 | 66.40±2.57 | 67.35±6.14 | 66.53±5.45 | 68.44±5.10 |
| GLNNw2 | 91.55±1.11 | 83.37±2.59 | 77.14±3.98 | 73.21±2.85 | 66.67±2.93 | 67.62±6.17 | 66.12±5.49 | 69.66±4.82 |
| GLNNw4 | 91.19±1.41 | 84.27±2.49 | 77.12±3.58 | 73.25±2.73 | 67.21±2.70 | 67.48±6.58 | 66.40±4.92 | 68.03±4.74 |
| Prospect-SAGE | **93.95±1.34** | 92.92±1.07 | **92.27±2.26** | **92.09±0.57** | **68.84±5.65** | **69.66±1.64** | **71.02±2.30** | **72.25±2.90** |
| Prospect-MLP | **93.99±0.76** | **94.01±0.59** | **93.95±0.34** | **93.21±0.56** | **72.11±2.06** | **74.01±2.90** | **73.20±1.53** | **73.61±2.99** |

Table 8: The full robustness results on Chameleon and UAI. The mean and std over five splits are reported. The top two performing models are highlighted in bold, with the best further underlined.

| | Chameleon (HR=0.230) | | | | UAI (HR=0.364) | | |
|---|---|---|---|---|---|---|---|
| | 5% | 10% | 15% | 20% | 5% | 10% | 15% |
| MLP | 42.65±0.86 | | | | 61.74±2.11 | | |
| MLPw2 | 42.91±0.48 | | | | 64.13±1.40 | | |
| MLPw4 | 41.08±2.05 | | | | 62.71±1.91 | | |
| GCN | 36.30±3.98 | 31.70±3.28 | 28.91±1.56 | 27.39±1.95 | 56.72±4.68 | 55.47±3.56 | 54.22±3.17 |
| SGC | 34.92±3.12 | 29.78±2.42 | 27.39±3.31 | 27.26±3.30 | 58.78±3.34 | 58.42±3.69 | 56.52±2.64 |
| SAGE | 40.35±4.28 | 38.67±3.71 | 36.85±4.53 | 35.20±4.35 | 60.02±3.21 | 59.53±4.61 | 60.18±2.65 |
| RGCN | 37.53±3.77 | 31.43±1.62 | 29.57±2.52 | 28.34±2.29 | 49.89±2.85 | 50.00±3.249 | 48.40±2.74 |
| SVD | 39.36±2.03 | 32.22±4.23 | 29.66±2.41 | 28.42±2.19 | 48.65±1.14 | 47.24±1.53 | 44.87±1.18 |
| Jaccard | 41.16±1.51 | 40.76±0.63 | 36.71±2.63 | 37.97±0.96 | 54.08±4.18 | 51.99±8.29 | 50.64±2.69 |
| Guard | 39.55±2.19 | 38.61±2.26 | 36.88±1.57 | 36.13±1.51 | 20.28±10.99 | 18.26±3.07 | 20.36±8.27 |
| ProGNN | 37.20±1.93 | 31.92±3.20 | 29.78±3.06 | 28.77±2.74 | 49.22±5.22 | 46.25±3.99 | 38.43±11.55 |
| STABLE | 42.68±3.76 | 41.61±3.10 | 35.45±5.98 | 34.38±5.34 | 51.78±2.08 | 49.07±2.72 | 47.63±2.26 |
| EvenNet | 37.53±3.08 | 32.25±2.77 | 30.68±1.81 | 29.41±2.06 | 67.8±2.029 | 67.48±1.64 | 66.91±2.18 |
| GLNN | 39.00±3.12 | 36.99±3.40 | 37.14±3.91 | 35.57±4.10 | 62.46±2.91 | 62.64±3.18 | 62.02±2.00 |
| GLNNw2 | 39.62±3.26 | 37.31±3.23 | 35.41±5.28 | 33.80±5.84 | 62.89±3.10 | 63.11±2.77 | 63.15±1.22 |
| GLNNw4 | 39.20±3.31 | 37.59±3.18 | 35.57±5.12 | 34.75±5.02 | 62.62±2.39 | 62.62±3.21 | 62.75±1.99 |
| Prospect-SAGE | **47.89±0.69** | **46.30±1.80** | **46.27±1.16** | **45.69±1.04** | **69.86±0.58** | **69.06±0.50** | **69.52±0.46** |
| Prospect-MLP | **44.56±1.71** | **45.10±1.25** | **44.81±0.74** | **44.57±1.92** | **68.31±0.59** | **68.35±0.44** | **69.10±0.45** |

### G.2.2 SOME ADDITIONAL OBSERVATIONS

We have some interesting observations from the adversarial robustness and clean accuracy experiments. **Observation 1:** Remarkably, *SAGE demonstrates robustness higher than some defense methods.* This likely stems from the root projection, i.e., the first term in Eq. 2, that can retain clean node feature information, constraining poisoning from neighborhood aggregation affected by the poisoned structure. Our observation agrees with Zhu et al. (2022) attributing robustness to separate root and neighborhood embedding operators. **Observation 2:** GLNNs with SAGE teachers exhibit poisoning robustness approaching SAGE, as expected given their integration. Interestingly, *GLNNs even improve over teachers regarding adversarial robustness* in many cases, e.g., by ~2% on Citeseer-Meta-20 (see Table 5). This may result from the expressive power gap between GNNs and MLPs (Chen et al., 2021; Zhang et al., 2022), which makes the poisoned structure knowledge fail to fully transfer, allowing MLPs to receive less poisoned structure knowledge. **Observation 3:** *The clean accuracy gains of PROSPECT over standalone models, while present, are less pronounced than robustness improvements.* As shown by Tables 2, 5, 6, 7, and 8, Prospect-MLP and Prospect-

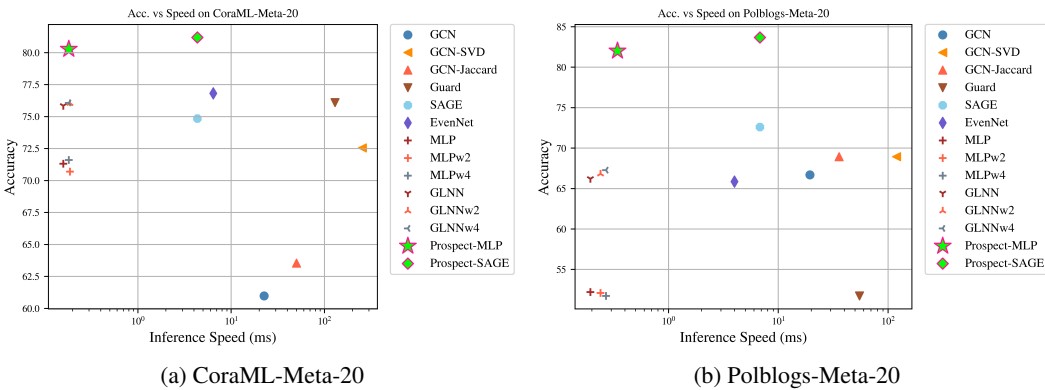

(a) CoraML-Meta-20          (b) Polblogs-Meta-20

Figure 7: Acc. vs. inference speed in the production setting. The x-axis is logarithmically scaled.

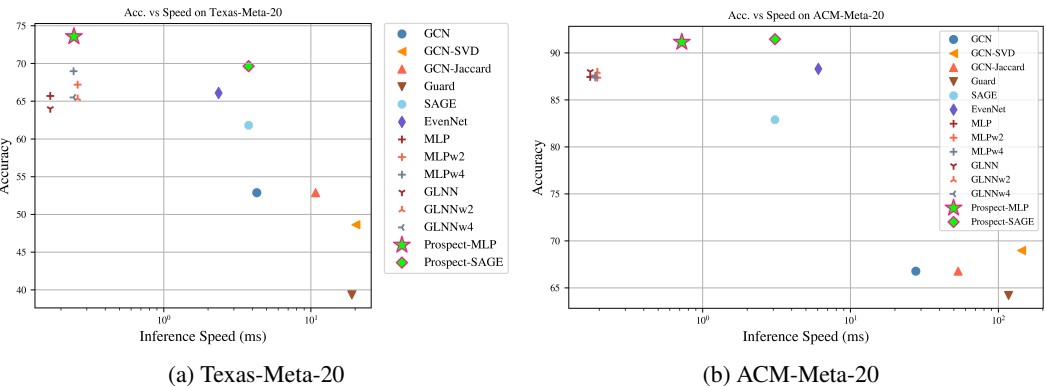

(a) Texas-Meta-20          (b) ACM-Meta-20

Figure 8: Acc. vs. inference speed in the production setting. The x-axis is logarithmically scaled.

SAGE improve less significantly over MLP and SAGE on clean graphs than attacked graphs. This aligns with Theorem 1 in the sense that the Prospect Condition of higher MLP ground-truth probability is less frequently met on clean graphs, limiting distillation benefits. But on attacked graphs, the degraded accuracy of SAGE makes satisfying Prospect Condition more easily, enabling more corrections from MLP-to-GNN distillation.

### G.2.3 Inference setting & Additional results of inference speed comparison

The semi-inductive setting is introduced in Appendix G.1. Following Zhang et al. (2022), for test nodes, 80% of them are for transductive testing, and 20% for inductive testing where the inductive nodes and their edges are withheld during training on the observed graph $\mathcal{G}_{obs} = (\mathbf{A}_{obs}, \mathbf{X}_{obs})$. And the full graph $\mathcal{G} = (\mathbf{A}, \mathbf{X})$ is only accessible at test time. We first attack the full graph with MetaAttack to get $\hat{\mathcal{G}} = \left(\hat{\mathbf{A}}, \mathbf{X}\right)$ and then hold out the elements associated to inductive nodes to get the observed training graph $\hat{\mathcal{G}}_{obs} = \left(\hat{\mathbf{A}}_{obs}, \mathbf{X}_{obs}\right)$. We train models on $\hat{\mathcal{G}}_{obs}$ supported by $\mathcal{V}_L \cup \mathcal{V}_{val} \cup \mathcal{V}_{test}^{trans}$ and test on $\hat{\mathcal{G}}$ supported by $\mathcal{V}_L \cup \mathcal{V}_{val} \cup \mathcal{V}_{test}^{trans} \cup \mathcal{V}_{test}^{ind}$ across all inference speed experiments. Though the GNNs in most defense methods are transductive and hence not naturally suitable for semi-inductive tasks, only ProGNN and RGCN completely violate such semi-inductive setting: ProGNN purifies $\hat{\mathcal{G}}_{obs}$ during training, unusable for unseen inductive test nodes; RGCN learns feature distributions for involved nodes during training, restricting robust inference to the observed nodes in $\hat{\mathcal{G}}_{obs}$. Excluding ProGNN and RGCN, we report the overall accuracy on $\mathcal{V}_{test}^{trans} \cup \mathcal{V}_{test}^{ind}$ versus the inference latency for 10 nodes on Cora/Citeseer/CoraML/Polblogs/Texas/ACM-Meta-20 in Figures 2, 7, and 8.

