# OpenReview forum: "PROSPECT: Learn MLPs Robust against Graph Adversarial Structure Attacks"
_ICLR.cc/2024/Conference — Submitted to ICLR 2024_

### Official Review · Reviewer_Gu8Y · 2023-10-21

**Soundness:** 3 good
**Presentation:** 1 poor
**Contribution:** 2 fair
**Rating:** 6
**Confidence:** 4

**Summary:**

This paper proposes an online GNN-MLP distillation framework, PROSPECT, that is able to handle heterophily and robust to graph structural attacks.
The online framework is enhanced via a specified learning rate scheduler.
The effectiveness of both the framework and scheduler is theoretically verified.
The robustness of PROSPECT is empirically validated via extensive experiments against structural attacks.

**Strengths:**

1. It is interesting to study how distillation methods perform under heterophily.
2. Good theoretical guarantees for the proposed methods.
3. Desirable robustness against structural attacks.

**Weaknesses:**

1. **About Motivation.** The most concern of mine is that the improvement with respect to robustness and performance under heterophily results from the introduction of distillation but not the proposed PROSPECT.
Based on the analysis, any GNN-MLP distillation framework seems to surpass a single GNN in the two fields.
While the advantage of PROSPECT over previous distillation methods regarding clean accuracy is slightly discussed on page 4, how PROSPECT excels against attacks and heterophily is unclear.
2. **Number of Hyperparams.** The whole framework, together with the scheduler, includes quite a few hyperparameters compared to the baselines.
It would be a concern if the hyperparameters varied a lot between different datasets.
Also, if the performance is highly affected by the hyperparameters.
3. **Baseline Missing.** A robust defense model that also tackles heterophily is not included. It would be better to have a comparison to it. [1]
4. **Presentation.** The current version is not friendly to those unfamiliar with distillation and graph adversarial attacks. It would be better if more details about distillation (like the online/offline settings) and a pseudo-code of PROSPECT were offered.

Other concerns are discussed in Questions.

[1] Deng, Chenhui, et al. "GARNET: Reduced-Rank Topology Learning for Robust and Scalable Graph Neural Networks." Learning on Graphs Conference. PMLR, 2022.

**Questions:**

1. **Training time.** While the framework shows the desirable inference efficiency, what about its training time?
2. **About Figure 1.** It is good to see that the proposed QACA works through the ablation study in Figure 1.
However, the fixed version seems noncompetitive against other defense baselines.
Is the QACA scheduler the one that actually contributes to the performance?
Can the previous distillation methods be enhanced by QACA as well?
3.  **Concerns when faced with other attacks.**
It makes sense that MLP is robust against structural attacks and acts as good teachers to GNNs, but when faced with feature attacks or graph injection attacks, the introduction of distillation could lead to worse performance.
As the robustness is only tested against Metattack, it would be more convincing if PROSPECT is tested under more attack settings.

---

> ### Author Response · Authors · 2023-11-21
> **Reponse (Part 1 Weakness 1)**
>
> * ***The benefits of GNN-to-MLP distillation.***
>
> Prior works, e.g., GLNN,first train a GNN teacher, then the GNN-to-MLP direction distillation is leveraged to transfer the knowledge from a pretrained GNN to a student MLP for fast inference. As shown by the clean accuracy results in Table 2, GLNN (i.e., GNN-to-MLP distillation)  cannot consistently improve the teacher SAGE (on heterophilic graphs) but succeeds in boosting the student MLP.  That is **GNN-to-MLP distillation is important to get a decent MLP w.r.t. clean accuracy.** The observation 2 in Appendix G.2.2 have revealed that GLNN often exceeds an alone SAGE teacher w.r.t. adversarial robustness, meaning that **GNN-to-MLP distillation can sometimes slightly mitigate the poisoning attacks.**
>
> * ***PROSPECT incorporates both GNN-to-MLP and MLP-to-GNN distillation.***
>
> Our PROSPECT simultaneously performs both GNN-to-MLP direction distillation and MLP-to-GNN direction distillation to jointly learn decent GNN and MLP. While the benefits of GNN-to-MLP distillation are shown by the above discussion, the benefits of MLP-to-GNN distillation is supported by Theorem 1. The goal of PROSPECT is to merge the knowledge from both MLP and GNN sides via these two distillation directions. Such knowledge fusion brings about desired effects, such as the robustness and adaptability to heterophily.
>
> * ***How PROSPECT excels GLNN and SAGE against (structure) attacks?*** Substantially by MLP-to-GNN distillation.
>
>   As stated by the paragraph between Remark1 and Remark2 (at p.5) and discussed by Observation 3 in Appendix G.2.2, the Prospect Condition is usually met for node $i$ under successful structure attacks that drastically degrade the SAGE performance but have no influence on the MLP due to its independence to graph structure.
>
>   One thing worth noting is that for the attacked node $i$, the MLP does NOT need to correctly classify it to be a good teacher because any successful structure attacks can always significantly decrease the SAGE classificaiton score on the ground-truth class $c$ of node $i$ such that even an MLP can output a higher score on class $c$. This makes the defense mechanism of PROSPECT more applicable.
>
> * ***How PROSPECT excels GLNN and SAGE against heterophily?*** By mutual distillation.
>
>   On one hand, as shown by Table 2 (at p.8), MLP can have decent performance on heterophilous graphs. On Texas and UAI, MLP even outperforms SAGE, meaning that MLP has some useful information to exploit. On the other hand, given their respective emphases on node attributes and topological structures, MLPs and GNNs are very likely to exhibit divergence in their correct prediction test sets V_{mlp} and  V_{gnn}.  So the union of these two sets (i.e., the knowledge fusion of MLP and GNN) V_{union} could lead to a larger correct prediction set than alone MLP or alone GNN. We have observed this phenomena across all clean and attacked (LLC) datasets used in the paper. Owing to the space limit, we only show the case on clean datasets in **Table A**. With the offline procedure, the SAGE teacher in GLNN can never acquire the knowledge from MLP. In contrast, the mutual disitllation mechanism of PROSPECT makes it possible to merge the knowledge from both GNN and MLP sides, and this is why PROSPECT excels alone MLP/SAGE and GLNN in clean (heterophily)  accuracy.
>
> According to the above analysis, the improvement of PROSPECT over GLNN (solely based on GNN-to-MLP distillation) and alone GNN results from the introduction of the neglected MLP-to-GNN distillation and the knowledge fusion achieved through mutual interaction.
>
>
> **Table A**: Test Acc. (%) on the correct test sets of MLP and GNN, and the union correct set, with seed=15
> |                       | cora  | citeseer | polblogs | acm   | cora_ml | texas | chameleon | uai   |
> | --------------------- | ----- | -------- | -------- | ----- | ------- | ----- | --------- | ----- |
> | $\mathcal{V}_{gnn}$   | 83.40 | 73.76    | 95.71    | 91.65 | 84.43   | 68.71 | 54.50     | 63.24 |
> | $\mathcal{V}_{mlp}$   | 62.78 | 66.23    | 52.04    | 85.91 | 68.42   | 68.03 | 40.23     | 65.04 |
> | $\mathcal{V}_{union}$ | 87.42 | 80.04    | 97.44    | 95.99 | 89.01   | 78.23 | 64.60     | 71.96 |

---

> ### Author Response · Authors · 2023-11-21
> **Reponse (Part 2 Weakness 2-4, Question 1)**
>
> **Weakness 2**: Number of Hyperparams.
>
> * ***The hyperparameters of PROSPECT:*** The hyperparameters introduced by PROSPECT include the loss weights for the logit distillation terms of the individual MLP and GNN optimization objectives, and the knowledge distillation temperatures of them. These are inevitable hyperparameters present in all mutual distillation architectures.
>
> * ***The hyperparameters of QACA:*** Although QACA contains hyperparameters including individual learning rates, weight decay values, minimum optimization cycle, and minimum learning rates of MLP and GNN, only the last three hyperparameters are actually introduced by the QACA scheduler. The other hyperparameters are originally present in the individual MLP and GNN models themselves.
>
> * ***Why these hyperparameters?*** **1)** The differences in these hyperparameters across datasets are primarily to accommodate the behaviour differences between the MLP and GNN on datasets with different characteristics (see **Table A** for a glance). How to design mutual distillation schemes between GNNs and MLPs that can self-adapt to different dataset characteristics is an important future research direction that would require a series of complete independent studies to elaborate (we are working on such a project). Adding this would make this already lengthy paper even more complex and diffuse in focus.
>
> **Weakness 3**: Baseline Missing.
>
> We have supplemented many experiments, including the comprehensive comparison between GPRGNN [1], GARNET [2] and our PROSPECT under 4 other attacks in addition to Metattack. Please refer to the **Appended Experiments** section of joint response above for details.
>
> **Weakness 4**: Presentation.
>
> * ***More details about graph distillation***
>   We can only add an introduction about offline and online graph distillation models to the appendix due to the length limit of main text.
> * ***A pseudo-code***
>   By aggregating the losses of the MLP and GNN components to derive an overall loss term loss, then invoking loss.backward(), the PROSPECT training procedure resembles optimizing a solitary model, except that the MLP and GNN each have their own optimizers and QACA scheduler. If the reviewer feels necessary, we can add a pseudocode in the appendix.
>
> **Question 1**: Training time.
> The MLP, GCN and SAGE can be formulated as $H=XW$, $H=AXW$ and $H=XW_1+AXW_2$  , respectively. And the basic operations to build them are  $Z=WX$ and $H=AZ$. Han et al. [3] have reported the forward and backward time of these two basic operations on serveral graphs. We extracte the data on Arxiv from the time comparison Tabel 1 in [3] into **Table B** below. According to the snippet, we can roughly estimate the training time of PROSPECT (that comprises one SAGE and one MLP), which shows that the additional training overhead incurred by PROSPECT over SAGE is almost negligible.
> Moreover, in the supplemented GR-BCD attack experiments on Arxiv (see the joint response), we report the training times of PROSPECT and GARNET. These results show that the training overhead of PROSPECT is almost the same as using GCN alone, and much lower than GARNET.
>
> **Table B**: Forward and backward times (ms). The first two rows are from Table 1 of [3], and the last two rows are the estimations based on the first rows.
> | Operation | Forward | Backward | Total   |
> | --------- | ------- | -------- | ------- |
> | $Z=WX$    | 0.32    | 1.09     | 1.42    |
> | $H=AZ$    | 1.09    | 1028.08  | 1029.17 |
> | SAGE      | 1.73    | 1030.26  | 1032.01 |
> | PROSPECT  | 2.05    | 1031.35  | 1033.43 |

---

> ### Author Response · Authors · 2023-11-21
> **Reponse (Part 3 Question 2-3)**
>
> **Question 2**: About Figure 1.
>
> * ***"However, the fixed version seems noncompetitive against other defense baselines. "***
>   As discussed in the **Contributions/The key to future mutual distillation** section of joint response, simply combining GNNs and MLPs for mutual distillation may easily lead to failure because of the model heterogeneity issue identified and (partially) solved by us.
>
> * ***"Is the QACA scheduler the one that actually contributes to the performance?"***
>   **1)** As discussed in **Weakness 1**, GNN-to-MLP and MLP-to-GNN distillation, i.e., the interaction between GNN and MLP contributes to the clean accuracy and robustness due to the accumulated knowledge post knowledge fusion (through mutual distillation). This gain has nothing to do with a specific optimization method. **2)** However, as pointed out in the **Contributions/The key to future mutual distillation** section of joint response, simply combining GNN and MLP can lead to failure and that's why we need a QACA optimization paradigm. **3)** In short, we think that the PROSPECT framework itself determines its capability, but this capability is locked. And QACA is just the key that opens the lock.*
> * ***"Can the previous distillation methods be enhanced by QACA as well?"***
>   **1)** QACA addresses underlying knowledge discrepancies within heterogeneous model distillation, hence offline frameworks, e.g.,GLNN, do not stand to benefit. Potentially, QACA mitigates conflicts even for general heterogeneous mutual distillation, but space constraints precluded detailed analysis beyond GNN-MLP. **2)** As pioneering GNN-MLP mutual distillation research, this paper validates QACA efficacy for this task. Intuitively, distilling between similar models unlikely exhibits severe conflicts, limiting QACA’s boons for GNN-GNN. However, QACA could assist future GNN-GNN models incorporating knowledge conflicts.
>
> **Question 3**: Concerns when faced with other attacks.
>
> * ***Graph injection attack***
>   Graph injection attacker, e.g., $G^2A2C$ [4], first adds a virtual node $a$ for the target node $i$ and then connect it to the original graph. During the neighborhood aggregation, the feature of node $i$ will be interfered such that its score on the correct category $c$ will decrease significantly. Now the case fails back to the first question in **Weakness 1** and the point becomes that whether the MLP is affected by node $a$ such that the MLP have a smaller score on $c$ than that of the GNN and thus becomes unqualified to be a teacher. Since the injected nodes are not used to train the MLP (no neighbor aggregation), they do not impact the MLP. Thus PROSPECT can work like under the structure attack, e.g., Metattack, as shown by our submission. Our supplementary experiments (see Section **Appended Experiments/Graph injection attacks** in the joint response) with the SOTA injection attack $G^2A2C$ demonstrate this.
> * ***Node feature attack***
>   We discuss the node feature attack scenario in the
>   **Concerns about joint, injection and adaptive attacks** section of joint response.
> * ***"It would be more convincing if PROSPECT is tested under more attack settings."***
>   To improve the persuasiveness regarding robustness, results across four additional attack methods have been included in the **Appended Experiments** section of the joint response.
>
> [1] E. Chien, et al., “Adaptive Universal Generalized PageRank Graph Neural Network,” ICLR'2021.
>
> [2] Deng, Chenhui, et al. "GARNET: Reduced-Rank Topology Learning for Robust and Scalable Graph Neural Networks," LoG'2022.
>
> [3] X. Han, T. Zhao, Y. Liu X. Hu, and N. Shah, “MLPInit: Embarrassingly Simple GNN Training Acceleration with MLP Initialization,” ICLR'2023.
>
> [4] M. Ju, et al. "Let Graph Be the Go Board: Gradient-Free Node Injection Attack for Graph Neural Networks via Reinforcement Learning," AAAI'2023.

---

> > ### Comment · Reviewer_Gu8Y · 2023-11-23
> >
> > Thank the authors for their efforts in clarification. Most of my concerns have been addressed. I would be glad to increase my score.

---

### Official Review · Reviewer_9crP · 2023-10-29

**Soundness:** 2 fair
**Presentation:** 3 good
**Contribution:** 2 fair
**Rating:** 3
**Confidence:** 4

**Summary:**

This work introduces PROSPECT, a graph distillation framework for learning robust GNNs/MLPs against graph adversarial attacks. Specifically, authors leverage two loss functions for GNN-to-MLP and MLP-to-GNN distillation. By alternately minimizing the loss functions, PROSPECT learns robust GNNs/MLPs that are resistant to graph attacks. Experimental results show that PROSPECT improves clean and adversarial accuracy over defense baselines on both homophilous and heterophilous graphs.

**Strengths:**

- Overall, the paper is well-written.
- Authors have evaluated PROSPECT on both homophilous and heterophilous datasets.

**Weaknesses:**

- Missing relevant defense models for evaluation. Authors mentioned that prior purification methods are computationally expensive and restricted to homophilous graphs. However, there are some recent studies (e.g., [1]) that have addressed these limitations. It would largely improve the paper to include comparisons against proper baselines.
- Missing adaptive attack results. As shown by [2], most prior defense GNN methods can be easily broken by adaptive attacks, which are aware of the given defense method during attacking. Thus, it is very important to adaptively attack PROSPECT to demonstrate its true robustness.
- The heterophilous datasets used in this work (e.g. Chameleon) are known to have some critical issues (e.g., train-test data leakage) [3]. Hence, the experimental results would be more compelling if authors could evaluate on the datasets from [3,4].
- Missing detailed hyperparameter settings for baselines. Note that many defense methods (e.g., ProGNN) require decent hyperparameter tuning to achieve their best performance. It is unclear whether authors put in enough efforts on tuning those baselines.
- Claim 3 is too strong. Given that PROSPECT is only integrated with GraphSAGE in this work, it is improper to claim PROSPECT can boost clean accuracy of GNNs, unless authors conduct more experiments with different GNNs integrated into PROSPECT.

[1]: Deng et al., “GARNET: Reduced-Rank Topology Learning for Robust and Scalable Graph Neural Networks”, LoG'22. \
[2]: Mujkanovic et al., “Are Defenses for Graph Neural Networks Robust?”, NeurIPS'22. \
[3]: Platonov et al., “A critical look at the evaluation of GNNs under heterophily: are we really making progress?”, ICLR'23. \
[4]: Lim et al., “Large Scale Learning on Non-Homophilous Graphs: New Benchmarks and Strong Simple Methods”, NeurIPS'21.

**Questions:**

- It does not seem reasonable to adopt GCN as the surrogate model for MetaAttack on heterophilous datasets. Why didn't the authors choose heterophilous GNNs as the surrogate model?
- Are baseline models trained with cosine annealing learning rate scheduler?

---

> ### Author Response · Authors · 2023-11-21
> **Response**
>
> Dear Reviewer 9crP,
>
> We really appreciate your comments. We hope our point-to-point response can address your concerns.
>
> **Weakness 1**: "Missing relevant defense models for evaluation."
>
> We thank the reviewer for the reminder and add GPRGNN [1] GARNET [2] baselines. GARNET is indeed the master of current graph purification methods, but its scalability is limited by the (truncated) SVD decomposition and is weaker than PROSPECT, as shown by the supplementary experiments (see Section **Appended Experiments/Arxiv attacked by GR-BCD** of the joint reponse).
>
> **Weakness 2**: "Missing adaptive attack results."
>
> We explain in the **Concerns about joint, injection and adaptive attacks/Adaptive attacks** section of joint response why the adaptive attack test is missing and add the experiments with the black-box unit test datasets provided by Mujkanovic et al. [3].
>
> **Weakness 3**: "The heterophilous datasets used in this work are known to have some critical issues."
>
> We thank the reviewer for pointing out this critical issue. After seeing the comments, we tried to make attack datasets with the dataset proposed in [4], but due to time and computing resource constraints, we were unable to supplement the corresponding experimental results during the discussion period. Therefore, we hope you can refer more to the results on UAI, a safe heterophilic data set.
>
> **Weakness 4**: "Missing detailed hyperparameter settings for baselines."
>
> Initially, we observed inconsistencies in current pre-attacked dataset creation paradigms. Specifically, certain works only utilize a sole train/validation/test split, risking less comprehensive evaluations. Hence, we strived to construct over 10 pre-attacked datasets under 4 attack budgets and 5 splits. During baseline hyperparameter tuning however, even abiding by original author guidelines, optimizing across the 5 splits proved challenging for some methods. We are benchmarking existing robust GNN techniques, tuning their hyperparameters accordingly, and will update performances if improvements emerge. Moreover, the diverse pre-processed attack datasets alongside all benchmark outcomes will be publicly disclosed in future endeavors.
>
> **Weakness 5**:  "Claim 3 is too strong."
>
> Thanks for pointing this out, we've narrowed the claims in Claim 3 (now renamed to Remark 3) to SAGE.
>
> **Question 1**:  "Why didn't the authors choose heterophilous GNNs as the surrogate model?"
>
> We also agree with that heterophilous GNNs seems more reasonable, but for the following reasons, we have temporarily compromised. **1)** We found that GCN, as a surrogate for heterophilic data, can actually produce effective attacks, so this does not prevent the evaluation of the defense model. **2)** Attacking under the unified agent model, namely GCN, can provide a reference benchmark for exploring attack and defense on heterophilic graphs. **3)** the MetaAttack and Nettack implementations from current tools, e.g., DeepRobust [5], do not support flexible proxy model settings.
>
> **Question 2**:  "Are baseline models trained with cosine annealing learning rate scheduler?"
>
> The baseline models are not trained with cosine annealing.
> The performance improvement of PROSPECT mainly comes from the knowledge fusion of GNN and MLP sides. QACA works by solving the knowledge conflict problem in the fusion process but it never brings additional knowledge to one alone GNN or MLP. That is, training SAGE and MLP individually with cosine annealing does not bring additional knowledge and thus is not necessary.
>
> [1] E. Chien, et al., “Adaptive Universal Generalized PageRank Graph Neural Network,” ICLR'2021.
>
> [2] C. Deng, et al., “GARNET: Reduced-Rank Topology Learning for Robust and Scalable Graph Neural Networks,” LoG'2022.
>
> [3] F. Mujkanovic, et al., “Are Defenses for Graph Neural Networks Robust?,” NeurIPS'2023.
>
> [4] O. Platonov, et al., “A critical look at the evaluation of GNNs under heterophily: Are we really making progress?,” ICLR'2023.
>
> [5] Y. Li, et al., "DeepRobust: A Platform for Adversarial Attacks and Defenses," AAAI' 2021.

---

> > ### Comment · Reviewer_9crP · 2023-11-22
> > **Follow-up**
> >
> > Thanks for the detailed response and additional clarifications. However, my major concern on the adaptive attack is still valid. Firstly, I hope authors could understand that adaptive attack is very important for assessing the model robustness, as demonstrated in [F. Mujkanovic, et al., NeurIPS'2023]. Thus, it is less convincing to claim the proposed model is robust when it's solely evaluated under the transfer attack setting. More importantly, I don't see any reason why adaptive attack is restricted to a single model. Isn't it true that gradient-based adaptive attacks (e.g., PGD and Meta-PGD used in [F. Mujkanovic, et al., NeurIPS'2023]) can be naturally adopted for the proposed model to maximize Equations 3a and 3b? Notably, it's defender's responsibility to design proper adaptive attacks on the proposed model for revealing its "true" robustness.

---

### Official Review · Reviewer_GM8H · 2023-11-01

**Soundness:** 3 good
**Presentation:** 3 good
**Contribution:** 2 fair
**Rating:** 5
**Confidence:** 3

**Summary:**

To address 1) low adaptability to heterophily, 2) absent generalizability to early GNNs, and 3) inadequate inference scalability of current defense methods for GNNs, the authors introduce PROSPECT, a defense framework incorporating an online mutual distillation approach between a GNN and an MLP, which enhances the performance on node-level classification tasks for both poisoned and clean graphs. Additionally, the authors apply a quasi-alternating cosine annealing (QACA) learning rate scheduler to improve the optimization process. The proposed approach is supported by detailed theoretical analysis and extensive empirical experiments to validate its effectiveness in mitigating the identified issues in GNNs.

**Strengths:**

- Incorporating the mutual distillation method into the existing GD-MLP to impart the MLP’s robustness to adversarial structure attack to GNNs is a concept of interest and significance in practice.

- The theoretical assessment of MLP-to-GNN distillation and QACA learning rate schedular is well discussed and convincing to me.

- Reducing inference time to the level of an MLP while improving the robustness to untargeted poisoning attacks and performance on clean graphs is both uplifting and commendable, with a series of empirical experiments conducted on both graphs of homophily and heterophily.

**Weaknesses:**

While the application of mutual distillation to the GD-MLP is inspiring, it’s worth noting that the novelty of the proposed approach appears somewhat limited, since there has been extensive research and discourse surrounding mutual distillation and distillation from GNNs to MLP. The absence of a specific design tailored to accommodate the unique characteristics of graph-structured networks might dilute the distinctiveness of this approach.

**Questions:**

- The empirical examination of PROSPECT's robustness against a specific untargeted poisoning attack, Metattack, is commendable. However, it could potentially enhance its persuasiveness by extending the experiment to include targeted attack scenarios and additional untargeted attack methods, such as GraD.

- Considering that PROSPECT is designed to be adaptable for various GNNs, it might be practical and insightful to explore its performance under more powerful GNN architectures, such as EvenNet and GPRGNN. This could offer insights into the performance limitations and strengths of the proposed framework.

---

> ### Author Response · Authors · 2023-11-21
> **Response**
>
> Dear Reviewer GM8H,
>
> We really appreciate your positive comments. We hope the response can address your concerns and clarify our contribution.
>
> **Weakness 1**: Limited novelty and absence of design tailored to GNN.
>
> In the joitn response at the top of the page, we reiterate and discuss the contribution and broader impact of our work in the **Contributions** section, hoping that the discussion can solve your concerns.
>
> **Question 1**: More attack scenarios.
>
> Following your suggestion, we have evaluated PROSPECT with 4 other kinds of attacks [1,2,3,4], including both untargeted and targeted ones. (The details of these new experiments are provided in Section **Appended Experiments** of the joint response.)
>
> **Question 2**: More backbones.
>
> We absolutely agree with your opinion and plan to conduct relevant experiments. However, due to time constraints, we deeply regret that we are unable to complete the experiments and provide results during the discussion period.
>
> [1] D. Zügner, A. Akbarnejad, and S. Günnemann, “Adversarial Attacks on Neural Networks for Graph Data,” KDD'2018.
>
> [2] M. Ju, Y. Fan, C. Zhang, and Y. Ye, “Let Graph Be the Go Board: Gradient-Free Node Injection Attack for Graph Neural Networks via Reinforcement Learning,” AAAI'2023.
>
> [3] F. Mujkanovic, S. Geisler, S. Günnemann, and A. Bojchevski, “Are Defenses for Graph Neural Networks Robust?,” NeurIPS'2023.
>
> [4] S. Geisler, et al., “Robustness of Graph Neural Networks at Scale,” NeurIPS'2021.

---

### Official Review · Reviewer_JnCK · 2023-11-01

**Soundness:** 2 fair
**Presentation:** 2 fair
**Contribution:** 3 good
**Rating:** 3
**Confidence:** 4

**Summary:**

The authors propose a bidirectional distillation method, called Prospect, between a GNN and MLP to mitigate the vulnerability of GNNs w.r.t. structure perturbations. The authors propose a custom learning schedule based on cosine annealing with warm restarts and prove some properties of their method using cSBMs. The authors demonstrate the effectiveness of their approach on homophilous and heterophilous datasets using a transfer poisoning attack.

**Strengths:**

1. Theoretical motivation of architecture on cSBMs
1. Prospect consistently outperforms the baselines in the empirical evaluation
1. Prospect is demonstrated to defend against adversarial transfer attacks on homophilic and heterophilic datasets
1. Prospect can handle scalable GNN architectures like GraphSAGE

**Weaknesses:**

1. The paper is full of overstating claims like "Theorem 1 implies the adversarial robustness of Prospect". While Theorem 1 might imply robustness on cSBMs, it is merely conjectured that this robustness extrapolates to real-world graphs.
1. Poisoning defense heavily uses the restricted threat model to solely perturb the graph structure and not the node features. It is one thing to follow the many other works in this simplifying assumption that focuses on the distinct characteristics of the robustness of GNNs; however, exploiting the clean node features seems highly questionable. The authors should discuss this and evaluate w.r.t., e.g., a joint attack on the graph structure and node features.
1. The empirical evaluation is insufficient: The authors solely evaluate using a non-adaptive transfer attack. As pointed out previously, it is vital to assess neural networks with adaptive attacks [C, D] to get a proper estimate of the model's robustness.
1. The authors claim scalability. Thus, they should compare to other works using large graphs [A, B]
1. Just because an MLP is robust w.r.t. structure perturbations does not imply it is useful. Moreover, there might be an interaction between the GNN and MLP. If the authors make a claim about evasion (e.g. second last line of page 3), they should also verify that empirically.
1. The presentation of the Theorems in the main part could be improved. It is unclear how the reader should deduce Theorem 1 from Proposition 1&2. Perhaps it would be better to move the propositions to the appendix and instead add a proof sketch to the main part.

[A] Adversarial attack on large scale graph, Li et al., IEEE Transactions on Knowledge and Data Engineering 2021.
[B] Robustness of Graph Neural Networks at Scale, Geisler et al., NeurIPS 2021
[C] On Adaptive Attacks to Adversarial Example Defenses, Carlini et al., NeurIPS 2020
[D] Are Defenses for Graph Neural Networks Robust?, Mujkanovic et al., NeurIPS 2022

**Questions:**

1. Can the authors please provide a full list of assumptions required for Proposition 1 & 2 as well as Theorem 1? For improved readability, the assumptions could be stated more explicitly and organized in the main text.
1. Could the theory be extended to further data-generating distributions like Barabasi–Albert? [E]

[E] Community recovery in a preferential attachment graph, Hajek and Sankagiri, IEEE Transactions on Information Theory 2019.

---

> ### Author Response · Authors · 2023-11-21
> **Response (Part 1 Weakness1-5)**
>
> Dear Reviewer JnCK,
>
> We really appreciate your comments. We hope our point-to-point response can address your concerns.
>
> **Weakness 1**: Overstating claims.
>
> We apologize that there were overstated claims in our original manuscript. We have re-examined the paper, eliminating overstated claims. We understand the importance of avoiding overstatements in academic writing, and your feedback has helped us improve the manuscript in this regard. Please let us know if there are any remaining issues with overclaiming and we will be happy to address them.
>
> **Weakness 2**: “The authors should discuss this and evaluate w.r.t., e.g., a joint attack on the graph structure and node features.”
>
> Please refer to the discussion section **Concerns about joint, injection and adaptive attacks** in the joint response.
>
> **Weakness 3**: “The empirical evaluation is insufficient.”
>
> We explained in the **Concerns about joint, injection and adaptive attacks/Adaptive attacks** section of joint response why the test with an adaptive attack is not performed at the first place. Due to the adaption issue pointed out here and urgent discussion deadline, we can only conduct part of black box unit tests provided by [A]. The experiment results can found in Section **Appended Experiments** of the joint response.
>
> **Weakness 4**: “The authors claim scalability. Thus, they should compare to other works using large graphs.”
>
> Although we have made every effort to supplement all experiments, time and computational constraints prevent us from conducting experiments on data beyond the pre-made large-scale perturbation datasets. We are so sorry that we only directly adopt the pre-attacked arxiv dataset from [C] and did not make pre-attacked datasets with the method from [B]. Nonetheless, we believe it is necessary to briefly introduce [B] in our paper.
>
> **Weakness 5.1**: “Just because an MLP is robust w.r.t. structure perturbations does not imply it is useful. Moreover, there might be an interaction between the GNN and MLP.”
>
> You did capture the key point of PROSPECT - the naive interactive behavior resulted from simply combining GNN and MLP for mutual distillation will cause the mutual distillation between these two types of heterogeneous models to fail, as stressed out in the **Contributions/The key to future mutual distillation** section of joint response.
>
> However, it should be noted that the performance gain of PROSPECT mainly comes from the fusion of the respective knowledge of GNN and MLP (regardless of attack or clean scenarios), so its theoretical performance has nothing to do with a specific optimization method like QACA.
>
> We think that the knowledge conflict caused by the model heterogeneity locks the performance of PROSPECT, and QACA is just the key to this lock.
>
> **Weakness 5.2**: “If the authors make a claim about evasion (e.g. second last line of page 3), they should also verify that empirically.”
>
> We apologize for misleading you with the imprecise statement on page 3. In fact, we want to say that PROSPECT-MLP trained by the PROSPECT framework does not require graph structure for inference, so it is completely robust to (structure) evasion attacks. Considering that both Prospect-MLP and Prospect-SAGE can be deployed for inference, we attribute this property to the PROSPECT framework.
>
> By the way, the performance of PROSPECT-MLP under evasion structure attacks is actually the same as that in the clean accuracy table, i.e., Table 2 of our paper.

---

> ### Author Response · Authors · 2023-11-21
> **Response (Part 2 Weakness6, Question 1-2 )**
>
> **Weakness 6**:"The presentation of the Theorems in the main part could be improved. It is unclear how the reader should deduce Theorem 1 from Proposition 1&2. Perhaps it would be better to move the propositions to the appendix and instead add a proof sketch to the main part."
>
> We are very sorry that we did not add proof schetch due to page limit. However, we are happy to add a proof sketch summarizing the high-level approach before formally proving Theorem 1, if you think this would further enhance the presentation.
>
> We give the main proof schetch of Theorem 1 and point out how Prop.1 and Prop.2 relate to the proof below.
>
> 1. Prop. 1 gives the distribution of processed features  $\mathbf{h}$ obtained by feeding the CSBM raw data to a SAGE layer.
> 2. Prop. 2 gives the ideal optimal decision boundary of the processed featuress $\mathbf{h}$.
> 3. Based on the gradients of the cross-entropy and KLD losses w.r.t. the SAGE layer parameters, we can get different SAGE weights that output different processed features with and without MLP-to-GNN distillation, denoted by $\mathbf{h}^{kd}$ and $\mathbf{h}$, respectively. In this process, the conclusion of Prop. 1 and some other auxiliary propositons in the appendix are required.
> 4. Per Prop.2, the optimal decision boundaries of $\mathbf{h}^{kd}$ and $\mathbf{h}$ can be derived. Then we comapre the (expected) distances of $\mathbf{h}_i^{kd}$ and $\mathbf{h}_i$ from their respective optimal decision boundaries (i.e., the mis-classified probabilities).
> 5. Prove that $\mathbf{h}_i^{kd}$ is more far away from the optimal decision boundary of  $\mathbf{h}^{kd}$. And thus the distillation is helpful to SAGE.
>
> As cSBM may be unfamiliar to some readers, keeping the propositions provides context about the data characteristics right before Theorem 1. So perhaps we should keep them before Theorem 1 in the main text. How do you think?
>
> **Question 1**: "Can the authors please provide a full list of assumptions required for Proposition 1 & 2 as well as Theorem 1?"
>
> Prop. 1 is the probabilistic description of the data generated by the CSBM model and the features obtained after oen SAGE layer. Prop. 2 gives the best decision boundary for the obtained features. Neither of these contains any assumptions per se.
>
> Regarding Theorem 1, the assumption about data is that the raw graph data are generated by a CSBM or aCSBM model. Another assumption implicit in the proof is that the learning rate is not too large and the number of nodes in the graph is not too small. As mentioned in the example on page 23, this is easily satisfied by almost all popular (and real-life) graph datasets and learning rate settings.
>
> **Question 2**: "Could the theory be extended to further data-generating distributions like Barabasi–Albert?"
>
> We examined the Barabasi–Albert data-generating distribution described in [D]. The biggest difference from the graph data generator used in our paper is probably that it does not contain node features, whereas Theorem 1 illustrates the benefits of MLP-to-GNN distillation by examining the distance between node features and the optimal decision boundary. Therefore, it cannot be directly generalized to the data generator you mentioned. Nevertheless, we guess that treating the category probability distribution of each node as its node feature may yield some interesting results with the Barabasi–Albert data-generating distribution.
>
> By the way, the key to generalizing Theorem 1 to other data-generating distributions is that you need to know and derive the distribution of node features after SAGE convolution under "different matrices of strictly positive affinities for vertices of different labels" [D].
>
> [A] F. Mujkanovic, et al., “Are Defenses for Graph Neural Networks Robust?,” NeurIPS'2023.
>
> [B] Li et al., "Adversarial attack on large scale graph,"  IEEE TKDE, 2021.
>
> [C] Geisler et al., "Robustness of Graph Neural Networks at Scale," NeurIPS'2021.
>
> [D] Community recovery in a preferential attachment graph, Hajek and Sankagiri, IEEE TIT, 2019.

---

> > ### Comment · Reviewer_JnCK · 2023-11-23
> > **Follow-up**
> >
> > I thank the authors for the exhaustive response and multitude of conducted experiments.
> >
> > I understand that the Prospect-MLP inherits perfect robustness w.r.t. evasion perturbations. However, if only deploying a Prospect-MLP it would be particularly important to understand its generalization capabilities in an *inductive* setting since the clean test nodes are known during training in a transductive setting w/o poisoning attack.
> >
> > > Therefore, current attack methods such as Nettack that can simultaneously attack structures and features will also spend the vast majority of the budget (probably over 90%) on graph topology perturbation, even under the joint attack setting.
> >
> > This finding might be true, e.g., for a GCN. However, this is no proof that this is also the most effective attack strategy on Prospect. Moreover, a GCN does not explicitly incorporate the assumption of clean node features to improve its robustness.
> >
> > It should be noted that applying an adaptive attack might not be simply applying an existing code base to a new defense. The authors should spend considerable effort on breaking their defense to increase confidence in its robustness.
> >
> > In summary, even though I think that Prospect could be a valuable contribution to the field, due to the missing adaptive evaluation and missing evaluation on feature perturbations, I tend to maintain my score.

---

### Author Response · Authors · 2023-11-21
**The Joint Response (Part 1 Contributions)**

Dear all Reviewers,

We really appreciate your time reading our paper and providing constructive comments. This comment is a **joint response** to some common concerns and presents the supplementary experiments. In addition to the efforts to address all the weaknesses and questions in a point-by-point manner, we have tried to improve this submission in the following ways.

* Emphasizing contributions and impact in Section **Contributions**.
* Explaining common concerns in this joint response.
* Adding comparison baselines (GPRGNN [1] and GARNET [2]).
* Including new robustness experiments under 4 other attacks in **Appended Experiments**.

## Contributions

* ***New thoughts***: Past research has shown that standalone MLPs struggle to compete with GNNs on graph node classification tasks, leading many researchers to overlook MLPs during the rapid development focused on improving GNNs. Although recent works like GLNN [3] and MLPInit [4] leverage MLPs to enable efficient inference and training, they still overlook the potential of MLPs (and the potential of merging MLP and GNN knowledge) for improving classification accuracy and defending against structural attacks, especially on heterophilic graphs. This paper validates these potentials, with the aim of re-examining the potential of MLPs for graph learning and providing new inspirations (GNN-MLP knowledge merging) for robust, scalable and accurate graph learning models.
* ***The key to future mutual distillation***: MLPs and GNNs are two very different types of models. Simply combining GNNs and MLPs in mutual distillation risks failure, as evidenced by the fixed learning rate baseline in Section 6.3 of our paper.  This work identifies this issue and contributes an effective, validated solution - the QACA scheduler. This may provide a basic optimization paradigm for future sophisticated, GNN-tailored mutual distillation models.
* ***Simple yet effective***: Although the framework and scheduler designed in this paper are simple in form, their effectiveness is supported by related theorems (Theorems 1 and 2) and extensive experiments. We believe that models which are simple in form, theoretically grounded, and deliver good empirical performance are no less innovative than more sophisticated, GNN-tailored models.
* ***Broader impact***:  This exploratory work is not to design sophisticated and complex models. Instead, the main goals are to explore the potential of MLPs (and that of merging the knowledge of MLP and GNN) for (robust and scalable) graph learning. This paper proposes an vanilla yet effective framework for GNN-MLP mutual distillation, and provides theoretical and empirical support, in order to furnish initial tools for more sophisticated GNN-MLP mutual distillation models in the future. Moreover, the derivation process of Theorem 1 can be generalized to, with a little effort, distilling models into GNNs including but not limited to SAGE, while Theorem 2 can be extended to more general scenarios of mutual distillation between heterogeneous models (e.g. the mutual distillation between GNNs and CNNs).

---

### Author Response · Authors · 2023-11-21
**The Joint Response (Part 2 Concerns about joint, injection and adaptive attacks)**

* ***Why only structure attacks are considered?*** The attackers tend not to perturb the node features.
  As introduced in the first paragraph of the Introduction, past works and our experience show that for GNN attack algorithms, the benefits of modifying node features are usually not as significant as modifying graph topology. Therefore, current attack methods such as Nettack that can simultaneously attack structures and features will also spend the vast majority of the budget (probably over 90%) on graph topology perturbation, even under the joint attack setting. This is why almost all defense works directly set the experimental setting to structure attacks.
* ***Graph injection attack***
  Graph injection attacker, e.g., $G^2A2C$ [5], first adds a virtual node $a$ for the target node $i$ and then connect it to the original graph. During the neighborhood aggregation, the feature of node $i$ will be interfered such that its score on the correct category $c$ will decrease significantly. As shown by Theorem 1, the robustness key of PROSPECT is that MLP can provide relatively clean information and the point becomes that whether the MLP is affected by node $a$ such that the MLP have a smaller score on $c$ than that of the GNN and thus becomes unqualified to provide clean information.  Since the injected nodes are not involved in training MLP (due to no neighbor aggregation), they have no effect on Prospect-MLP and thus PROSPECT can work like under non-injection structure attacks, e.g., Metattack. Our supplementary experiments (see **Appended Experiments/Graph injection attacks**) with the SOTA injection attack $G^2A2C$ support this.
* ***Whatif there is an attacker insists on attacking node features?***
  Although not very common and beyond the scope of this study, we can still have a discussion about this.  As discussed in the last point, the perturbed node features can be weakened by the neighborhood aggregation of GNNs. When facing the node feature attacks, e.g., $x_i\to \hat{x}_i$, the GNN side of PROSPECT can eliminate their influence and thus output roughly correct knowledge, i.e., a prediction distribution $p_i$ worth learning, to the engaged MLP, which will later learns to map $\hat{x}_i$ to $p_i$. If  $x_i$ and  $\hat{x}_i$ are too different, such an example will bias the learning of MLP and degrade its performance on other clean node samples. But thanks to the unnoticeable constraints of attackers,  $x_i$ and $\hat{x}_i$ are usually similar, the involved MLP is unlikely to suffer a (strong) bias. Due to the time constraint, we did not generate a convincing node feature attack dataset for experiments, but we plan to add this part of the experiment in future versions.
* ***Adaptive attacks***

  * ***Why not adaptive?*** For the following two reasons, we did not add discussions about adaptive attacks in the original version: **1)** Adaptive attacks require perfect knowledge that is not common in practice, while PROSPECT targets more practical robust graph learning problems; **2)** The design steps of adaptive attacks [6] actually have an assumption that the defense model is a single one. When designing adaptive attacks against PROSPECT which contains multiple submodels, it is an open question which submodel's gradient should be exploited to guide the attack. If exploiting multiple submodels, how to fuse the (gradient) information from different submodels to instruct the adaptive attack also needs thorough discussion and validation from theoretical and empirical perspectives. This likely warrants a full paper rather than fitting within the scope of this text.
  * ***A compromised "adaptive" attack***  In view of the relevant suggestions made by many reviewers, although the adaptive attack on PROSPECT cannot be directly designed with the framework of Mujkanovic et al. [6], we used the two black box unit test data sets provided by them for testing. As each of these unit tests constists of a bunch of structure attacks generated with various (robust) GNN surrogates and part of them, e.g., GCN, have similar behaviours to the GNN part of PROSPECT, we take the test as an approximated adaptive attack test. The experiment details are in Section **Appended Experiments/"Adaptive" attacks** below.

---

### Author Response · Authors · 2023-11-21
**The Joint Response (Part 3 Appended Experiments)**

## Appended Experiments

### Arxiv attacked by GR-BCD

We evaluate with the clean and pre-attacked Arxiv dataset provided by [7], and the data can be found at [this repo](https://github.com/sigeisler/robustness_of_gnns_at_scale).

**App. Table 1**: Test Acc. (%) on (attacked) Arxiv. In parentheses is the the total training time (s).
|               | Arxiv        | Arxiv-GRBCD-0.05 | Arxiv-GRBCD-0.1 |
| ------------- | ------------ | ---------------- | --------------- |
| GCN           | 72.05 (30)   | 48.39 (47)       | 42.8 (46)       |
| GCN-GARNET    | 68.12 (1580) | 60.07 (1161)     | 60.05 (1186)    |
| Prospect-SAGE | 70.56 (110)  | 64.71 (161)      | 63.33 (138)     |
| Prospect-MLP  | 70.99  (110) | 65.12 (161)      | 64.23 (138)     |

### "Adaptive" attacks

The black box unit test datasets are under the `unit_test` directory of the [official repo](https://github.com/LoadingByte/are-gnn-defenses-robust). We attempted to generate new attacked datasets using the repository but could not find a convenient utility. Therefore, only the datasets originally provided by the authors are utilized here.

Originally, we intended to adopt RAUC as an evaluation metric. However, an official implementation could not be identified within the intricate codebase. We proceeded to implement a RAUC computation function and test this function with the results shown by Figures 2&3 from the original paper [6], but failed to match the results from these two figures. To preclude potential biases from our own (probably wrong) RAUC implementation, we opted to employ accuracy as the metric instead.

We evaluated our defense against attacks generated from different surrogate models under the maximum attack budget on each dataset.

**App. Table 2**: Test Acc. (%) on Citeseer from [6]. The rows are models to transfer attacks to while the columns are surrogates.
|               | GCN   | Jaccard-GCN | SVD-GCN | RGCN  | ProGNN | GNN-Guard | GRAND | SoftMedianGDC |
| ------------- | ----- | ----------- | ------- | ----- | ------ | --------- | ----- | ------------- |
| GCN           | 63.51 | 67.36       | 72.75   | 60.31 | 62.56  | 68.31     | 61.37 | 55.33         |
| SAGE          | 56.87 | 53.67       | 71.45   | 49.35 | 50.18  | 57.17     | 51.9  | 51.84         |
| GPR           | 67.83 | 68.66       | 74.53   | 67.06 | 67.83  | 68.9      | 70.79 | 67.24         |
| GCN-GARNET    | 66.77 | 66.41       | 72.33   | 58.95 | 58.65  | 65.28     | 63.98 | 57.29         |
| GPR-GARNET    | 66.71 | 69.02       | 74.05   | 66.59 | 68.78  | 70.97     | 70.73 | 66.59         |
| EvenNet       | 58.59 | 55.92       | 72.57   | 51.18 | 53.14  | 63.03     | 49.47 | 49.11         |
| Prospect-SAGE | 72.93 | 72.63       | 72.57   | 72.69 | 73.52  | 72.57     | 72.93 | 72.87         |
| Prospect-MLP  | 71.33 | 71.09       | 72.51   | 71.68 | 70.62  | 71.74     | 71.21 | 72.93         |

**App. Table 3**: Test Acc. (%) on CoraML from [6]. The rows are models to transfer attacks to while the columns are surrogates.

|               | GCN   | Jaccard-GCN | SVD-GCN | RGCN  | ProGNN | GNN-Guard | GRAND | SoftMedianGDC |
| ------------- | ----- | ----------- | ------- | ----- | ------ | --------- | ----- | ------------- |
| GCN           | 72.13 | 72.74       | 75.96   | 70.42 | 68.86  | 75.35     | 70.02 | 62.78         |
| SAGE          | 65.74 | 67.51       | 77.46   | 56.69 | 55.28  | 71.73     | 58.12 | 58.23         |
| GPR           | 79.93 | 80.23       | 79.33   | 79.63 | 76.16  | 79.02     | 79.33 | 76.56         |
| GCN-GARNET    | 71.88 | 73.04       | 75.15   | 71.13 | 66.05  | 75.25     | 69.87 | 63.23         |
| GPR-GARNET    | 80.48 | 79.18       | 82.44   | 78.52 | 75.15  | 80.08     | 78.62 | 75.8          |
| EvenNet       | 64.84 | 69.27       | 83.50   | 60.97 | 56.29  | 78.57     | 61.62 | 62.93         |
| Prospect-SAGE | 80.94 | 78.42       | 80.13   | 80.68 | 79.38  | 79.48     | 79.18 | 79.23         |
| Prospect-MLP  | 68.76 | 69.22       | 69.67   | 68.16 | 67.61  | 69.11     | 67.76 | 68.51         |

---

### Author Response · Authors · 2023-11-21
**The Joint Response (Part 3 Appended Experiments)**

### Graph injection attacks

We evaluate the models using the state-of-the-art $G^2A2C$ injection attack [5] on the Cora, Citeseer, and Pubmed datasets. Note that the data splits utilized originate from the public splits in GCN [8]. Following the methodology of [5], for every test node correctly classified by a 2-layer GCN [8], we inject one adversarial node. Since this attack introduces noisy node features, we additionally present the results of the GARNET variant (denoted GARNET*) using TSVD on node features as discussed in Appendix L.2 of [2].


**App. Table 4**: Misclassification rate (%) on the datasets attacked by $G^2A2C$ [5]
|               | Cora-G2A2C | Citeseer-G2A2C | Pubmed-G2A2C |
| ------------- | ---------- | -------------- | ------------ |
| GCN           | 20.68±1.0  | 37.84±0.59     | 42.98±2.81   |
| SAGE          | 21.30±0.79 | 34.54±2.14     | 48.20±4.55   |
| GPR           | 18.62±0.62 | 31.22±0.99     | 38.44±1.10   |
| GCN-GARNET    | 18.06±0.48 | 33.9±0.68      | 25.08±0.39   |
| GCN-GARNET\*  | 18.92±0.95 | 30.54±0.42     | 46.22±7.66   |
| GPR-GARNET    | 18.28±1.03 | 31.42±0.48     | 25.42±0.60   |
| GPR-GARNET\*  | 19.54±0.46 | 29.84±0.77     | 32.76±0.88   |
| EvenNet       | 18.26±0.83 | 31.06±0.36     | 22.34±0.34   |
| Prospect-SAGE | 17.96±0.50 | 26.46±1.26     | 22.60±0.46   |
| Prospect-MLP  | 18.04±0.39 | 26.32±0.52     | 22.82±1.54   |

### Nettack

For Nettack, we adopt the settings of Tables 2&3 from [2]. We append the results of SAGE, EvenNet [9], and our PROSPECT.

**App. Table 5**: Test Acc. (%) with the Nettack setting used in Table 2 from [2]
|               | Cora-Nettack-5 | Pubmed-Nettack-5 |
| ------------- | -------------- | ---------------- |
| GCN           | 55.66±1.95     | 66.67±1.34       |
| SAGE          | 60.24±1.7      | 79.46±3.55       |
| SVD           | 60.30±2.25     | 79.56±0.48       |
| ProGNN        | 65.38±1.65     | 71.89±1.56       |
| LFR           | 53.73±2.17     | 68.49±2.44       |
| GCN-GARNET    | 67.04±2.05     | 86.12±0.86       |
| GPR           | 62.89±1.95     | 76.99±1.16       |
| GPR-SVD       | 63.52±3.27     | OOM              |
| GPR-ProGNN    | 63.74±2.57     | OOM              |
| GPR-GARNET    | 71.45±2.73     | 89.52±0.45       |
| EvenNet       | 66.27±0.85     | 68.17±1.34       |
| Prospect-SAGE | 71.08±2.16     | 88.01±0.25       |
| Prospect-MLP  | 71.33±1.60     | 89.57±0.26       |

**App. Table 6**: Test Acc. (%) with the Nettack setting used in Table 3 from [2]
|               | Chameleon-Nettack-5 | Squirrel-Nettack-5 |
| ------------- | ------------------- | ------------------ |
| SAGE          | 65.12±1.39          | 32.91±2.17         |
| GPR           | 66.26±1.71          | 39.45±2.36         |
| GPR-SVD       | 60.37±2.86          | 31.20±1.84         |
| GPR-ProGNN    | 57.07±1.82          | 27.27±1.87         |
| GPR-GARNET    | 71.83±2.11          | 43.64±1.53         |
| EvenNet       | 68.29±0.86          | 29.27±2.17         |
| Prospect-SAGE | 71.10±2.62          | 41.36±3.31         |
| Prospect-MLP  | 58.41±2.91          | 38.73±2.85         |

---

### Author Response · Authors · 2023-11-21
**The Joint Response (Part 3 Appended Experiments)**

### Mettack

Since the Metattack datasets proposed in our submission is more comprehensive, we additionally assess GPRGNN and GARNET on these datasets for comparison. Cora-Meta-10 means the Cora dataset attacked by Metattack with a budget of 10%.

**App. Table 7**: Test Acc. (%) on the Metattack datasets from our submission.

|               | Cora-Clean | Cora-Meta-10 | Cora-Meta-20 | Cite-Clean | Cite-Meta-10 | Cite-Meta-20 | Chameleon-Clean | Chameleon-Meta-10 | Chameleon-Meta-20 |
| ------------- | ---------- | ------------ | ------------ | ---------- | ------------ | ------------ | --------------- | ----------------- | ----------------- |
| MLP           | 65.69±1.43 | 65.69±1.43   | 65.69±1.43   | 66.01±1.37 | 66.01±1.37   | 66.01±1.37   | 42.65±0.86      | 42.65±0.86        | 42.65±0.86        |
| MLPw4         | 67.02±1.12 | 67.02±1.12   | 67.02±1.12   | 66.43±1.73 | 66.43±1.73   | 66.43±1.73   | 41.08±2.05      | 41.08±2.05        | 41.08±2.05        |
| SAGE          | 83.56±0.86 | 77.20±1.90   | 71.39±2.74   | 74.53±1.01 | 72.29±0.52   | 67.97±2.31   | 51.66±2.21      | 38.67±3.71        | 35.20±4.35        |
| STABLE        | 83.09±0.58 | 80.09±0.38   | 77.58±1.53   | 74.44±0.56 | 73.95±0.74   | 72.50±0.88   | 46.66±1.57      | 41.61±3.10        | 34.38±5.34        |
| EvenNet       | 84.89±0.35 | 80.80±0.63   | 75.14±1.79   | 74.46±0.80 | 72.97±1.19   | 69.40±1.14   | 51.73±1.22      | 32.25±2.77        | 29.41±2.06        |
| GPR           | 84.57±0.57 | 79.74±0.32   | 75.21±0.97   | 74.88±1.23 | 72.3±1.28    | 67.83±1.8    | 51.30±0.87      | 35.5±3.95         | 34.28±4.45        |
| GCN-GARNET    | 82.39±1.24 | 78.04±0.75   | 75.05±0.9    | 72.63±0.92 | 72.11±1.5    | 70.05±2.03   | 56.15±1.65      | 40.53±2.47        | 36.41±4.18        |
| GPR-GARNET    | 83.75±0.83 | 79.64±1.22   | 80.44±0.94   | 74.67±0.70 | 74.68±1.15   | 73.7±1.30    | 46.73±1.86      | 40.53±2.45        | 38.42±1.46        |
| GLNNw4        | 80.21±1.51 | 77.34±1.87   | 72.76±2.82   | 75.60±0.52 | 73.37±0.71   | 70.37±2.72   | 49.36±2.07      | 37.59±3.18        | 34.75±5.02        |
| Prospect-SAGE | 83.05±1.05 | 81.13±1.45   | 79.59±1.00   | 75.20±0.70 | 75.23±1.21   | 75.87±0.96   | 55.88±1.12      | 46.30±1.80        | 45.69±1.04        |
| Prospect-MLP  | 82.87±1.05 | 80.72±1.63   | 78.64±0.77   | 75.81±0.68 | 75.15±0.98   | 75.47±1.10   | 53.43±1.45      | 45.10±1.25        | 44.57±1.92        |

## Reference
[1] E. Chien, et al., “Adaptive Universal Generalized PageRank Graph Neural Network,” ICLR'2021.

[2] C. Deng, et al., "GARNET: Reduced-Rank Topology Learning for Robust and Scalable Graph Neural Networks," LoG'2022.

[3] S. Zhang, Y. Liu, Y. Sun, and N. Shah, “Graph-less Neural Networks: Teaching Old MLPs New Tricks via Distillation,” ICLR'2022.

[4] X. Han, T. Zhao, Y. Liu X. Hu, and N. Shah, “MLPInit: Embarrassingly Simple GNN Training Acceleration with MLP Initialization,” ICLR'2023.

[5] M. Ju, Y. Fan, C. Zhang, and Y. Ye, “Let Graph Be the Go Board: Gradient-Free Node Injection Attack for Graph Neural Networks via Reinforcement Learning,” AAAI'2023.

[6] F. Mujkanovic, S. Geisler, S. Günnemann, and A. Bojchevski, “Are Defenses for Graph Neural Networks Robust?,” NeurIPS'2023.

[7] S. Geisler, et al., “Robustness of Graph Neural Networks at Scale,” NeurIPS'2021.

[8] T. N. Kipf and M. Welling, “Semi-supervised classification with graph convolutional networks,” ICLR'2017.

[9] R. Lei, et al., “EvenNet: Ignoring odd-hop neighbors improves robustness of graph neural networks,” NeurIPS'2022.

---

### Meta-Review · Area_Chair_iYWg · 2023-12-14

**Metareview:**

This paper introduces PROSPECT, a defense framework against graph adversarial attacks by combining GNNs and MLPs through distillation. Although the idea is interesting and the authors are able to show robustness guarantees (under strong assumptions), reviewers raised concerns about the experimental evaluations. While some concerns were addressed, the critical issue of evaluating resilience against adaptive attacks remained unresolved following the discussion phase. Reviewers think this a crucial missing element and thus we decide to reject the paper. We encourage the authors to conduct further experiments on adaptive attacks and test PROSPECT on larger graphs in future iterations.

**Justification For Why Not Higher Score:**

Lacking adaptive attack, and lacking experiments on larger graphs (see the meta review above).

**Justification For Why Not Lower Score:**

N/A

---

### Decision · Program_Chairs · 2024-01-16

Reject